# GWAS and meta-analysis identifies 49 genetic variants underlying critical COVID-19

Erola Pairo-Castineira[1,2,3,564], Konrad Rawlik[1,564], Andrew D. Bretherick[1,2,4], Ting Qi[5,6], Yang Wu[7], Isar Nassiri[8], Glenn A. McConkey[9], Marie Zechner[1,3], Lucija Klaric[2], Fiona Griffiths[1,3], Wilna Oosthuyzen[1,3], Athanasios Kousathanas[10], Anne Richmond[2], Jonathan Millar[1,3,11], Clark D. Russell[1], Tomas Malinauskas[8], Ryan Thwaites[12], Kirstie Morrice[13], Sean Keating[11], David Maslove[14], Alistair Nichol[15], Malcolm G. Semple[16,17], Julian Knight[8], Manu Shankar-Hari[11,18], Charlotte Summers[19], Charles Hinds[20], Peter Horby[21], Lowell Ling[22], Danny McAuley[23,24], Hugh Montgomery[25], Peter J. M. Openshaw[12,26], Colin Begg[27], Timothy Walsh[11], Albert Tenesa[2,3,28], Carlos Flores[29,30,31,32], José A. Riancho[33,34,35], Augusto Rojas-Martinez[36], Pablo Lapunzina[37,38,39], GenOMICC Investigators*, SCOURGE Consortium*, ISARICC Investigators*, The 23andMe COVID-19 Team*, Jian Yang[5,6], Chris P. Ponting[2], James F. Wilson[2,28], Veronique Vitart[2], Malak Abedalthagafi[40,41], Andre D. Luchessi[42,43], Esteban J. Parra[43], Raquel Cruz[37,44], Angel Carracedo[37,44,45,46], Angie Fawkes[13], Lee Murphy[13], Kathy Rowan[47], Alexandre C. Pereira[48], Andy Law[3], Benjamin Fairfax[8], Sara Clohisey Hendry[1,3] & J. Kenneth Baillie[1,2,3,11 ✉]

Critical illness in COVID-19 is an extreme and clinically homogeneous disease phenotype that we have previously shown[1] to be highly efficient for discovery of genetic associations[2]. Despite the advanced stage of illness at presentation, we have shown that host genetics in patients who are critically ill with COVID-19 can identify immunomodulatory therapies with strong beneficial effects in this group[3]. Here we analyse 24,202 cases of COVID-19 with critical illness comprising a combination of microarray genotype and whole-genome sequencing data from cases of critical illness in the international GenOMICC (11,440 cases) study, combined with other studies recruiting hospitalized patients with a strong focus on severe and critical disease: ISARIC4C (676 cases) and the SCOURGE consortium (5,934 cases). To put these results in the context of existing work, we conduct a meta-analysis of the new GenOMICC genome-wide association study (GWAS) results with previously published data. We find 49 genome-wide significant associations, of which 16 have not been reported previously. To investigate the therapeutic implications of these findings, we infer the structural consequences of protein-coding variants, and combine our GWAS results with gene expression data using a monocyte transcriptome-wide association study (TWAS) model, as well as gene and protein expression using Mendelian randomization. We identify potentially druggable targets in multiple systems, including inflammatory signalling (*JAK1*), monocyte–macrophage activation and endothelial permeability (*PDE4A*), immunometabolism (*SLC2A5* and *AK5*), and host factors required for viral entry and replication (*TMPRSS2* and *RAB2A*).

The design of the GenOMICC study and the rationale for focusing on critical illness has been previously described[1,2]. In brief, patients with confirmed COVID-19 requiring continuous cardiorespiratory monitoring or organ support (a generalizable definition for critical illness) were recruited in 2020–2022. We first performed ancestry-specific GWAS analyses according to the methods that we described previously[1,2]. Using the results of these GWAS analyses, previously reported results obtained using GenOMICC participants with whole-genome sequencing data[2] and data from GenOMICC Brazil, we performed trans-ancestry and -platform meta-analyses within the GenOMICC study for a critically ill COVID-19 phenotype and a hospitalized COVID-19 phenotype (Extended Data Fig. 1). The results of these GenOMICC-only meta-analyses are presented for both critically ill and hospitalized phenotypes (Table 1 and Extended Data Fig. 2). To put these results into the context of existing knowledge, we performed comprehensive meta-analyses, drawing on further GWAS results, including data shared by the SCOURGE consortium and published data from the COVID-19 Human Genetics Initiative (HGIv6, 2021)[4]. The characteristics of the contributing studies are summarized in Supplementary Tables 13 and 14 for the critically ill and hospitalized phenotypes, with further details on each study provided in

---

**Table 1 | Genome-wide significant associations with critical COVID-19, listing independent lead variants**

| Chr:pos(b38) | rsid | EA | OR | OR$_{CI}$ | P | P$_{cond}$ | Nearest gene | Citation |
|---|---|---|---|---|---|---|---|---|
| 1:9067157 | rs2478868 | A | 0.92 | 0.9–0.95 | $1.5 \times 10^{-10}$ | | SLC2A5 | GenOMICC[new] |
| 1:64948270 | rs12046291 | A | 1.1 | 1.07–1.13 | $5.1 \times 10^{-11}$ | | JAK1 | GenOMICC[new] |
| 1:77501822 | rs71658797 | A | 1.1 | 1.09–1.18 | $2.8 \times 10^{-9}$ | | AK5 | GenOMICC[new] |
| 1:155066988 | rs114301457 | T | 2.4 | 1.81–3.18 | $1.5 \times 10^{-9}$ | | EFNA4 | GenOMICC[2] |
| 1:155175305 | rs7528026 | A | 1.3 | 1.25–1.44 | $1.5 \times 10^{-15}$ | | TRIM46 | GenOMICC[2] |
| 1:155197995 | rs41264915 | A | 1.2 | 1.17–1.26 | $7.6 \times 10^{-24}$ | | THBS3 | HGI[18] |
| 1:155278322 | rs11264349 | A | 0.94 | 0.92–0.97 | $7.3 \times 10^{-5}$ | $3.9 \times 10^{-13}$ | HCN3 | GenOMICC[new] |
| 2:60480453 | rs1123573 | A | 1.1 | 1.09–1.15 | $1 \times 10^{-14}$ | | BCL11A | GenOMICC[2] |
| 3:45796521 | rs2271616 | T | 1.2 | 1.14–1.23 | $1.1 \times 10^{-16}$ | | SLC6A20 | HGI[18] |
| 3:45818159 | rs17713054* | A | 2 | 1.96–2.13 | $7.7 \times 10^{-254}$ | | LZTFL1 | SCGG[19] |
| 3:45873093 | rs35482426 | CTT | 0.53 | 0.5–0.57 | $6.1 \times 10^{-91}$ | | LZTFL1 | SCGG[19] |
| 3:101790631 | rs11706494 | A | 1.1 | 1.05–1.11 | $9.4 \times 10^{-9}$ | | NXPE3 | GenOMICC[new] |
| 3:146522652 | rs343314 | T | 1.2 | 1.09–1.21 | $4.6 \times 10^{-8}$ | | PLSCR1 | GenOMICC[2] |
| 4:25446871 | rs7664615 | A | 1.1 | 1.07–1.14 | $1.5 \times 10^{-8}$ | | ANAPC4 | GenOMICC[new] |
| 4:105673359 | rs72670002 | A | 1.1 | 1.09–1.21 | $4.4 \times 10^{-8}$ | | ARHGEF38 | GenOMICC[new] |
| 4:167824478 | rs1073165 | A | 1.1 | 1.05–1.11 | $1.1 \times 10^{-9}$ | | – | GenOMICC[new] |
| 5:132422622 | rs2269821 | A | 1.1 | 1.08–1.16 | $3 \times 10^{-10}$ | | IRF1-AS1 | GenOMICC[2] |
| 6:31153455 | rs111837807 | T | 0.8 | 0.77–0.84 | $8.6 \times 10^{-26}$ | | CCHCR1 | GenOMICC[1] |
| 6:31571991 | rs2071590 | A | 1.1 | 1.06–1.11 | $3.1 \times 10^{-10}$ | | LTA | GenOMICC[new] |
| 6:32702687 | rs2858305 | T | 0.93 | 0.9–0.95 | $2.1 \times 10^{-9}$ | | HLA-DQA1 | GenOMICC[2] |
| 6:41522644 | rs41435745 | C | 1.4 | 1.31–1.51 | $1.5 \times 10^{-20}$ | | FOXP4 | HGI[18] |
| 7:75623396 | rs1179620 | T | 0.92 | 0.9–0.95 | $2.3 \times 10^{-9}$ | | HIP1 | GenOMICC[new] |
| 7:100032719 | rs2897075 | T | 1.1 | 1.06–1.11 | $8.9 \times 10^{-11}$ | | ZKSCAN1 | GenOMICC[new] |
| 8:60532539 | rs13276831 | T | 1.1 | 1.05–1.1 | $1.7 \times 10^{-8}$ | | RAB2A | GenOMICC[new] |
| 9:21206606 | rs28368148 | C | 0.59 | 0.49–0.7 | $5.3 \times 10^{-9}$ | | IFNA10 | GenOMICC[2] |
| 9:33425186 | rs60840586 | G | 1.1 | 1.07–1.14 | $9.7 \times 10^{-9}$ | | AQP3 | SCOURGE[20] |
| 9:133271182 | rs879055593 | T | 1.1 | 1.1–1.16 | $1 \times 10^{-16}$ | | ABO | SCGG[19] |
| 10:79946568 | rs721917 | A | 0.93 | 0.9–0.95 | $7.6 \times 10^{-9}$ | | SFTPD | HGI[4] |
| 11:1219991 | rs35705950 | T | 0.86 | 0.82–0.89 | $3.8 \times 10^{-14}$ | | MUC5B | HGI[4] |
| 11:34482745 | rs61882275 | A | 0.88 | 0.86–0.91 | $1 \times 10^{-22}$ | | ELF5 | GenOMICC[2] |
| 12:112919637 | rs2660 | A | 1.1 | 1.08–1.13 | $2.8 \times 10^{-15}$ | | OAS1 | GenOMICC[1] |
| 12:132481571 | rs11614702 | A | 1.1 | 1.08–1.13 | $2.1 \times 10^{-16}$ | | FBRSL1 | GenOMICC[2] |
| 13:112881427 | rs12585036 | T | 1.2 | 1.12–1.18 | $9.6 \times 10^{-22}$ | | ATP11A | GenOMICC[2] |
| 16:89196249 | rs117169628 | A | 1.2 | 1.12–1.2 | $2.6 \times 10^{-16}$ | | SLC22A31 | GenOMICC[2] |
| 17:40003082 | rs12941811 | T | 0.93 | 0.91–0.95 | $1.1 \times 10^{-9}$ | | PSMD3 | GenOMICC[new] |
| 17:46085231 | rs8080583 | A | 0.89 | 0.86–0.91 | $1.8 \times 10^{-16}$ | | KANSL1 | 21 |
| 17:49863303 | rs77534576 | T | 1.3 | 1.24–1.43 | $8.7 \times 10^{-16}$ | | TAC4 | 21 |
| 19:4717660 | rs12610495 | A | 0.8 | 0.77–0.82 | $9.1 \times 10^{-51}$ | | DPP9 | GenOMICC[1] |
| 19:10352442 | rs34536443 | C | 1.5 | 1.39–1.61 | $2.2 \times 10^{-28}$ | | TYK2 | GenOMICC[1] |
| 19:10414696 | rs142770866 | A | 1.2 | 1.19–1.3 | $9.4 \times 10^{-21}$ | | PDE4A | GenOMICC[new] |
| 19:48702915 | rs516246 | T | 0.9 | 0.88–0.93 | $1.4 \times 10^{-15}$ | | FUT2 | GenOMICC[2] |
| 19:50374423 | rs35463555 | A | 1.1 | 1.07–1.13 | $1.9 \times 10^{-13}$ | | NR1H2 | HGI[4] |
| 20:6489447 | rs2326788 | A | 0.93 | 0.9–0.95 | $1.5 \times 10^{-8}$ | | CASC20 | GenOMICC[new] |
| 21:33229937 | rs188401375 | C | 0.74 | 0.66–0.84 | $3.1 \times 10^{-6}$ | $4.7 \times 10^{-9}$ | IFNAR2 | GenOMICC[1] |
| 21:33237639 | rs9636867 | A | 0.83 | 0.81–0.85 | $5.2 \times 10^{-48}$ | | IFNAR2 | GenOMICC[1] |
| 21:33287378 | rs8178521 | T | 1.1 | 1.1–1.17 | $6.2 \times 10^{-15}$ | | IL10RB | GenOMICC[2] |
| 21:33980963 | rs76608815 | T | 1.2 | 1.14–1.23 | $7.4 \times 10^{-17}$ | | ATP5PO | GenOMICC[2] |
| 21:41479527 | rs915823 | A | 1.1 | 1.06–1.13 | $2.1 \times 10^{-9}$ | | TMPRSS2 | GenOMICC[new] |
| X:15523993 | rs35697037 | A | 1 | 1.03–1.06 | $6.8 \times 10^{-9}$ | | ACE2 | HGI[4] |

Chr:pos(b38), chromosome and position on human genome build 38; rsid, lead variant rsid; EA, effect allele; OR, odds ratio; OR$_{CI}$, odds ratio confidence interval; P, P value against null hypothesis of odds ratio of one; P$_{cond}$, P value in conditional analysis in variants with $P > 5 \times 10^{-8}$; nearest gene, the nearest or most plausible nearby gene; citation, the first report of genome-wide significant association. 'GenOMICC[new]' indicates findings presented in this Article. Additional details are provided in Supplementary Table 15 and full results of conditional analysis are provided in Supplementary Table 16. Supplementary Table 17 contains details of lead variants from the analysis of the hospitalized phenotype.

the Supplementary Information. We used a mathematical subtraction approach, as done in our previous work[2], to remove signals of previous GenOMICC releases from HGIv6, yielding an independent dataset.

As no replication cohorts exist for these meta-analyses, we used the heterogeneity across studies to assess the reliability of individual findings (Supplementary Table 15). Owing to the unusually extreme phenotype in the GenOMICC study, some heterogeneity is expected for the strongest associations when compared with studies with more permissive inclusion criteria. Importantly, significant heterogeneity was not detected for any of the findings that we report here (Supplementary Table 15). Comparing effect estimates between studies using a regression approach that takes into account estimation errors (Methods), we detected systematic differences in effect sizes between studies (Extended Data Fig. 3). For example, effects for the HGI critical illness phenotype (which was designed to parallel the GenOMICC inclusion criteria) are smaller than those obtained using prospective recruitment in GenOMICC by a factor of 0.68. As the effect sizes in GenOMICC are consistently larger than other studies, and GenOMICC contributes a disproportionately large signal to meta-analyses of both critical and hospitalized phenotypes (Extended Data Fig. 4), between-study heterogeneity is likely to reflect the careful case ascertainment and extreme phenotype in GenOMICC compared with other studies.

We found 49 common genetic associations with critical COVID-19 meeting our criteria for genome-wide significance in the absence of heterogeneity (Extended Data Fig. 2 and Table 1). Findings from previous reports were consistently replicated (Extended Data Table 2). Conditional analysis revealed two additional lead variants (Table 1) and statistical fine-mapping provided credible sets of putative causal variants for a majority of lead variants (Supplementary Figs. 27–44 and Supplementary Table 5). Gene-level analyses found 196 significantly associated genes at a Bonferroni-corrected threshold (Supplementary Table 10). There were no genome-wide significant differences in the effects between sexes in a sex-stratified meta-analysis using a subset of cohorts (Supplementary Fig. 1).

## Therapeutic implications

Our analysis is limited to common variants that are detectable on genotyping arrays and imputation panels. Although most lead variants are not directly causal, in some cases, they highlight molecular mechanisms that alter clinical outcomes in COVID-19, and may have direct therapeutic relevance. To investigate the disease mechanisms, we first quantified the effect of inferred gene expression on critical illness in three relevant tissue/cell types. Many of the genes that we have found to be implicated in critical COVID-19 (refs. 1,2) are highly expressed in the monocyte–macrophage system, which has poor coverage in existing expression quantitative trait loci (eQTL) datasets. For this reason, we constructed a new TWAS model in primary monocytes obtained from 176 individuals (Methods). We found significant associations after Bonferroni correction between critical COVID-19 and predicted gene expression in lung (33), blood (21), monocyte (37) and all-tissue (107) meta-analysis (Supplementary Table 2 and Supplementary Table 11). We extended these findings using generalized summary-level data Mendelian randomization (GSMR) for RNA expression (Fig. 2, Extended Data Table 1, Supplementary Figs. 11–18 and Supplementary Table 4).

In parallel, we assessed the effect of genetically determined variation in circulating protein levels on the critical illness phenotype using GSMR[5]. We identified 15 unique proteins linked to critical illness, as summarized in Extended Data Table 1 (Supplementary Table 3). Of the significant results, we found causal evidence implicating five new proteins in comparison to our previous GSMR analysis[2]: QSOX2, CREB3L4, myeloperoxidase (MPO), ADAMTS13 and mannose-binding lectin-2 (MBL2) (Supplementary Fig. 10). These include well-studied biomarkers and potential drug targets in sepsis—the innate immune pattern recognition receptor MBL2 and the neutrophil effector enzyme MPO. ADAMTS13 modulates von Willebrand-factor-mediated platelet thrombus formation and may have a role in the hypercoagulable state in critical COVID-19 (Extended Data Fig. 5).

Three genes containing non-synonymous protein-coding changes associated with severe disease were also found to have significant effects from differential gene expression: *SLC22A31* (ref. 2) (Fig. 1), *SFTPD*[4] (Fig. 1) and *TKY2* (ref. 1) (Extended Data Fig. 6). Further biological and clinical research will be required to dissect the genetic evidence at these loci. In the example of *TYK2*, there is now a therapeutic test of the genetic predictions. Our previous report of association between higher expression and critical illness[1] led directly to the inclusion of a new drug, baricitinib, in a large clinical trial; the result demonstrated a clear therapeutic benefit[3]. This therapeutic signal is consistent across multiple trials, providing the first proof-of-concept for drug target identification using genetics in critical illness and infectious disease.

To assess the immediate therapeutic use of our results for repurposing of existing compounds, we considered the drug therapies under consideration by the UK COVID-19 Therapeutic Advisory Panel (UK-CTAP), a national independent review group supported by an expert due-diligence panel[6]. Consistent evidence from gene-level GWAS (Supplementary Table 6 and Supplementary Table 10) and post-GWAS analyses was identified for several licensed compounds (Supplementary Table 12). For example, we found an association in another gene encoding a protein that is inhibited by baricitinib and other JAK inhibitors—the intracellular signalling kinase, *JAK1*, which is stimulated by numerous cytokines including type I interferons and IL-6. Mendelian randomization analysis of RNA expression revealed a significant positive association between the expression of the gene encoding a canonical inflammatory cytokine, tumour necrosis factor (*TNF*), and severe disease (Fig. 2). This suggests that inhibition of TNF signalling may be an effective therapy in severe COVID-19.

Our additional expression data in monocytes reveal a marked tissue-specific effect on expression of *PDE4A*. This phosphodiesterase regulates the production of multiple inflammatory cytokines by myeloid cells. In contrast to the negative correlations seen in the lungs and blood, we show that a genetic tendency for higher expression of *PDE4A* in monocytes is associated with critical COVID-19 (Supplementary Table 11). Inhibition of PDE4A by several existing drugs is under investigation in multiple inflammatory diseases[7], reduces pulmonary endothelial permeability[8] and appears to be safe in small clinical trials in patients with COVID-19.

The postulated biological role of genes associated with critical COVID-19 in GWAS, TWAS and GSMR results is shown in Extended Data Fig. 5, which highlights the preponderance of genes with expression or functions in the mononuclear phagocyte system. This includes *SLC2A5*, encoding the GLUT5 fructose transporter, which is strongly inducible in primary macrophages in response to inflammatory stimulation[9], and *XCR1*, a dendritic cell receptor with a critical role in cytotoxic T cell-mediated antiviral immunity[10]. *NPNT*, a significant meta-TWAS association in the genome-wide significant region on chromosome 4 (chr4:105673359; Supplementary Table 11), encodes a pulmonary basement membrane protein that may have a protective role in acute lung injury[11].

## Host–pathogen interaction

Our results also demonstrate the capacity of host genetics to reveal core mechanisms of disease. Multiple genes implicated in viral entry are associated with severe disease. In addition to *ACE2*, we detect a genome-wide significant association in *TMPRSS2*, a key host protease that facilitates viral entry that we have previously studied as a candidate gene[12]. This effect may be viral-lineage specific[13]. A strong GWAS association is seen in *RAB2A* (Table 1), with TWAS evidence suggesting that more expression of this gene is associated with worse disease (Supplementary Table 11). *RAB2A* is highly ranked in our previous

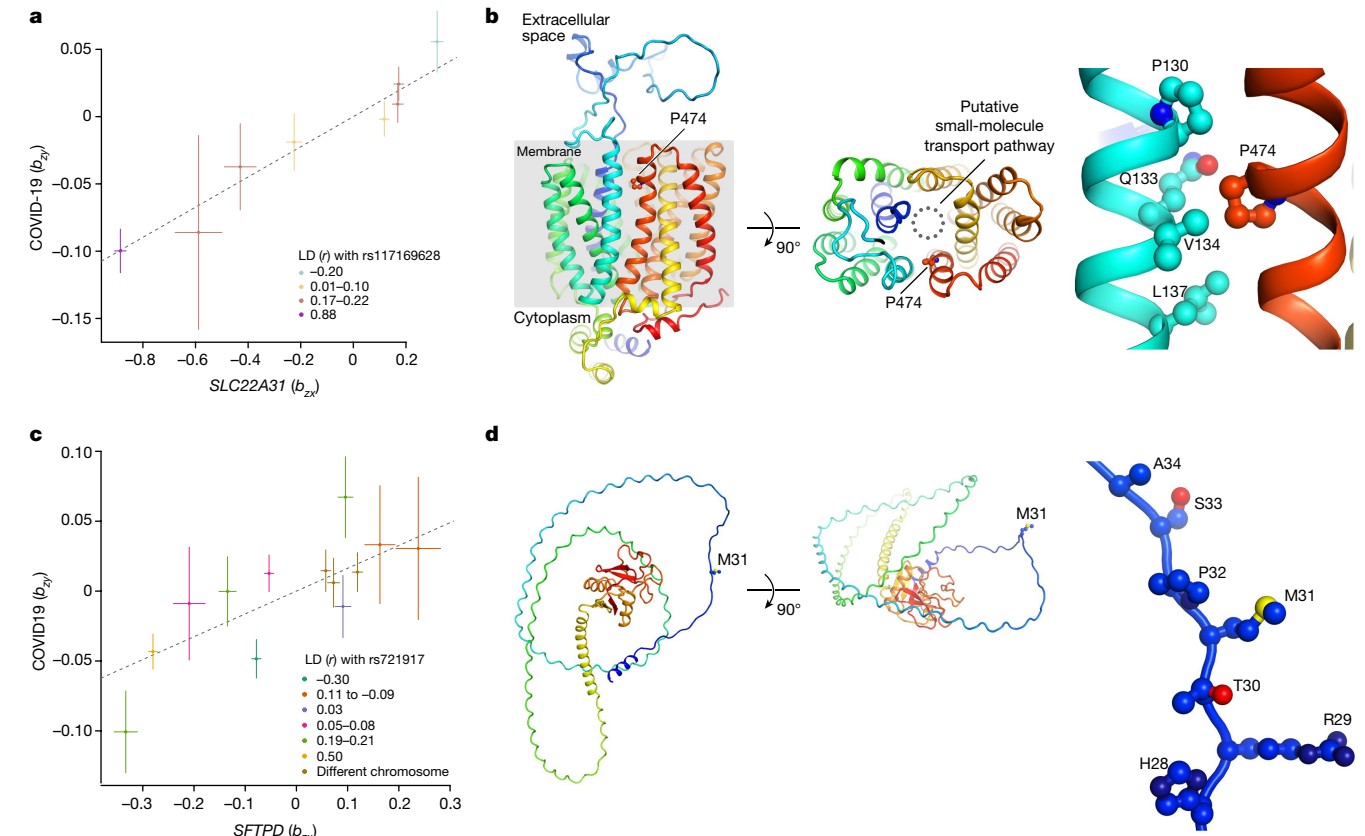

**Fig. 1 | Functional genomics analyses for *SLC22A31* and *SFTPD*. a**, Effect-size plot for the effect of multiple variants on *SLC22A31* expression (eQTLgen, *x* axis) against increasing susceptibility to critical COVID-19 ($\beta_{xy} = 0.11$; $P_{xy} = 1.3 \times 10^{-9}$). The colour shows linkage disequilibrium (LD) with the missense variant rs117169628. **b**, Three cartoon views of an AlphaFold[22] model of putative solute carrier family 22 member 31 (SLC22A31; UniProtKB: A6NKX4). The side chains of Pro474 and interacting amino acids are shown as connected spheres. A putative channel for small-molecule transport across the cell membrane is indicated by a dashed circle. Pro474 is predicted to be located in the transmembrane helix and point towards a putative transport pathway of a small molecule. The risk variant, P474L (Ala at rs117169628) would be expected to introduce more flexibility to the transmembrane helix and might therefore affect the transport properties of SLC22A31. Pro474 is predicted to be in a tightly packed environment, and may therefore affect the folding of SLC22A31. **c**, Effect-size plot for effect of multiple variants on SFTPD expression (eQTLgen, *x* axis) against increasing susceptibility to critical COVID-19 ($\beta_{xy} = 0.16$; $P_{xy} = 9.7 \times 10^{-6}$). Colour shows linkage disequilibrium with the missense variant rs721917. **d**, Three cartoon views of an AlphaFold[22] model of pulmonary surfactant-associated protein D (SFTPD; UniProtKB: P35247). The side chain of the variant Met31 is shown as connected spheres. Met31 is predicted to be located in the secondary-structure-lacking region of SFTPD. In the diagram on the right, oxygen and nitrogen atoms are coloured red and blue respectively, and the sulfur atom is coloured yellow.

meta-analysis by information content[14] study of host genes implicated in SARS-CoV-2 interaction using in vitro and clinical data[15], and is consistent with CRISPR screen data showing that *RAB2A* is required for viral replication[16].

Although our focus on critical illness enhances discovery power (Extended Data Fig. 4), it has the disadvantage of combining genetic signals for multiple stages in disease progression, including viral exposure, infection and replication, and development of inflammatory lung

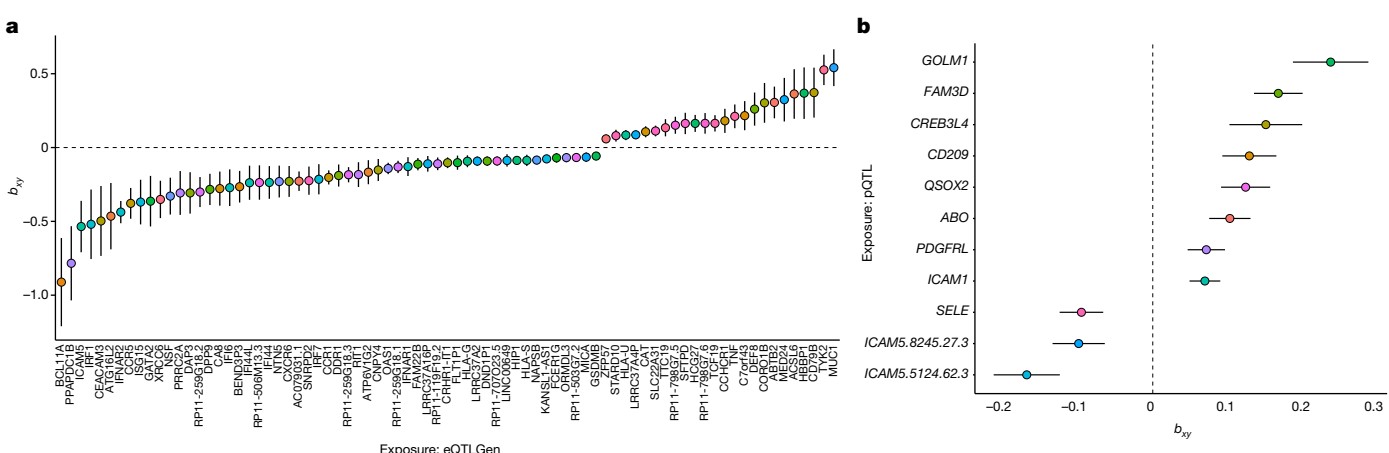

**Fig. 2 | GSMR effect sizes. a,b**, The predicted effect of change in protein concentration (**a**) and gene expression (**b**) on the risk of critical COVID-19 is shown for proteins and genes significantly linked to critical COVID-19 by GSMR (false-discovery rate (FDR) < 0.01). The bars show 95% confidence intervals.

disease. From these data alone we cannot identify when in disease progression the causal effect is mediated, although clinical evidence helps to make some predictions[17] (Extended Data Fig. 5). As most cases included were recruited before vaccinations and treatments became available (Extended Data Fig. 7), at present, our study does not have sufficient statistical power to dissect the genetic effects of treatments or vaccination. These effects may include the masking of true associations, or the detection of genetic effects mediated by vaccine or drug response, rather than COVID-19 susceptibility. However, the absence of divergent genetic effects between studies (Supplementary Figs. 2–5) or consistent changes in effect allele frequency among cases over time (Supplementary Figs. 45–48) suggests that treatment and vaccination have not substantially affected the association between the specific variants that we report and the risk of critical illness.

As we performed a meta-analysis of multiple studies that may have slightly different definitions of the phenotype, effect sizes differ between studies (Supplementary Figs. 2–5). This, together with ancestry-specific effects[1], may explain the heterogeneity in strong GWAS signals, such as the LZTFL1 signal in Table 1. Different studies also have sets of variants that are not completely overlapping, so $P$ values between variants in high linkage disequilibrium are more different than expected. Although most of the studies contain individuals from multiple ancestries, a large majority of the individuals are of European ancestry. In future research, there is a scientific and moral imperative to include the full diversity of human populations.

Together, these results deepen our understanding of the pathogenesis of critical COVID-19 and highlight new biological mechanisms of disease, several of which have immediate potential for therapeutic targeting.

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

¹Baillie Gifford Pandemic Science Hub, Centre for Inflammation Research, The Queen's Medical Research Institute, University of Edinburgh, Edinburgh, UK. ²MRC Human Genetics Unit, Institute of Genetics and Cancer, University of Edinburgh, Western General Hospital, Edinburgh, UK. ³Roslin Institute, University of Edinburgh, Edinburgh, UK. ⁴Pain Service, NHS Tayside, Ninewells Hospital and Medical School, Dundee, UK. ⁵School of Life Sciences, Westlake University, Hangzhou, China. ⁶Westlake Laboratory of Life Sciences and Biomedicine, Hangzhou, China. ⁷Institute for Molecular Bioscience, The University of Queensland, Brisbane, Queensland, Australia. ⁸Wellcome Centre for Human Genetics, University of Oxford, Oxford, UK. ⁹Faculty of Biological Sciences, University of Leeds, Leeds, UK. ¹⁰Genomics England, London, UK. ¹¹Intensive Care Unit, Royal Infirmary of Edinburgh, Edinburgh, UK. ¹²National Heart and Lung Institute, Imperial College London, London, UK. ¹³Edinburgh Clinical Research Facility, Western General Hospital, University of Edinburgh, Edinburgh, UK. ¹⁴Department of Critical Care Medicine, Queen's University and Kingston Health Sciences Centre, Kingston, Ontario, Canada. ¹⁵Clinical Research Centre at St Vincent's University Hospital, University College Dublin, Dublin, Ireland. ¹⁶NIHR Health Protection Research Unit for Emerging and Zoonotic Infections, Institute of Infection, Veterinary and Ecological Sciences University of Liverpool, Liverpool, UK. ¹⁷Respiratory Medicine, Alder Hey Children's Hospital, Institute in The Park, University of Liverpool, Alder Hey Children's Hospital, Liverpool, UK. ¹⁸Centre for Inflammation Research, The Queen's Medical Research Institute, University of Edinburgh, Edinburgh, UK. ¹⁹Department of Medicine, University of Cambridge, Cambridge, UK. ²⁰William Harvey Research Institute Barts and the London School of Medicine and Dentistry, Queen Mary University of London, London, UK. ²¹Centre for Tropical Medicine and Global Health, Nuffield Department of Medicine, University of Oxford, Oxford, UK. ²²Department of Anaesthesia and Intensive Care, The Chinese University of Hong Kong, Prince of Wales Hospital, Hong Kong, China. ²³Wellcome-Wolfson Institute for Experimental Medicine, Queen's University Belfast, Belfast, UK. ²⁴Department of Intensive Care Medicine, Royal Victoria Hospital, Belfast, UK. ²⁵UCL Centre for Human Health and Performance, London, UK. ²⁶Imperial College Healthcare NHS Trust, London, UK. ²⁷Royal Hospital for Children, Glasgow, UK. ²⁸Centre for Global Health Research, Usher Institute of Population Health Sciences and Informatics, Edinburgh, UK. ²⁹Genomics Division, Instituto Tecnológico y de Energías Renovables, Santa Cruz de Tenerife, Spain. ³⁰Research Unit, Hospital Universitario N.S. de Candelaria, Santa Cruz de Tenerife, Spain. ³¹Centre for Biomedical Network Research on Respiratory Diseases (CIBERES), Instituto de Salud Carlos III, Madrid, Spain. ³²Department of Clinical Sciences, University Fernando Pessoa Canarias, Las Palmas de Gran Canaria, Spain. ³³IDIVAL, Santander, Spain. ³⁴Universidad de Cantabria, Santander, Spain. ³⁵Hospital U M Valdecilla, Santander, Spain. ³⁶Tecnologico de Monterrey, Escuela de Medicina y Ciencias de la Salud y Hospital San Jose TecSalud, Monterrey, Mexico. ³⁷Centre for Biomedical Network Research on Rare Diseases (CIBERER), Instituto de Salud Carlos III, Madrid, Spain. ³⁸Instituto de Genética Médica y Molecular (INGEMM), Hospital Universitario La Paz-IDIPAZ, Madrid, Spain. ³⁹ERN-ITHACA-European Reference Network, Paris, France. ⁴⁰Genomic Research Department, King Fahad Medical City, Riyadh, Saudi Arabia. ⁴¹Department of Pathology & Laboratory Medicine, Emory University Hospital, Atlanta, GA, USA. ⁴²Department of Clinical Analysis and Toxicology, Federal University of Rio Grande do Norte, Natal, Brazil. ⁴³Department of Anthropology, University of Toronto at Mississauga, Mississauga, Ontario, Canada. ⁴⁴Centro Singular de Investigación en Medicina Molecular y Enfermedades Crónicas (CIMUS), Universidade de Santiago de Compostela, Santiago de Compostela, Spain. ⁴⁵Instituto de Investigación Sanitaria de Santiago (IDIS), Santiago de Compostela, Spain. ⁴⁶Fundación Pública Galega de Medicina Xenómica, Sistema Galego de Saúde (SERGAS) Santiago de Compostela, Santiago de Compostela, Spain. ⁴⁷Intensive Care National Audit & Research Centre, London, UK. ⁴⁸Heart Institute, University of Sao Paulo, Butanta, Brazil. ⁵⁶⁴These authors contributed equally: Erola Pairo-Castineira, Konrad Rawlik. *A list of authors and their affiliations appears online. ✉e-mail: j.k.baillie@ed.ac.uk

## GenOMICC Investigators

### Co-Investigators

J. Kenneth Baillie[1,2,3,11], Colin Begg[27], Sara Clohisey[3], Charles Hinds[20], Peter Horby[21], Julian Knight[8], Lowell Ling[22], David Maslove[14], Danny McAuley[23,24], Johnny Millar[3], Hugh Montgomery[25], Alistair Nichol[15], Peter J. M. Openshaw[12,26], Alexandre C. Pereira[48], Chris P. Ponting[2], Kathy Rowan[47], Malcolm G. Semple[16,17], Manu Shankar-Hari[18], Charlotte Summers[19] & Timothy Walsh[11]

### Management, Laboratory and Data Team

Emma Aitkin[3], Latha Aravindan[49], Ruth Armstrong[3], J. Kenneth Baillie[3,11], Heather Biggs[50], Ceilia Boz[3], Adam Brown[3], Primmy Chikowore[3], Richard Clark[13], Sara Clohisey[3], Audrey Coutts[13], Judy Coyle[3], Louise Cullum[3], Sukamal Das[49], Nicky Day[3], Lorna Donnelly[13], Esther Duncan[3], Angie Fawkes[13], Paul Finernan[3], Max Head Fourman[3], Anita Furlong[50], James Furniss[3], Bernadette Gallagher[3], Tammy Gilchrist[13], Ailsa Golightly[3], Fiona Griffiths[3], Katarzyna Hafezi[13], Debbie Hamilton[3], Ross Hendry[3], Naomi Kearns[3], Andy Law[3], Dawn Law[3], Rachel Law[3], Sarah Law[3], Rebecca Lidstone-Scott[3], Christen Lauder[13], Louise Macgillivray[13], Alan Maclean[13], Hanning Mal[3], Sarah McCafferty[13], Ellie McMaster[3], Jen Meikle[3], Shona C. Moore[16], Kirstie Morrice[13], Lee Murphy[13], Sheena Murphy[49], Hellen Mybaya[3], Miranda Odam[3], Wilna Oosthuyzen[3], Chenqing Zheng[51], Jiantao Chen[51], Nick Parkinson[3], Trevor Paterson[3], Petra Tucker[50], Katherine Schon[50], Andrew Stenhouse[3], Mihaela Das[49], Maaike Swets[3,52], Helen Szoor-McElhinney[3], Filip Taneski[3], Lance Turtle[16], Tony Wackett[3], Mairi Ward[3], Jane Weaver[3], Nicola Wrobel[13] & Marie Zechner[3]

### Guys and St Thomas' Hospital, London, UK

Jacqueline Pan[53], Neus Grau[53], Tim Owen Jones[53], Rosario Lim[53], Martina Marotti[53], Christopher Whitton[53], Aneta Bociek[53], Sara Campos[53], Gill Arbane[53], Manu Shankar-Hari[53], Marlies Ostermann[53], Mina Cha[53], Fabiola DAmato[53], Eirini Kosifidou[53], Shelley Lorah[53] & Kyma Morera[53]

### James Cook University Hospital, Middlesbrough, UK

Sarah Bircham[54], Laura Brady[54], Keith Hugill[54], Jeremy Henning[54], Stephen Bonner[54], Evie Headlam[54], Jessica Jones[54], Abigail List[54], Joanne Morley[54], Amy Welford[54], Bobette Kamangu[54], Anitha Ratnakumar[54] & Abiola Shoremekun[54]

### Barts Health NHS Trust, London, UK

Zoe Alldis[55], Raine Astin-Chamberlain[55], Fatima Bibi[55], Jack Biddle[55], Sarah Blow[55], Matthew Bolton[55], Catherine Borra[55], Ruth Bowles[55], Maudrian Burton[55], Yasmin Choudhury[55], Amber Cox[55], Amy Easthope[55], Patrizia Ebano[55], Stavros Fotiadis[55], Jana Gurasashvili[55], Rosslyn Halls[55], Pippa Hartridge[55], Delordson Kallon[55], Jamila Kassam[55], Ivone Lancoma-Malcolm[55], Maninderpal Matharu[55], Peter May[55], Oliver Mitchelmore[55], Tabitha Newman[55], Mital Patel[55], Jane Pheby[55], Irene Pinzuti[55], Zoe Prime[55], Oleksandra Prysyazhna[55], Julian Shiel[55], Melanie Taylor[55], Carey Tierney[55], Olivier Zongo[55], Suzanne Wood[55], Anne Zak[55] & David Collier[55]

### Royal Stoke University Hospital, Staffordshire, UK

Manuela Mundy[56], Christopher Thompson[56], Lisa Pritchard[56], Minnie Gellamucho[56], David Cartlidge[56], Nageswar Bandla[56], Lucy Bailey[56], Michelle Davies[56], Jane Delaney[56] & Leanne Scott[56]

### North Middlesex University Hospital NHS trust, London, UK

Marwa Abdelrazik[57], Frater Alasdair[57], David Carter[57], Munzir Elhassan[57], Arunkumar Ganesan[57], Samuel Jenkins[57], Zoe Lamond[57], Dharam Purohit[57], Kumar Rohit[57], Malik Saleem[57], Alanna Wall[57], Kugan Xavier[57], Dhanalaksmi Bakthavatsalam[57], Kirolos Gehad[57], Pakeerathan Gnanapragasam[57], Kapil Jain[57], Swati Jain[57], Abdul Malik[57], Naveen Pappachan[57], Jeronimo Moreno-Cuesta[57], Anne Haldeos[57], Rachel Vincent[57] & Maryjane Oziegb[57]

### King's College Hospital, London, UK

Anna Cavazza[58], Maeve Cockrell[58], Eleanor Corcoran[58], Maria Depante[58], Clare Finney[58], Ellen Jerome[58], Abigail Knighton[58], Monalisa Nayak[58], Evita Pappa[58], Rohit Saha[58], Sian Saha[58], Andrew Dodd[58], Kevin O'Reilly[58], Mark McPhail[58], Emma Clarey[58], Harriet Noble[58] & John Smith[58]

### Charing Cross Hospital, St Mary's Hospital and Hammersmith Hospital, London, UK

Phoebe Coghlan[59], Stephen Brett[59], Anthony Gordon[59], Maie Templeton[59], David Antcliffe[59], Dorota Banach[59], Sarah Darnell[59], Ziortza Fernandez[59], Eleanor Jepson[59], Amal Mohammed[59], Roceld Rojo[59], Sonia Sousa Arias[59], Anita Tamang Gurung[59] & Jenny Wong[59]

### The Royal Liverpool University Hospital, Liverpool, UK

Jaime Fernandez-Roman[60], David O. Hamilton[60], Emily Johnson[60], Brian Johnston[60], Maria Lopez Martinez[60], Suleman Mulla[60], Alicia A. C. Waite[60], Karen Williams[60], Victoria Waugh[60], Ingeborg Welters[60], Jessica Emblem[60], Maria Norris[60] & David Shaw[60]

### John Radcliffe Hospital, Oxford, UK

Archana Bashyal[61], Sally Beer[61], Paula Hutton[61], Stuart McKechnie[61], Neil Davidson[61], Soya Mathew[61], Grace Readion[61], Jung Ryu[61] & Jean Wilson[61]

### Addenbrooke's Hospital, Cambridge, UK

Shruti Agrawal[62], Kay Elston[62], Megan Jones[62], Eoghan Meaney[62], Petra Polgarova[62], Muhammad Elbehery[62], Charlotte Summers[62], Esther Daubney[62], Anthony Ng[62], Jocelyn Marshall[62], Nazima Pathan[62], Katerina Stroud[62] & Deborah White[62]

### Nottingham University Hospital, Nottingham, UK

Angela Andrew[63], Saima Ashraf[63], Amy Clark[63], Martin Dent[63], Margaret Langley[63], Cecilia Peters[63], Lucy Ryan[63], Julia Sampson[63], Shuying Wei[63], Alice Baddeley[63], Megan Meredith[63], Lucy Morris[63], Alexandra Gibbons[63] & Lisa McLoughlin[63]

### St George's Hospital, London, UK

Carlos Castro Delgado[64], Victoria Clark[64], Deborah Dawson[64], Lijun Ding[64], Georgia Durrant[64], Obiageri Ezeobu[64], Abiola Harrison[64], William James Hurt[64], Rebecca Kanu[64], Ashley Kinch[64], Susannah Leaver[64], Ana Lisboa[64], Jisha Mathew[64], Kamal Patel[64], Romina Pepermans Saluzzio[64], John Rawlins[64], Tinashe Samakomva[64], Nirav Shah[64], Christine Sicat[64], Joana Texeira[64], Joana Gomes De Queiroz[64], Edna Fernandes Da Gloria[64], Elena Maccacari[64], Nikki Yun[64], Soumendu Manna[64], Sarah Farnell-Ward[64], Maria Maizcordoba[64], Maria Thanasi[64] & Hawakin Haji Ali[64]

### BHRUT (Barking Havering)—Queens Hospital and King George Hospital, Essex, UK

Janice Hastings[65], Lina Grauslyte[65], Musarat Hussain[65], Bobby Ruge[65], Sam King[65], Tatiana Pogreban[65], Lace Rosaroso[65], Helen Smith[65], Mandeep-Kaur Phull[65], Nikkita Adams[65], George Franke[65], Aparna George[65], Erika Salciute[65], Joanna Wong[65], Karen Dunne[65], Luke Flower[65], Emma Sharland[65] & Sukhmani Sra[65]

### Royal Infirmary of Edinburgh, Edinburgh, UK

Gillian Andrew[66], Marie Callaghan[66], Lucy Barclay[66], Lucy Marshall[66], Kenneth Baillie[66], Maria Amamio[66], Sophie Birch[66], Kate Briton[66], Sarah Clark[66], Katerine Doverman[66], Dave Hope[66], Corrienne Mcculloch[66], Scott Simpson[66] & Jo Singleton[66]

### Kingston Hospital, Surrey, UK

Rita Fernandez[67], Meryem Allen[67], David Baptista[67], Rebecca Crowe[67], Jonathan Fox[67], Jacyntha Khera[67], Adam Loveridge[67], India McKenley[67], Eriko Morino[67], Andres Naranjo[67], Denise O'Connor[67], Richard Simms[67], Kathryn Sollesta[67], Andrew Swain[67], Harish Venkatesh[67], Rosie Herdman-Grant[67] & Anna Joseph[67]

### Queen Alexandra Hospital, Portsmouth, UK

Angela Nown[68], Steve Rose[68], David Pogson[68], Helen Boxall[68], Lutece Brimfield[68], Helen Claridge[68], Zoe Daly[68], Shenu George[68] & Andrew Gribbin[68]

### Royal Gwent Hospital, Newport, UK

Yusuf Cheema[69], Sean Cutler[69], Owen Richards[69], Anna Roynon-Reed[69], Shiney Cherian[69], Anne Emma Heron[69], Gemma Williams[69], Tamas Szakmany[69], Abby Waters[69], Kim Collins[69], Jill Dunhill[69], Ffion Jones[69], Rebecca Morris[69], Lucy Ship[69] & Amy Cardwell[69]

### Royal Blackburn Teaching Hospital, Blackburn, UK

Syamlan Ali[70], Ravi Bhatterjee[70], Rachel Bolton[70], Srikanth Chukkambotla[70], Dabheoc Coleman[70], Jack Dalziel[70], Joseph Dykes[70], Christopher Fine[70], Bethan Gay[70], Wendy Goddard[70], Drew Goodchild[70], Rhiannan Harling[70], Muhammad Hijazi[70], Sarah Keith[70], Meherunnisa Khan[70], Roseanna Matt[70], Janet Ryan-Smith[70], Samuel Saad[70], Philippa Springle[70], Jacqueline Thomas[70], Nick Truman[70], Aayesha Kazi[70], Matthew Smith[70], Heather Collier[70], Chloe Davison[70], Stephen Duberley[70], Jeanette Hargreaves[70], Janice Hartley[70], Tahera Patel[70] & Ellen Smith[70]

### Stepping Hill Hospital, Stockport, UK

Alissa Kent[71], Emma Goodwin[71], Ahmed Zaki[71], Clare Tibke[71], Susan Hopkins[71], Hywel Gerrard[71], Matthew Jackson[71], Sara Bennett[71], Liane Marsh[71] & Rebecca Mills[71]

### Northumbria Healthcare NHS Foundation Trust, North Shields, UK

Jessica Bell[72], Helen Campbell[72], Angela Dawson[72], Steve Dodds[72], Stacey Duffy[72], Lisa Gallagher[72], Gemma McCafferty[72], Stacey Short[72], Tracy Smith[72], Kirsty Thomas[72], Claire Walker[72], Jessica Reynolds[72], Bryan Yates[72], Hayley McKie[72], Maria Panteli[72], Maria Thompson[72] & Gail Waddell[72]

### Countess of Chester Hospital, Chester, UK

Sarah De Beger[73], Azmerelda Abraheem[73], Charlie Dunmore[73], Rumanah Girach[73], Rhianna Jones[73], Emily London[73], Imrun Nagra[73], Farah Nasir[73], Hannah Sainsbury[73], Clare Smedley[73], Stephen Brearey[73], Caroline Burchett[73], Kathryn Cawley[73], Maria Faulkner[73], Helen Jeffrey[73], Peter Bamford[73], Firdaus Shaikh[73], Lauren Slack[73] & Angela Davies[73]

### Pinderfields General Hospital, Wakefield, UK

Hollie Brooke[74], Jose Cebrian Suarez[74], Ruth Charlesworth[74], Karen Hansson[74], John Norris[74], Alice Poole[74], Rajdeep Sandhu[74], Elizabeth Smithson[74], Muthu Thirumaran[74], Veronica Wagstaff[74], Sarah Buckley[74], Brendan Sloan[74], Alastair Rose[74], Amy Major[74] & Alexandra Metcalfe[74]

### Ninewells Hospital, Dundee, UK

Christine Almaden-Boyle[75], Pauline Austin[75], Susan Chapman[75], Alexandre Eros[75], Louise Cabrelli[75], Stephen Cole[75], Clare Whyte[75] & Matt Casey[75]

### Croydon University Hospital, Croydon, UK

Vasileios Bafitis[76], George Tsinaslanidis[76], Cassandra George[76], Reena Khade[76], Christopher Black[76] & Sundar Raj Ashok[76]

### Morriston Hospital, Swansea, UK

Sean Farley[77], Elaine Brinkworth[77], Rachel Harford[77], Carl Murphy[77], Marie Williams[77], Luke Newey[77], Hannah Toghill[77], Sophie Lewis[77], Tabitha Rees[77], Ceri Battle[77], Mark Baker[77], Jenny Travers[77] & Karen Chesters[77]

**Queen Elizabeth University Hospital, Glasgow, UK**
Nicola Baxter[78], Andrew Arnott[78], Gordan McCreath[78], Christopher McParland[78], Laura Rooney[78], Malcolm Sim[78], Steven Henderson[78], Lynn Abel[78], Carol Dalton[78], Sophie Kennedy-Hay[78], Lynn O'Donohoe[78], Megan O'Hare[78], Izabela Orlikowska[78] & Natasha Parker[78]

**Broomfield Hospital, Chelmsford, UK**
Fiona McNeela[79], Amanda Lyle[79], Alistair Hughes[79], Jayachandran Radhakrishnan[79] & Sian Gibson[79]

**Heartlands Hospital, Birmingham, UK**
Hollie Bancroft[80], Mary Bellamy[80], Jacqueline Daglish[80], Salma Kadiri[80], Faye Moore[80], Joanne Rhodes[80], Mirriam Sangombe[80], Zhane Peterkin[80], James Scriven[80] & Margaret Carmody[80]

**Royal Sussex County Hospital, Brighton, UK**
Juliet Cottle[81], Emily Peasgood[81], Laura Ortiz-Ruiz de Gordoa[81], Claire Phillips[81] & Denise Skinner[81]

**York Hospital, York, UK**
Zoe Cinquina[82], Kate Howard[82], Rosie Joy[82], Samantha Roche[82], Isobel Birkinshaw[82], Joseph Carter[82], Jo Ingham[82], Nicola Marshall[82], Harriet Pearson[82] & Zoe Scott[82]

**Queen Elizabeth Hospital, Birmingham, UK**
Jo Dasgin[83], Jaspret Gill[83], Annette Nilsson[83], Amy Bamford[83], Diana Hull[83], James Scriven[83], Nafeesah Ahmadhaider[83], Michelle Bates[83] & Christopher McGhee[83]

**Royal Glamorgan Hospital, Pontyclun, UK**
Hannah Ellis[84], Gwenllian Sera Howe[84], Jayaprakash Singh[84], Natalie Stroud[84], Lisa Roche[84], Ceri Lynch[84], Bethan Deacon[84], Carla Pothecary[84], Justyna Smeaton[84] & Kevin Agravante[84]

**Barnet Hospital, London, UK**
Vinodh Krishnamurthy[85], Cynthia Diaba[85], Lincy John[85], Lai Lim[85] & Rajeev Jha[85]

**Wythenshawe Hospital, Manchester, UK**
Jasmine Egan[86], Timothy Felton[86], Susannah Glasgow[86], Grace Padden[86], Ozerah Choudhr[86], Joanne Bradley-Potts[86], Stuart Moss[86], Saejohn Lingeswaran[86], Peter Alexander[86], Craig Brandwood[86], Sofia Fiouni[86], Luke Ward[86], Schvearn Allen[86], Jane Shaw[86] & Christopher Smith[86]

**Medway Maritime Hospital, Gillingham, UK**
Oluronke Adanini[87], Rebecca Collins[87], Maines Msiska[87], Linda Ofori[87], Nikhil Bhatia[87] & Hayley Dolan[87]

**Royal Berkshire NHS Foundation Trust, Berkshire, UK**
Mark Brunton[88], Jess Caterson[88], Holly Coles[88], Liza Keating[88], Emma Tilney[88], Nicola Jacques[88], Matthew Frise[88], Jennifer Armistead[88], Shauna Bartley[88], Parminder Bhuie[88], Sabi Rai[88] & Gabriela Tomkova[88]

**Whiston Hospital, Prescot, UK**
Sandra Greer[89], Karen Shuker[89] & Ascanio Tridente[89]

**The Royal Oldham Hospital, Manchester, UK**
Emma Dobson[90], Jodie Hunt[90], Redmond Tully[90], Joy Dearden[90], Andrew Drummond[90], Prakash Kamath[90], Emily Bullock[90], Michelle Mulcahy[90], Shelia Munt[90], Grainne O'Connor[90], Jennifer Philbin[90], Chloe Rishton[90], Chloe Scott[90] & Sarah Winnard[90]

**Chesterfield Royal Hospital Foundation Trust, Chesterfield, UK**
Nurkamalia Hasni[91], Rachel Gascoyne[91], Joanne Hawes[91], Kelly Pritchard[91], Lesley Stevenson[91], Amanda Whileman[91], Sarah Beavis[91], Lauren Bishop[91], Cindy Cart[91], Katie Dale[91], Mary Kelly-Baxter[91], Adam Mendelski[91], Emma Moakes[91], Rheanna Smith[91], Jan Woodward[91] & Stephanie Wright[91]

**Aberdeen Royal Infirmary, Aberdeen, UK**
Angela Allan[92], Adriana Botello[92], Jade Liew[92], Jasmine Medhora[92], Erin Trumper[92], Felicity Savage[92], Teresa Scott[92], Marc Place[92] & Callum Kaye[92]

**Royal Devon and Exeter Hospital, Exeter, UK**
Sarah Benyon[93], Suzie Marriott[93], Linda Park[93], Helen Quinn[93], Daisy Skyes[93], Lily Zitter[93], Kizzy Baines[93], Elizabeth Gordon[93], Samantha Keenan[93] & Andrew Pitt[93]

**Glasgow Royal Infirmary, Glasgow, UK**
Katharine Duffy[94], Jane Ireland[94], Gary Semple[94], Lynne Turner[94], Susanne Cathcart[94], Dominic Rimmer[94], Alex Puxty[94], Kathryn Puxty[94], Andrew Hurst[94], Jennifer Miller[94], Susan Speirs[94] & Lauren Walker[94]

**Blackpool Victoria Hospital, Blackpool, UK**
Zena Bradshaw[95], Joanna Brown[95], Sarah Melling[95], Stephen Preston[95], Nicola Slawson[95], Scott Warden[95], Alanna Beasley[95], Emma Stoddard[95], Leonie Benham[95], Jason Cupitt[95], Melanie Caswell[95], Lisa Elawamy[95] & Ashleigh Wignall[95]

**Southampton General Hospital, Southampton, UK**
Belinda Roberts[96], Hannah Golding[96], Samantha Leggett[96], Michelle Male[96], Martyna Marani[96], Kirsty Prager[96], Toran Williams[96], Kim Golder[96], Oliver Jones[96], Rebecca Cusack[96], Clare Bolger[96], Rachel Burnish[96], Michael Carter[96], Susan Jackson[96], Karen Salmon[96] & Jonathan Biss[96]

**Ashford and St Peter's Hospital, Surrey, UK**
Maia Aquino[97], Maria Croft[97], Victoria Frost[97], Ian White[97] & Keshnie Govender[97]

**Derriford Hospital, Plymouth, UK**
Natasha Webb[98], Liana Stapleton[98], Colin Wells[98], Nikitas Nikitas[98], Ana Sanchez-Rodriguez[98], Kayleigh Spencer[98] & Bethan Stowe[98]

**East Surrey Hospital, Redhill, UK**
Yvonne Izzard[99], Michelle Poole[99], Sonja Monnery[99], Sallyanne Trotman[99], Valerie Beech[99], Edward Combes[99] & Teishel Joefield[99]

**Poole Hospital, Poole, UK**
Patrick Covernton[100], Sarah Savage[100], Elizabeth Woodward[100], Julie Camsooksai[100], Henrik Reschreiter[100], Charlotte Barclay[100], Yasmin DeAth[100], Judith Dube[100], Charlotte Humphrey[100], Sarah Jenkins[100], Emma Langridge[100], Rebecca Milne[100], Beverley Wadams[100] & Megan Woolcock[100]

**Royal Alexandra Hospital, Paisley, UK**
Michael Brett[101], Brian Digby[101], Lisa Gemmell[101], James Hornsby[101], Patrick MacGoey[101], Pauline O'Neil[101], Richard Price[101], Radha Sundaram[101], Lynn Abel[101], Natalie Rodden[101], Nicola Thomson[101], Kevin Rooney[101], Susan Currie[101], Natasha Parker[101], Lauren Walker[101] & Philip Henderson[101]

**St James's University Hospital and Leeds General Infirmary, Leeds, UK**
Bethan Ogg[102], Simon Whiteley[102], Liz Wilby[102], Kate Long[102], Shailamma Matthew[102], Sheila Salada[102], Susan Trott[102], Sarah Watts[102], Zoe Friar[102] & Abigail Speight[102]

**Bedford Hospital, Bedford, UK**
Victoria Bastion[103], Humza Chandna[103], Brice Djeugam[103], Muhammad Haseeb[103], Harriet Kent[103], Gamu Lubimbi[103], Sophie Murdoch[103], Alastair Thomas[103], Beena David[103], Rachel Lorusso[103], Ana Vochin[103], Melchizedek Penacerrada[103] & Retno Wulandari[103]

**Southport and Formby District General Hospital, Ormskirk, UK**
Charlotte Heath[104], Srinivas Jakkula[104], Anna Morris[104], Ashar Ahmed[104], Arvind Nune[104], Claire Buttriss[104] & Emma Whitaker[104]

**The Tunbridge Wells Hospital and Maidstone Hospital, Kent, UK**
Miriam Davey[105], David Golden[105], Amy Ackery[105], Fabio Fernandes[105], Bec Seaman[105] & Victoria Earl[105]

**Queen Elizabeth Hospital, Woolwich, London, UK**
Amy Collins[106], Waqas Khaliq[106], Rachel Adam[106] & Estefania Treus[106]

**North Manchester General Hospital, Manchester, UK**
Sarah Holland[107], Jordan Alfonso[107], Bethan Blackledge[107], Michelle Bruce[107], Laura Jayne Durrans[107], Ayaa Eltayeb[107], Jade Harris[107], Samuel Hey[107], Martin Hruska[107], Thomas Lamb[107], Joanne Rothwell[107], Adele Fitzgerald[107], Gabriella Lindergard[107], Helen T-Michael[107], Tracey Duncan[107], Sharon Baxter-Dore[107], Lisa Cooper[107], Claire Fox[107], Jacinta Guerin[107], Tracey Hodgkiss[107] & Karen Connolly[107]

**Royal Victoria Infirmary, Newcastle Upon Tyne, UK**
Paul McAlinden[108], Victoria Bridgett[108], Maggie Fearby[108], A. Gulati[108], Helen Hanson[108], Sinead Kelly[108], Louise McCormack[108], Rachel Nixon[108], Philip Robinson[108], Victoria Slater[108], Elaine Stephenson[108], Andrea Webster[108], K. Webster[108], Carole Hays[108], Anne Hudson[108], Bijal Patel[108], Ian Clement[108], John Davis[108], Sarah Francis[108] & Douglas Jerry[108]

**Hull Royal Infirmary, Hull, UK**
Caroline Abernathy[109], Louise Foster[109], Andrew Gratrix[109], Llucia Cabral-Ortega[109], Matthew Hines[109], Victoria Martinson[109], Elizabeth Stones[109] & Karen Winter[109]

**Manchester Royal Infirmary, Manchester, UK**
Esther Barrow[110], Katharine Wylie[110], Deborah Baines[110], Katie Birchall[110], Laurel Kolakaluri[110], Richard Clark[110], Anila Sukumaran[110], Craig Brandwood[110], Melanie Barker[110], Deborah Pariprooani[110], Lara Smith[110] & Charlotte Taylor[110]

**Royal Derby Hospital, Derby, UK**
Charlotte Downes[111], Melanie Hayman[111], Katie Riches[111], Priya Daniel[111], Deepak Subramanian[111], Kathleen Holding[111], Mary Hilton[111], Carly McDonald[111] & Georgina Richardson[111]

**Aintree University Hospital, Liverpool, UK**
Georgia Halladay[112], Peter Harding[112], Amie Reddy[112], Ian Turner-Bone[112], Laura Wilding[112], Robert Parker[112], Michaela Lloyd[112], Leanne Smith[112] & Charlie Kelly[112]

**Fairfield General Hospital, Bury, UK**
Maria Lazo[113], Alan Neal[113], Olivia Walton[113], Julie Melville[113], Jay Naisbitt[113], Emily Bullock[113] & Rosane Joseph[113]

**Norfolk and Norwich University hospital (NNUH), Norwich, UK**
Sara Callam[114], Lisa Hudig[114], Jocelyn Keshet-Price[114], Katie Stammers[114], Karen Convery[114], Georgina Randell[114] & Deirdre Fottrell-Gould[114]

**Milton Keynes University Hospital, Milton Keynes, UK**
Esther Mwaura[115], Sara-Beth Sutherland[115], Richard Stewart[115], Louise Mew[115] & Lynn Wren[115]

**Good Hope Hospital, Birmingham, UK**
Laura Thrasyvoulou[116], Heather Willis[116], James Scriven[116], Bridget Hopkins[116], Daniel Lenton[116] & Abigail Roberts[116]

**Queen Elizabeth Hospital Gateshead, Gateshead, UK**
Maria Bokhari[117], Rachael Lucas[117], Wendy McCormick[117], Jenny Ritzema[117], Vanessa Linnett[117], Amanda Sanderson[117] & Helen Wild[117]

**Royal Bolton Hospital, Bolton, UK**
Rebecca Flanagan[118], Robert Hull[118], Kat Rhead[118], Emma McKenna[118], Gareth Hughes[118], Jennifer Anderson[118], Kelly Jones[118], Scott Latham[118] & Heather Riley[118]

**Tameside General Hospital, Ashton Under Lyne, UK**
Martina Coulding[119], Martyn Clark[119], Jacqueline McCormick[119], Oliver Mercer[119], Darsh Potla[119], Hafiz Rehman[119], Heather Savill[119], Victoria Turner[119], Edward Jude[119] & Susan Kilroy[119]

**Salford Royal Hospital, Manchester, UK**
Elena Apetri[120], Cathrine Basikolo[120], Bethan Blackledge[120], Laura Catlow[120], Matthew Collis[120], Reece Doonan[120], Jade Harris[120], Alice Harvey[120], Karen Knowles[120], Stephanie Lee[120], Diane Lomas[120], Chloe Lyons[120], Liam McMorrow[120], Angiy Michael[120], Jessica Pendlebury[120], Jane Perez[120], Maria Poulaka[120], Nicola Proudfoot[120], Kathryn Slevin[120], Vicky Thomas[120], Danielle Walker[120], Paul Dark[120], Bethan Charles[120], Danielle McLaughlan[120], Melanie Slaughter[120], Dan Horner[120], Kathryn Cawley[120] & Tracy Marsden[120]

**Great Ormond St Hospital and UCL Great Ormond St Institute of Child Health NIHR Biomedical Research Centre, London, UK**
Joyann Andrews[121], Emily Beech[121], Olugbenga Akinkugbe[121], Alasdair Bamford[121], Holly Belfield[121], Gareth A. L. Jones[121], Tara McHugh[121], Hamza Meghari[121], Samiran Ray[121], Ana Luisa Tomas[121], Lauran O'Neill[121], Mark Peters[121], Michael Bell[121], Sarah Benkenstein[121], Catherine Chisholm[121], Charlene Davies[121], Klaudia Kupiec[121] & Caroline Payne[121]

**Southmead Hospital, Bristol, UK**
Joanna Halls[122], Hayley Blakemore[122], Elizabeth Goff[122], Kati Hayes[122], Kerry Smith[122], Deanna Stephens[122], Ruth Worner[122], Borislava Borislavova[122], Beverley Faulkner[122], Matt Thomas[122], Ruth Cookson[122], Emma Gendall[122], Georgina Larman[122], Rebecca Pope[122] & Artur Smalira[122]

**William Harvey Hospital, Ashford, UK**
Victoria Priestley[123], Tracey Cosier[123], Gemma Millen[123], James Rand[123], Natasha Schumacher[123], Roxana Sandhar[123], Heather Weston[123], Neil Richardson[123] & Lucy Cooper[123]

**Arrowe Park Hospital, Wirral, UK**
Cathy Jones[124], Ya-Wen Jessica Huang[124], Reni Jacob[124], Craig Denmade[124] & Lewis McIntyre[124]

**Royal Hampshire County Hospital, Hampshire, UK**
Dawn Trodd[125], Jane Martin[125], Geoff Watson[125], Emily Bevan[125] & Caroline Wreybrown[125]

**Bradford Royal Infirmary, Bradford, UK**
Shereen Bano[126], Ruth Bellwood[126], Michael Bentley[126], Matt Bromley[126], Lucy Gurr[126], Camilla Ledgard[126], Janet McGowan[126], Kate Pye[126], Kirsten Sellick[126], Amelia Stacey[126], Deborah Warren[126], Brian Wilkinson[126], Louise Akeroyd[126], Huma Shafique[126], James Morgan[126], Susan Shorter[126], Rachel Swinger[126], Emily Waters[126] & Tom Lawton[126]

**Glan Clwyd Hospital, Bodelwyddan, UK**
Elizabeth Allan[127], Kate Darlington[127], Ffyon Davies[127], Llinos Davies[127], Jack Easton[127], Sumit Kumar[127], Richard Lean[127], Callum Mackay[127], Richard Pugh[127], Xinyi Qiu[127], Stephanie Rees[127], Jeremy Scanlon[127], Joanne Lewis[127], Daniel Menzies[127], Annette Bolger[127], Gwyneth Davies[127], Jennifer Davies[127], Esther Garrod[127], Helen Jones[127], Rachel Manley[127] & Hannah Williams[127]

**Royal Bournemouth Hospital, Bournemouth, UK**
Jordan Frankham[128], Sally Pitts[128], Nigel White[128], Debbie Branney[128] & Heather Tiller[128]

**Bristol Royal Infirmary, Bristol, UK**
Georgia Efford[129], Zoe Garland[129], Lisa Grimmer[129], Bethany Gumbrill[129], Rebekah Johnson[129], Katie Sweet[129], Jeremy Bewley[129], Christina Coleman[129], Katie Corcoran[129], Eva Maria Hernandez Morano[129], Rachel Shiel[129], Denise Webster[129], Josephine Bonnici[129], Eleanor Daniel[129] & Abbie Dell[129]

**University Hospital North Durham, Darlington, UK and Darlington Memorial Hospital, Darlington, UK**
Melanie Kent[130,131], Ami Wilkinson[130,131], Ellen Brown[130,131], Andrea Kay[130,131], Suzanne Campbell[130,131], Amanda Cowton[130,131], Mark Birt[130,131], Vicki Greenaway[130,131], Kathryn Potts[130,131], Clare Hutton[130,131] & Andrew Shepperson[130,131]

**Basildon Hospital, Basildon, UK**
Miranda Forsey[132], Alice Nicholson[132], Mark Vertue[132], Joanne Riches[132], Agilan Kaliappan[132] & Anne Nicholson[132]

**University College Hospital, London, UK**
Niall MacCallum[133], Eamon Raith[133], Georgia Bercades[133], Ingrid Hass[133], David Brealey[133], Gladys Martir[133], Anna Reyes[133], Deborah Smyth[133] & Maria Zapatamartinez[133]

**Whittington Hospital, London, UK**
Ana Alvaro[134], Champa Jetha[134], Louise Ma[134], Lauren Booker[134], Loreta Mostoles[134], Anezka Pratley[134], Abdelhakim Altabaibeh[134], Chetan Parmar[134] & Kayleigh Gilbert[134]

**Western General Hospital, Edinburgh, UK**
Susie Ferguson[135], Amy Shepherd[135], Sheila Morris[135], Jo Singleton[135], Rosie Baruah[135], Maria Amamio[135], Sophie Birch[135], Kate Briton[135], Sarah Clark[135], Katherine Doverman[135], Lucy Marshall[135] & Scott Simpson[135]

**Ipswich Hospital, Ipswich, UK**
Georgina Lloyd[136], Stephanie Bell[136], Vanessa Rivers[136] & Bally Purewal[136]

**Hereford County Hospital, Hereford, UK**
Kate Hammerton[137], Susan Anderson[137], Janine Birch[137], Emma Collins[137] & Ryan Oleary[137]

**Sunderland Royal Hospital, Sunderland, UK**
Sarah Cornell[138], Jordan Jarmain[138], Kimberley Rogerson[138], Fiona Wakinshaw[138], Lindsey Woods[138], Anthony Rostron[138], Zeynep Elcioglu[138] & Alistair Roy[138]

**Queens Hospital Burton, Burton-On-Trent, UK**
Gillian Bell[139], Holly Dickson[139], Louise Wilcox[139], Amro Katary[139] & Katy English[139]

**Musgrove Park Hospital, Taunton, UK**
Joanne Hutter[140], Corinne Pawley[140], Patricia Doble[140], Charmaine Shovelton[140], Marius Vaida[140], Rebecca Purnell[140] & Ashly Thomas[140]

**The Royal Papworth Hospital, Cambridge, UK**
Lenka Cagova[141], Adama Fofano[141], Helen Holcombe[141], Alice Michael Mitchell[141], Lucy Mwaura[141], Krithivasan P. Raman[141], Lucie Garnr[141], Sue Mepham[141], Kitty Paques[141], Alain Vuylsteke[141], Jennifer Mackie[141], Carmen Pearn[141] & Julie Zamikula[141]

**University Hospital Lewisham, London, UK**
Mark Birt[142], Estefania Treus Gude[142], Maggie Nyirenda[142], Lisa Capozzi[142], Rosie Reece-Anthony[142], Waqas Khaliq[142], Hazma Noor[142] & Alfa Cresia Nilo[142]

**The Princess Alexandra Hospital, Harlow, UK**
Michelle Grove[143], Amelia Daniel[143], Amy Easthope[143], Joanne Finn[143], Nikki White[143], Rajnish Saha[143], Bibi Badal[143] & Karen Ixer[143]

**University Hospital of Wales, Cardiff, UK**
Donna Duffin[144], Ben Player[144], Helen Hill[144], Jade Cole[144], Jenny Brooks[144], Michelle Davies[144], Rhys Davies[144], Lauren Hunt[144], Emma Thomas[144] & Angharad Williams[144]

**West Middlesex Hospital, Isleworth, UK**
Metod Oblak[145], Mini Thankachen[145], Jamie Irisari[145], Amrinder Sayan[145] & Monica Popescu[145]

**Royal Albert Edward Infirmary, Wigan, UK**
Cheryl Finch[146], Andrew Jamieson[146], Alison Quinn[146], Joshua Cooper[146], Sarah Liderth[146] & Natalia Waddington[146]

**Stoke Mandeville Hospital, Buckinghamshire, UK**
Iona Burn[147], Katarina Manso[147], Ruth Penn[147], Julie Tebbutt[147], Danielle Thornton[147], James Winchester[147], Geraldine Hambrook[147] & Pradeep Shanmugasundaram[147]

**Royal Lancaster Infirmary, Lancaster, UK**
Jayne Craig[148], Kerry Simpson[148], Andrew Higham[148] & Louise Sibbett[148]

**Basingstoke and North Hampshire Hospital, Basingstoke, UK**
Sheila Paine[149], Annabel Reed[149], Jo-Anna Conyngham[149], McDonald Mupudzi[149], Rachel Thomas[149], Mary Wright[149], Denise Griffin[149], Richard Partridge[149], Maria Alvarez Corral[149], Nycola Muchenje[149], Mildred Sitonik[149] & Caroline Wrey Brown[149]

**Worthing Hospital, Worthing, UK and St Richard's Hospital, Chichester, UK**
Aaron Butler[150,151], Linda Folkes[150,151], Heather Fox[150,151], Amy Gardner[150,151], David Helm[150,151], Gillian Hobden[150,151], Kirsten King[150,151], Jordi Margalef[150,151], Michael Margarson[150,151], Tim Martindale[150,151], Emma Meadows[150,151], Dana Raynard[150,151], Yvette Thirlwall[150,151], Yolanda Baird[150,151], Raquel Gomez[150,151], Darren Martin[150,151], Luke Hodgson[150,151], Clinton Corin[150,151], Erikka Sidall[150,151], Densie Szabo[150,151] & Sharon Floyd[150,151]

**The Alexandra Hospital, Redditch and Worcester Royal Hospital, Worcester, UK**
Hannah Davies[152], Karen Austin[152], Olivia Kelsall[152], Hannah Wood[152], Hannah Davies[152], Peter Anderson[152], Katie Archer[152], Andrew Burtenshaw[152], Sarah Clayton[152], Naiara Cother[152], Nicholas Cowley[152], Caroline Davis[152], Stephen Digby[152], Alison Durie[152], Alison Harrison[152], Emma Low[152], Michael McAlindon[152], Alex McCurdy[152], Aled Morgan[152], Tobias Rankin[152], Jessica Thrush[152], Helen Tranter[152], Charlie Vigurs[152], Laura Wild[152], Karen Austin[152], Olivia Kelsall[152] & Hannah Wood[152]

**Royal Cornwall Hospital, Truro, UK**
Thomas Cornell[153], Kate Ralph[153], Sarah Bean[153], Karen Burt[153], Michael Spivey[153], Carol Richards[153] & Rachel Tedstone[153]

**Watford General Hospital, Watford, UK**
Siobhain Carmody[154], Xiaobei Zhao[154], Valerie Page[154], Mark Louie Guanco[154], Elvira Hoxha[154] & Camilla Zorloni[154]

**Macclesfield District General Hospital, Macclesfield, UK**
Charlotte Dean[155], Emma Jones[155], Emma Carter[155], Joshua Dunn[155], Thomas Kong[155], Mervin Mahenthran[155], Chris Marsh[155], Maureen Holland[155], Natalie Keenan[155], Mohamed Mahmoud[155], Marc Lyons[155], Joanne Bradley-Potts[155], Helen Wassall[155] & Meghan Young[155]

**Royal Surrey County Hospital, Guildford, UK**
Paul Bradley[156], Dorota Burda[156], Sinead Donlon[156], Lesley Harden[156], Celia Harris[156], Irving Mayangao[156], Rugia Montaser[156], Sheila Mtuwa[156], Charles Piercy[156], Eleanor Smith[156], Sarah Stone[156], Jerik Verula[156], Helen Blackman[156], Cheryl Marriott[156], Natalia Michalak[156], Ben Creagh-Brown[156], Armorel Salberg[156], Naomi Boyer[156] & Veronika Pristopan[156]

**Rotherham General Hospital, Rotherham, UK**
Victoria Maynard[157], Rachel Walker[157], Anil Hormis[157], Dawn Collier[157], Cheryl Graham[157], Vicky Maynard[157], Jake McCormick[157] & Jake Warrington[157]

**Craigavon Area Hospital, County Armagh, Northern Ireland**
Denise Cosgrove[158], Denise McFarland[158], Judith Ratcliffe[158] & Rob Charnock[158]

**King's Mill Hospital, Nottingham, UK**
Inez Wynter[159], Mandy Gill[159], Jill Kirk[159], Paul Paul[159], Valli Ratnam[159] & Sarah Shelton[159]

**Dumfries and Galloway Royal Infirmary, Dumfries, UK**
Catherine Jardine[160], Alasdair Hay[160] & Dewi Williams[160]

**Prince Charles Hospital, Merthyr Tydfil, UK**
Bethan Deacon[161], Latha Durga[161], Meg Hibbert[161], Gareth Kennard-Holden[161], Chrsitopher Woodford[161], Carla Pothecary[161], Lisa Roche[161], Dariusz Tetla[161], Kevin Agravante[161] & Justyna Smeaton[161]

**Ysbyty Gwynedd, Bangor, UK**
Alicia Price[162], Alice Thomas[162], Chris Thorpe[162], Ellen Knights[162] & Donna Ward[162]

**Royal Preston Hospital, Preston, UK**
Shondipon Laha[163], Mark Verlander[163] & Alexandra Williams[163]

**The Great Western Hospital, Swindon, UK**
Rachel Prout[164], Helen Langton[164], Malcolm Watters[164], Charlotte Hunt[164] & Catherine Novis[164]

**Lincoln County Hospital, Lincoln, UK**
Sarwat Arif[165], Amy Cunningham[165], Claire Hewitt[165], Julia Hindale[165], Karen Jackson-Lawrence[165], Sarah Shepardson[165], Maryanne Wills[165], Susie Butler[165], Silivia Tavares[165], Russell Barber[165], Annette Hilldrith[165] & Kelly Hubbard[165]

**University Hospital of North Tees, Stockton on Tees, UK**
Dawn Egginton[166], Michele Clark[166], Sarah Purvis[166], Simon Sinclair[166] & Vicky Collins[166]

**Glangwili General Hospital, Camarthen, UK**
Bethan Landeg[167], Craig Sell[167], Samantha Coetzee[167], Alistair Gales[167], Igor Otahal[167], Becky Icke[167], Meena Raj[167], Caroline Williams[167], Jill Williams[167] & Lucy Hill[167]

**Southend University Hospital, Westcliff-on-Sea, UK**
Abdul Kayani[168], Bridgett Masunda[168], Prisca Gondo[168] & Nigara Atayeva[168]

**Lister Hospital, Stevenage, UK**
Carina Cruz[169] & Natalie Pattison[169]

**Diana Princess of Wales Hospital, Grimsby, UK**
Caroline Burnett[170], Jonathan Hatton[170], Elaine Heeney[170], Maria Newton[170], Hassan Al-Moasseb[170], Teresa Behan[170], Jasmine Player[170], Rachael Stead[170], Atideb Mitra[170] & Kirsty Nauyokas[170]

**West Suffolk Hospital, Bury St Edmunds, UK**
Sally Humphreys[171], Helen Cockerill[171] & Ruth Tampsett[171]

**Victoria Hospital, Kirkcaldy, UK**
Evgeniya Postovalova[172], Tina Coventry[172], Amanda McGregor[172], Susan Fowler[172], Mike Macmahon[172], Patricia Cochrane[172] & Sandra Pirie[172]

**Calderdale Royal Hospital, Halifax, UK and Huddersfield Royal Infirmary, Huddersfield, UK**
Sarah Hanley[173,174], Asifa Ali[173,174], Megan Brady[173,174], Sam Dale[173,174], Annalisa Dance[173,174], Lisa Gledhill[173,174], Jill Greig[173,174], Kathryn Hanson[173,174], Kelly Holdroyd[173,174], Marie Home[173,174], Tahira Ishaq[173,174], Diane Kelly[173,174], Lear Matapure[173,174], Deborah Melia[173,174], Samantha Mellor[173,174], Ekta Merwaha[173,174], Tonicha Nortcliffe[173,174], Lisa Shaw[173,174], Ryan Shaw[173,174], Tracy Wood[173,174], Lee-Ann Bayo[173,174], Miranda Usher[173,174], Alison Wilson[173,174], Ross Kitson[173,174], Jez Pinnell[173,174], Matthew Robinson[173,174] & Kaitlin Boltwood[173,174]

**Dorset County Hospital, Dorchester, UK**
Jenny Birch[175], Laura Bough[175], Rebecca Tutton[175], Barbara Winter-Goodwin[175], Josie Goodsell[175], Kate Taylor[175], Patricia Williams[175], Sarah Williams[175], Ashleigh Cave[175] & James Rees[175]

**Russell's Hall Hospital, Dudley, UK**
Janet Imeson-Wood[176], Jacqueline Smith[176], Vishal Amin[176], Komala Karthik[176], Rizwana Kausar[176], Elena Anastasescu[176], Karen Reid[176], Vikram Anumakonda[176] & Ella Stoddart[176]

**Royal United Hospital, Bath, UK**
Carrie Demetriou[177], Charlotte Eckbad[177], Lucy Howie[177], Sarah Mitchard[177], Lidia Ramos[177], Katie White[177], Sarah Hierons[177], Fiona Kelly[177], Alfredo Serrano-Ruiz[177] & Gabrielle Evans[177]

**St Mary's Hospital, Newport, UK**
Liz Nicol[178], Joy Wilkins[178], Kim Hulacka[178], Gabor Debreceni[178], Alison Brown[178] & Vikki Crickmore[178]

**George Eliot Hospital NHS Trust, Nuneaton, UK**
Kay Hill[179] & Thogulava Kannan[179]

**Yeovil Hospital, Yeovil, UK**
Zenaida Dagutao[180], Kate Beesley[180], Alison Lewis[180], Jess Perry[180], Sherly Antony[180], Sarah Board[180], Clare Buckley[180], Lucy Pippard[180], Alfonso Tanate[180], Diane Wood[180], Agnieszka Kubisz-Pudelko[180] & Ayman Gouda[180]

**Forth Valley Royal Hospital, Falkirk, UK**
Fiona Auld[181], Joanne Donnachie[181], Euan Murdoch[181], Lynn Prentice[181], Nikole Runciman[181], Dhaneesha Senaratne[181], Abigail Short[181], Laura Sweeney[181], Lesley Symon[181], Anne Todd[181], Patricia Turner[181], Erin McCann[181], Dario Salutous[181], Ian Edmond[181] & Lesley Whitelaw[181]

**Frimley Park Hospital, Surrey, UK**
Harish Venkatesh[182], Yvonne Bland[182], Istvan Kajtor[182], Lisa Kavanagh[182], Karen Singler[182] & George Linfield-Brown[182]

**Chelsea & Westminster NHS Foundation Trust, London, UK**
Luke Stephen Prockter Moore[183], Marcela Vizcaychipi[183], Laura Martins[183], Luke Moore[183], Rhian Bull[183] & Jaime Carungcong[183]

**Queen Elizabeth the Queen Mother Hospital, Margate, UK**
Louise Allen[184], Eva Beranova[184], Alicia Knight[184], Carly Price[184], Sorrell Tilbey[184], Sharon Turney[184], Tracy Hazelton[184], Gabriella Tutt[184], Mansi Arora[184], Salah Turki[184], Emily Sinfield[184], Joanne Deery[184] & Hazel Ramos[184]

**Royal Brompton Hospital, London, UK**
Daniele Cristiano[185], Natalie Dormand[185], Zohreh Farzad[185], Mahitha Gummadi[185], Sara Salmi[185], Geraldine Sloane[185], Mathew Varghese[185], Vicky Thwaites[185], Brijesh Patel[185], Liyanage Kamal[185] & Anelise Catelan Zborowski[185]

**Darent Valley Hospital, Dartford, UK**
Ryan Coe[186], Madeleine Anderson[186], Jane Beadle[186], Charlotte Coates[186], Katy Collins[186], Maria Crowley[186], Laura Johnson[186], Laura King[186], Remi Paramsothy[186], Janet Sargeant[186], Pedro Silva[186], Carmel Stuart[186], June Taylor[186], David Tyl[186], Phillipa Wakefield[186], Charlotte Kamundi[186], Olumide Olufuwa[186], Zakaulla Belagodu[186], Anca Gherman[186] & Naomi Oakley[186]

**University Hospital Crosshouse, Kilmarnock, UK**
John Allan[187], Tim Geary[187], Alistair Meikle[187], Peter O'Brien[187], Stephen Wood[187], Andrew Clark[187] & Gordon Houston[187]

**University Hospital Wishaw, Wishaw, UK**
Karen Black[188], Michelle Clarkson[188], Stuart D'Sylva[188], Alan Morrison[188], Kathryn Norman[188], Margaret Taylor[188], Suzanne Clements[188], Catriona Cohrane[188], Nora Gonzalez[188], Dominic Strachan[188], Claire Beith[188] & Kirsten Moar[188]

**University College Dublin, St Vincent's University Hospital, Dublin, Ireland**
Lorna Murphy[189], Michelle Smythe[189], Alistair Nichol[189] & Kathy Brickell[189]

**The Queen Elizabeth Hospital, King's Lynn, UK**
Inthakab Ali Mohamed Ali[190], Karen Beaumont[190], Mohamed Elsaadany[190], Kay Fernandes[190], Sameena Mohamed Ally[190], Harini Rangarajan[190], Varun Sarathy[190], Sivarupan Selvanayagam[190],

Dave Vedage[190], Matthew White[190], Zoe Coton[190], Aricsa Joshy[190], Mark Blunt[190] & Hollie Curgenven[190]

**Walsall Manor Hospital, Walsall, UK**
Liam Botfield[191], Catherine Dexter[191], Aditya Kuravi[191], Joanne Butler[191], Robert Chadwick[191], Poonam Ranga[191], Lisa Richardson[191], Emma Virgilio[191], Maddiha Anwer[191], Atul Garg[191], Donna Botfield[191] & Xana Marriott[191]

**Princess Royal Hospital, Brighton, UK**
Keely Stewart[192], Dee Mullan[192], Claire Phillips[192], Jane Gaylard[192], Justyna Nowak[192] & Denise Skinner[192]

**Barnsley Hospital, Barnsley, UK**
Sian Jones[193], Rikki Crawley[193], Abigail Crew[193], Mishell Cunningham[193], Allison Daniels[193], Laura Harrison[193], Susan Hope[193], Nicola Lancaster[193], Jamie Matthews[193], Gemma Wray[193], Alice Nicholson[193], Ken Inweregbu[193], Sarah Cutts[193] & Katharine Miller[193]

**Warrington General Hospital, Warrington, UK**
Ailbhe Brady[194], Rebekah Chan[194], Shane McIvor[194], Helena Prady[194], Bijoy Mathew[194], Jeff Little[194] & Tim Furniss[194]

**Royal Victoria Hospital, Belfast, NI**
Chris Wright[195], Bernadette King[195], Christopher Wasson[195], Aisling O'Neill[195], Christine Turley[195], Peter McGuigan[195], Erin Collins[195], Stephanie Finn[195], Jackie Green[195], Julie McAuley[195], Abitha Nair[195], Charlotte Quinn[195], Suzanne Tauro[195], Kathryn Ward[195], Michael McGinlay[195] & Kiran Reddy[195]

**Royal Hallamshire Hospital and Northern General Hospital, Sheffield, UK**
Norfaizan Ahmad[196], Samantha Anderson[196], Joann Barker[196], Kris Bauchmuller[196], Kathryn Birchall[196], Sarah Bird[196], Kay Cawthron[196], Luke Chetam[196], Joby Cole[196], Ben Donne[196], David Foote[196], Amber Ford[196], Helena Hanratty[196], Kate Harrington[196], Lisa Hesseldon[196], Kay Housley[196], Yvonne Jackson[196], Claire Jarman[196], Faith Kibutu[196], Becky Lenagh[196], Irene Macharia[196], Shamiso Masuko[196], Leanne Milner[196], Helen Newell[196], Lorenza Nwafor[196], Simon Oxspring[196], Patrick Phillips[196], Ajay Raithatha[196], Sarah Rowland-Jones[196], Jacqui Smith[196], Roger Thompson[196], Helen Trower[196], Sara Walker[196], James Watson[196], Matthew Wiles[196], Alison Lye[196], Jayne Willson[196], Gary Mills[196], Sansha Harris[196] & Eleanor Hartill[196]

**Harefield Hospital, London, UK**
Anthony Barron[197], Ciara Collins[197], Sundeep Kaul[197], Claire Nolan[197], Oliver Polgar[197], Claire Prendergast[197], Paula Rogers[197], Rajvinder Shokkar[197], Meriel Woodruff[197], Kanta Mahay[197], Vicky Thwaites[197], Anna Reed[197], Hayley Meyrick[197], Heather Passmore[197] & James Farwell[197]

**Cumberland Infirmary, Carlisle, UK**
Alison Brown[198], Susan O'Connell[198], Jane Gregory[198], Luigi Barberis[198], Rosemary Harper[198], Tim Smith[198] & Diane Armstrong[198]

**Eastbourne District General Hospital, East Sussex, UK and Conquest Hospital, East Sussex, UK**
Angie Bowey[199,200], Anne Cowley[199,200], Andrew Corner[199,200], Judith Highgate[199,200], Claire Rutherfurd[199,200], Jo-Anne Taylor[199,200], Sarah Goodwin[199,200] & Claire Rutherford[199,200]

**Salisbury District Hospital, Salisbury, UK**
Beena Eapen[201], Fiona Trim[201] & Phil Donnison[201]

**Airedale General Hospital, Keighley, UK**
Lisa Armstrong[202], Hayley Bates[202], Emma Dooks[202], Fiona Farquhar[202], Amy Kitching[202], Chantal McParland[202], Sophie Packham[202] & Brigid Hairsine[202]

**Leicester Royal Infirmary, Leicester, UK**
Anand Patil[203], Premetie Andreou[203], Dawn Hales[203], Megha Mathews[203], Rekha Patel[203], Peter Barry[203], Neil Flint[203], Jessica Hailstone[203], Navneet Ghuman[203], Bethany Leonard[203] & Rachel Lees[203]

**Peterborough City Hospital, Peterborough, UK and Hinchingbrooke Hospital, Huntingdon, UK**
Deborah Butcher[204,205], Katy Leng[204,205], Nicola Butterworth-Cowin[204,205] & Susie O'Sullivan[204,205]

**Colchester General Hospital, Colchester, UK**
Alison Ghosh[206] & Emma Williams[206]

**Princess Royal Hospital, Telford and Royal Shrewsbury Hospital, Shrewsbury, UK**
Colene Adams[207], Anita Agasou[207], Tracie Arden[207], Mandy Beekes[207], Amy Bowes[207], Pauline Boyle[207], Heather Button[207], Mandy Carnahan[207], Anne Carter[207], Danielle Childs[207], Jane Gaylard[207], Fran Hurford[207], Yasmin Hussain[207], Ayesha Javaid[207], James Jones[207], Michael Leigh[207], Terry Martin[207], Helen Millward[207], Nichola Motherwell[207], Dee Mullan[207], Julie Newman[207], Rachel Rikunenko[207], Jo Stickley[207], Julie Summers[207], Louise Ting[207], Helen Tivenan[207], Denise Donaldson[207], Nigel Capps[207], Emily Cale[207], Sanal Jose[207], Wendy Osbourne[207], Susie Pajak[207], Jayne Rankin[207] & Louise Tonks[207]

**University Hospital Monklands, Airdrie, UK**
Tracy Baird[208], Margaret Harkins[208], Jim Ruddy[208] & Joe West[208]

**Wrexham Maelor Hospital, Wrexham, Wales**
Joseph Duffield[209], Lewis Mallon[209], Oliver Smith[209], Sara Smuts[209], Andy Campbell[209], Cate Davies[209], Sarah Davies[209], Rachel Hughes[209], Lisa Jobes[209], Victoria Whitehead[209] & Clare Watkins[209]

**Royal Hospital for Children, Glasgow, UK**
Fiona Bowman[27], Barry Milligan[27], Colin Begg[27] & Liane McPherson[27]

**New Cross Hospital, Wolverhampton, UK**
Stella Metherell[210], Nichola Harris[210], Victoria Lake[210], Elizabeth Radford[210], Andy Smallwood[210], Shameer Gopal[210] & Katherine Vassell[210]

**University Hospital Hairmyres, East Kilbride, UK**
Dina Bell[211], Rosalind Boyle[211], Katie Douglas[211], Lynn Glass[211], Liz Lennon[211], Austin Rattray[211], Claire Beith[211] & Emma Lee[211]

**Warwick Hospital, Warwick, UK**
Danielle Jones[212], Penny Parsons[212], Ben Attwood[212], Paul Jefferson[212], Mohan Ranganathan[212], Inderjit Atwal[212], Bridget Campbell[212], Angela Day[212] & Camilla Stagg[212]

**Sandwell General Hospital and City Hospital, Birmingham, UK**
Emma Haynes[213], Cecilia Ahmed[213], Sarah Clamp[213], Julie Colley[213], Risna Haq[213], Anne Hayes[213], Sibet Joseph[213], Zahira Maqsood[213], Samia Hussain[213], Jonathan Hulme[213], Patience Domingos[213], Rita Kumar[213], Manjit Purewal[213] & Becky Taylor[213]

**Royal Manchester Children's Hospital, Manchester, UK**
Lara Bunni[214], Monica Latif[214], Claire Jennings[214], Shilu Jose[214], Rebecca Marshall[214], Aleksandra Metryka[214] & Gayathri Subramanian[214]

**Gloucestershire Royal Hospital, Gloucester, UK**
Adam Burgoyne[215], Susan O'Connell[215], Amanda Tyler[215], Joanne Waldron[215], Paula Hilltout[215] & Jayne Evitts[215]

**University Hospitals Coventry & Warwickshire NHS Trust, Coventry, UK**
Geraldine Ward[216], Pamela Bremmer[216], Carl Hawkins[216], Sophie Jackman[216] & Michal Ogorek[216]

**Torbay Hospital, Torquay, UK**
Kylie Ashby[217], Lorraine Thornton[217], Pauline Mercer[217], Matthew Halkes[217] & Adam Revill[217]

**Pilgrim Hospital, Lincoln, UK**
Bryony Saint[218], Jo Fletcher[218], Kimberley Netherton[218], Manish Chablani[218], Amy Kirkby[218], Amanda Roper[218] & Kinga Szymiczek[218]

**Prince Philip Hospital, Lianelli, UK**
Isobel Sutherland[219], Linda O'Brien[219], Igor Otahal[219], Joanne Connell[219], Kim Davies[219], Tracy Lewis[219], Zohra Omar[219] & Emma Perkins[219]

**Princess of Wales Hospital, Llantrisant, UK**
Lisa Roche[220], Sonia Sathe[220] & Ellie Davies[220]

**Northampton General Hospital NHS Trust, Northampton, UK**
Alex Lyon[221], Isheunesu Mapfunde[221], Charlotte Willis[221], Rachael Hitchcock[221], Kathryn Hall[221] & Christopher King[221]

**The Christie NHS Foundation Trust, Manchester, UK**
Andrew Fagan[222], Roonak Nazari[222], Lucy Worsley[222], Suzanne Allibone[222], Vidya Kasipandian[222], Amit Patel[222], Parisa Cutting[222], Roman Genetu[222], Ainhi Mac[222], Anthony Murphy[222], Sinead Ward[222] & Fatima Butt[222]

**James Paget University Hospital NHS Trust, Great Yarmouth, UK**
Amanda Ayers[223], Wendy Harrison[223], Katherine Mackintosh[223] & Julie North[223]

**Birmingham Children's Hospital, Birmingham, UK**
Lydia Ashton[224], Rehana Bi[224], Samantha Owen[224], Helen Winmill[224] & Barney Scholefield[224]

**Withybush General Hospital, Pembrokeshire, Wales**
Hannah Blowing[225], Erin Williams[225], Michaela Duskova[225], Michelle Edwards[225], Alun Rees[225], Helen Thomas[225], Rachel Hughes[225], Igor Otahal[225], Jolene Brooks[225], Janet Phipps[225] & Suzanne Brooks[225]

**Northwick Park Hospital, London, UK**
Catherine Dennis[226], Vicki Parris[226], Sinduya Srikaran[226], Anisha Sukha[226], Alistair McGregor[226] & Gerlynn Tiongson[226]

**North Devon District Hospital, Barnstaple, UK**
Katie Adams[227], Benedict Andrew[227], Adam Brayne[227], Sasha Carter[227], Louise Findlay[227], Emma Fisher[227], Peter Jackson[227], Duncan Kaye[227], Juliet Parkin[227], Victoria Tuckey[227], Jane Hunt[227], Nicholas Love[227], Lynne van Koutrick[227] & Ashley Hanson[227]

**Scunthorpe General Hospital, Scunthorpe, UK**
Kathy Dent[228], Elizabeth Horsley[228], Sandra Pearson[228], Sue Spencer[228], Dorothy Hutchinson[228], Jasmine Player[228], Dorota Potoczna[228], Muhammad Nauman Akhtar[228], Lisa-Jayne Cottam[228], Kirsty Nauyokas[228] & Jack Sanders[228]

**Royal Free Hospital, London, UK**
Sara Mingo Garcia[229], Glykeria Pakou[229], Cynthia Diaba[229], Helder Filipe[229], Lincy John[229], Amitaa Maharajh[229], Mark de Neef[229], Daniel Martin[229], Christine Eastgate[229] & Poh Choo Teoh[229]

**Raigmore Hospital, Inverness, UK**
Fiona Barrett[230], Clare Bradley[230], Avril Donaldson[230], Mairi Mascarenhas[230], Marianne O'Hara[230], Laura Okeefe[230], Noreen Clarke[230], Jonathan Whiteside[230], Rachael Campbell[230], Joanna Matheson[230], Deborah McDonald[230] & Donna Patience[230]

**West Cumberland Hospital, Whitehaven, UK**
Polly Rice[231], Tim Smith[231], Melanie Clapham[231], Rachel Mutch[231], Luigi Barberis[231], Rosemary Harper[231], Hannah Craig[231] & Una Poultney[231]

**Furness General Hospital, Barrow-in-Furness, UK**
Karen Burns[232] & Andrew Higham[232]

**Liverpool Heart and Chest Hospital, Liverpool, UK**
Sophie Twiss[233], Janet Barton[233], Linsha George[233], Clare Harrop[233], Sherly Mathew[233] & David Justin Wright[233]

**Scarborough General Hospital, Scarborough, UK**
Rachel Harrison[234], Jordan Toohie[234], Ben Chandler[234], Alison Turnbull[234], Janine Mallinson[234] & Kerry Elliott[234]

**Bronglais General Hospital, Aberystwyth, UK**
Rebecca Wolf-Roberts[235], Helen Tench[235], Igor Otahal[235], Maria Hobrok[235], Ronda Loosley[235], Heather McGuinness[235] & Tanya Sims[235]

**Alder Hey Children's Hospital, Liverpool, UK**
Deborah Afolabi[236], Kathryn Sian Allison[236], Taya Anderson[236], Rachael Dore[236], Dawn Jones[236], Naomi Rogers[236], Paula Saunderson[236], Jennifer Whitbread[236], Laura O'Malley[236], Laura Rad[236] & Daniel Hawcutt[236]

**Borders General Hospital, Melrose, UK**
Jonathan Aldridge[237], Melanie Tolson[237] & Sweyn Garrioch[237]

**Leighton Hospital, Cheshire, UK**
Joanne Tomlinson[238] & Michael Grosdenier[238]

**Kent & Canterbury Hospital, Canterbury, UK**
David Loader[239], Ritoo Kapoor[239] & Gemma Hector[239]

**Harrogate and District NHS Foundation Trust, Harrogate, UK**
Joslan Scherewode[240], Chunda Sri-Chandana[240], Lorraine Stephenson[240] & Sarah Marsh[240]

**The Royal Marsden Hospital, London, UK**
Arnold Dela Rosa[241], Shaman Jhanji[241], Thomas Bemand[241], Ryan Howle[241], Ravishankar Rao Baikady[241], Benjamin Thomas[241], Ethel Black[241] & Kate Tatham[241]

**Ealing Hospital, Southall, UK**
Sambasivarao Gurram[242], Ekaterina Watson[242], Vicki Parris[242], Sheena Quaid[242] & Alistair McGregor[242]

**St John's Hospital Livingston, Livingston, UK**
Anne Saunderson[243], Rachel O'Brien[243], Sam Moultrie[243], Jen Service[243], Clare Cheyne[243], Miranda Odam[243] & Alison Wiliams[243]

**Wexham Park Hospital, Slough, UK**
Nicky Barnes[244], Peter Csabi[244], Joana Da Rocha[244] & Louika Glynou[244]

**Sheffield Children's Hospital, Sheffield, UK**
Amy Huffenberger[245], Jade Bryant[245], Amy Pickard[245], Nicholas Roe[245], Arianna Bellini[245], Anton Mayer[245], Amy Burrow[245], Natalie Colley[245], Jayne Evans[245], Alex Howlett[245] & Zeinab Khalifeh[245]

**Homerton University Hospital Foundation NHS Trust, London UK**
Jerldine Pryce[246], Claire Gorman[246], Amy Easthope[246], Rebecca Brady[246], Elizabeth Timlick[246], Pierre Antoine[246] & Abhinhav Gupta[246]

**National Hospital for Neurology and Neurosurgery, London, UK**
John Hardy[247], Henry Houlden[247], Eleanor Moncur[247], Arianna Tucci[247], Eamon Raith[247], Ambreen Tariq[247] & David Brealey[247]

**The Royal Alexandra Children's Hospital, Brighton, UK**
Emma Tagliavini[248], Becky Ramsay[248], Katy Fidler[248], Kevin Donnelly[248] & Rebecca Hollis[248]

**Golden Jubilee National Hospital, Clydebank, UK**
Jocelyn Barr[249], Elizabeth Boyd[249], Val Irvine[249], Ben Shelley[249], Julie Buckley[249], Charlene Hamilton[249] & Kathryn Valdeavella[249]

[49]NIHR Clinical Research Network (CRN), North West London Core Team, Hammersmith Hospital, London, UK. [50]Cambridge University Hospitals NHS Foundation Trust, Cambridge, UK. [51]Biostatistics Group, State Key Laboratory of Biocontrol, School of Life Sciences, Sun Yat-sen University, Guangzhou, China. [52]Department of Infectious Diseases, Leiden University Medical Center, Leiden, The Netherlands. [53]Guys and St Thomas' Hospital, London, UK. [54]James Cook University Hospital, Middlesbrough, UK. [55]Barts Health NHS Trust, London, UK. [56]Royal Stoke University Hospital, Stoke-on-Trent, UK. [57]North Middlesex University Hospital NHS Trust, London, UK. [58]King's College Hospital, London, UK. [59]Charing Cross Hospital, St Mary's Hospital and Hammersmith Hospital, London, UK. [60]The Royal Liverpool University Hospital, Liverpool, UK. [61]John Radcliffe Hospital, Oxford, UK. [62]Addenbrooke's Hospital, Cambridge, UK. [63]Nottingham University Hospital, Nottingham, UK. [64]St George's Hospital, London, UK. [65]BHRUT (Barking Havering)—Queens Hospital and King George Hospital, Ilford, UK. [66]Royal Infirmary of Edinburgh, Edinburgh, UK. [67]Kingston Hospital, London, UK. [68]Queen Alexandra Hospital, Portsmouth, UK. [69]Royal Gwent Hospital, Newport, UK. [70]Royal Blackburn Teaching Hospital, Blackburn, UK. [71]Stepping Hill Hospital, Stockport, UK. [72]Northumbria Healthcare NHS Foundation Trust, North Shields, UK. [73]Countess of Chester Hospital, Chester, UK. [74]Pinderfields General Hospital, Wakefield, UK. [75]Ninewells Hospital, Dundee, UK. [76]Croydon University Hospital, Croydon, UK. [77]Morriston Hospital, Swansea, UK. [78]Queen Elizabeth University Hospital, Glasgow, UK. [79]Broomfield Hospital, Chelmsford, UK. [80]Heartlands Hospital, Birmingham, UK. [81]Royal Sussex County Hospital, Brighton, UK. [82]York Hospital, York, UK. [83]Queen Elizabeth Hospital, Birmingham, UK. [84]Royal Glamorgan Hospital, Pontyclun, UK. [85]Barnet Hospital, London, UK. [86]Wythenshawe Hospital, Manchester, UK. [87]Medway Maritime Hospital, Gillingham, UK. [88]Royal Berkshire NHS Foundation Trust, Reading, UK. [89]Whiston Hospital, Prescot, UK. [90]The Royal Oldham Hospital, Manchester, UK. [91]Chesterfield Royal Hospital Foundation Trust, Chesterfield, UK. [92]Aberdeen Royal Infirmary, Aberdeen, UK. [93]Royal Devon and Exeter Hospital, Exeter, UK. [94]Glasgow Royal Infirmary, Glasgow, UK. [95]Blackpool Victoria Hospital, Blackpool, UK. [96]Southampton General Hospital, Southampton, UK. [97]Ashford and St Peter's Hospital, Chertsey, UK. [98]Derriford Hospital, Plymouth, UK. [99]East Surrey Hospital, Redhill, UK. [100]Poole Hospital, Poole, UK. [101]Royal Alexandra Hospital, Paisley, UK. [102]St James's University Hospital and Leeds General Infirmary, Leeds, UK. [103]Bedford Hospital, Bedford, UK. [104]Southport and Formby District General Hospital, Ormskirk, UK. [105]The Tunbridge Wells Hospital and Maidstone Hospital, Kent, UK. [106]Queen Elizabeth Hospital, Woolwich, London, UK. [107]North Manchester General Hospital, Manchester, UK. [108]Royal Victoria Infirmary, Newcastle Upon Tyne, UK. [109]Hull Royal Infirmary, Hull, UK. [110]Manchester Royal Infirmary, Manchester, UK. [111]Royal Derby Hospital, Derby, UK. [112]Aintree University Hospital, Liverpool, UK. [113]Fairfield General Hospital, Bury, UK. [114]Norfolk and Norwich University Hospital (NNUH), Norwich, UK. [115]Milton Keynes University Hospital, Milton Keynes, UK. [116]Good Hope Hospital, Birmingham, UK. [117]Queen Elizabeth Hospital Gateshead, Gateshead, UK. [118]Royal Bolton Hospital, Bolton, UK. [119]Tameside General Hospital, Ashton Under Lyne, UK. [120]Salford Royal Hospital, Manchester, UK. [121]Great Ormond St Hospital and UCL Great Ormond St Institute of Child Health NIHR Biomedical Research Centre, London, UK. [122]Southmead Hospital, Bristol, UK. [123]William Harvey Hospital, Ashford, UK. [124]Arrowe Park Hospital, Wirral, UK. [125]Royal Hampshire County Hospital, Winchester, UK. [126]Bradford Royal Infirmary, Bradford, UK. [127]Glan Clwyd Hospital, Bodelwyddan, UK. [128]Royal Bournemouth Hospital, Bournemouth, UK. [129]Bristol Royal Infirmary, Bristol, UK. [130]University Hospital North Durham, Darlington, UK. [131]Darlington Memorial Hospital, Darlington, UK. [132]Basildon Hospital, Basildon, UK. [133]University College Hospital, London, UK. [134]Whittington Hospital, London, UK. [135]Western General Hospital, Edinburgh, UK. [136]Ipswich Hospital, Ipswich, UK. [137]Hereford County Hospital, Hereford, UK. [138]Sunderland Royal Hospital, Sunderland, UK. [139]Queens Hospital Burton, Burton-On-Trent, UK. [140]Musgrove Park Hospital, Taunton, UK. [141]The Royal Papworth Hospital, Cambridge, UK. [142]University Hospital Lewisham, London, UK. [143]The Princess Alexandra Hospital, Harlow, UK. [144]University Hospital of Wales, Cardiff, UK. [145]West Middlesex Hospital, Isleworth, UK. [146]Royal Albert Edward Infirmary, Wigan, UK. [147]Stoke Mandeville Hospital, Aylesbury, UK. [148]Royal Lancaster Infirmary, Lancaster, UK. [149]Basingstoke and North Hampshire Hospital, Basingstoke, UK. [150]Worthing Hospital, Worthing, UK. [151]St Richard's Hospital, Chichester, UK. [152]The Alexandra Hospital, Redditch and Worcester Royal Hospital, Worcester, UK. [153]Royal Cornwall Hospital, Truro, UK. [154]Watford General Hospital, Watford, UK. [155]Macclesfield District General Hospital, Macclesfield, UK. [156]Royal Surrey County Hospital, Guildford, UK. [157]Rotherham General Hospital, Rotherham, UK. [158]Craigavon Area Hospital, Portadown, UK. [159]King's Mill Hospital, Nottingham, UK. [160]Dumfries and Galloway Royal Infirmary, Dumfries, UK. [161]Prince Charles Hospital, Merthyr Tydfil, UK. [162]Ysbyty Gwynedd, Bangor, UK. [163]Royal Preston Hospital, Preston, UK. [164]The Great Western Hospital, Swindon, UK. [165]Lincoln County Hospital, Lincoln, UK. [166]University Hospital of North Tees, Stockton on Tees, UK. [167]Glangwili General Hospital, Camarthen, UK. [168]Southend University Hospital, Westcliff-on-Sea, UK. [169]Lister Hospital, Stevenage, UK. [170]Diana Princess of Wales Hospital, Grimsby, UK. [171]West Suffolk Hospital, Bury St Edmunds, UK. [172]Victoria Hospital, Kirkcaldy, UK. [173]Calderdale Royal Hospital, Halifax, UK. [174]Huddersfield Royal Infirmary, Huddersfield, UK. [175]Dorset County Hospital, Dorchester, UK. [176]Russell's Hall Hospital, Dudley, UK. [177]Royal United Hospital, Bath, UK. [178]St Mary's Hospital, Newport, UK. [179]George Eliot Hospital NHS Trust, Nuneaton, UK. [180]Yeovil Hospital, Yeovil, UK. [181]Forth Valley Royal Hospital, Falkirk, UK. [182]Frimley Park Hospital, Camberley, UK. [183]Chelsea & Westminster NHS Foundation Trust, London, UK. [184]Queen Elizabeth the Queen Mother Hospital, Margate, UK. [185]Royal Brompton Hospital, London, UK. [186]Darent Valley Hospital, Dartford, UK. [187]University Hospital Crosshouse, Kilmarnock, UK. [188]University Hospital Wishaw, Wishaw, UK. [189]University College Dublin, St Vincent's University Hospital, Dublin, Ireland. [190]The Queen Elizabeth Hospital, King's Lynn, UK. [191]Walsall Manor Hospital, Walsall, UK. [192]Princess Royal Hospital, Brighton, UK. [193]Barnsley Hospital, Barnsley, UK. [194]Warrington General Hospital, Warrington, UK. [195]Royal Victoria Hospital, Belfast, UK. [196]Royal Hallamshire Hospital and Northern General Hospital, Sheffield, UK. [197]Harefield Hospital, London, UK. [198]Cumberland Infirmary, Carlisle, UK. [199]Eastbourne District General Hospital, Eastbourne, UK. [200]Conquest Hospital, Saint Leonards-on-Sea, UK. [201]Salisbury District Hospital, Salisbury, UK. [202]Airedale General Hospital, Keighley, UK. [203]Leicester Royal Infirmary, Leicester, UK.

[204]Peterborough City Hospital, Peterborough, UK. [205]Hinchingbrooke Hospital, Huntingdon, UK. [206]Colchester General Hospital, Colchester, UK. [207]Princess Royal Hospital, Telford and Royal Shrewsbury Hospital, Shrewsbury, UK. [208]University Hospital Monklands, Airdrie, UK. [209]Wrexham Maelor Hospital, Wrexham, UK. [210]New Cross Hospital, Wolverhampton, UK. [211]University Hospital Hairmyres, East Kilbride, UK. [212]Warwick Hospital, Warwick, UK. [213]Sandwell General Hospital and City Hospital, Birmingham, UK. [214]Royal Manchester Children's Hospital, Manchester, UK. [215]Gloucestershire Royal Hospital, Gloucester, UK. [216]University Hospitals Coventry & Warwickshire NHS Trust, Coventry, UK. [217]Torbay Hospital, Torquay, UK. [218]Pilgrim Hospital, Lincoln, UK. [219]Prince Philip Hospital, Lianelli, UK. [220]Princess of Wales Hospital, Llantrisant, UK. [221]Northampton General Hospital NHS Trust, Northampton, UK. [222]The Christie NHS Foundation Trust, Manchester, UK. [223]James Paget University Hospital NHS Trust, Great Yarmouth, UK. [224]Birmingham Children's Hospital, Birmingham, UK. [225]Withybush General Hospital, Haverfordwest, UK. [226]Northwick Park Hospital, London, UK. [227]North Devon District Hospital, Barnstaple, UK. [228]Scunthorpe General Hospital, Scunthorpe, UK. [229]Royal Free Hospital, London, UK. [230]Raigmore Hospital, Inverness, UK. [231]West Cumberland Hospital, Whitehaven, UK. [232]Furness General Hospital, Barrow-in-Furness, UK. [233]Liverpool Heart and Chest Hospital, Liverpool, UK. [234]Scarborough General Hospital, Scarborough, UK. [235]Bronglais General Hospital, Aberystwyth, UK. [236]Alder Hey Children's Hospital, Liverpool, UK. [237]Borders General Hospital, Melrose, UK. [238]Leighton Hospital, Crewe, UK. [239]Kent & Canterbury Hospital, Canterbury, UK. [240]Harrogate and District NHS Foundation Trust, Harrogate, UK. [241]The Royal Marsden Hospital, London, UK. [242]Ealing Hospital, Southall, UK. [243]St John's Hospital Livingston, Livingston, UK. [244]Wexham Park Hospital, Slough, UK. [245]Sheffield Children's Hospital, Sheffield, UK. [246]Homerton University Hospital Foundation NHS Trust, London, UK. [247]National Hospital for Neurology and Neurosurgery, London, UK. [248]The Royal Alexandra Children's Hospital, Brighton, UK. [249]Golden Jubilee National Hospital, Clydebank, UK.

**SCOURGE Consortium**

Javier Abellan[250,251], René Acosta-Isaac[252], Jose María Aguado[253,254,255,256], Carlos Aguilar[257], Sergio Aguilera-Albesa[258,259], Abdolah Ahmadi Sabbagh[260], Jorge Alba[261], Sergiu Albu[262,263,264], Karla A. M. Alcalá-Gallardo[265], Julia Alcoba-Florez[266], Sergio Alcolea Batres[267], Holmes Rafael Algarin-Lara[268,269], Virginia Almadana[270], Julia Almeida[271,272,273,274], Berta Almoguera[37,275], María R. Alonso[276], Nuria Alvarez[276], Yady Álvarez-Benítez[268,269], Felipe Álvarez-Navia[277,278], Rodolfo Alvarez-Sala Walther[267], Álvaro Andreu-Bernabeu[255,279], Maria Rosa Antonijoan[280], Eunate Arana-Arri[281,282], Carlos Aranda[283,284], Celso Arango[255,279,285], Carolina Araque[286,287], Nathalia K. Araujo[288], Izabel M. T. Araujo[289], Ana C. Arcanjo[290,291,292], Ana Arnaiz[33,34,35], Francisco Arnalich Fernández[293], María J. Arranz[294], José Ramon Arribas Lopez[293], Maria-Jesus Artiga[295], Yubelly Avello-Malaver[296], Carmen Ayuso[37,275], Ana Margarita Baldión-Elorza[296], Belén Ballina Martín[260], Raúl C. Baptista-Rosas[297,298,299], Andrea Barranco-Díaz[269], María Barreda-Sánchez[300,301], Viviana Barrera-Penagos[296], Moncef Belhassen-Garcia[278,302], Enrique Bernal[300], David Bernal-Bello[303], Joao F. Bezerra[304], Marcos A. C. Bezerra[305], Natalia Blanca-López[306], Rafael Blancas[307], Lucía Boix-Palop[308], Alberto Borobia[309], Elsa Bravo[310], María Brion[311,312], Óscar Brochado-Kith[256,313], Ramón Brugada[312,314,315,316], Matilde Bustos[317], Alfonso Cabello[318], Juan J. Caceres-Agra[319], Esther Calbo[320], Enrique J. Calderón[321,322,323], Shirley Camacho[324], Marcela C. Campos[290], Yolanda Cañadas[284], Cristina Carbonell[277,278], Servando Cardona-Huerta[36], Antonio Augusto F. Carioca[325], Maria Sanchez Carpintero[283,284], Carlos Carpio Segura[267], Thássia M. T. Carratto[326], José Antonio Carrillo-Avila[327], Maria C. C. Carvalho[328], Carlos Casasnovas[37,329,330], Luis Castano[37,281,331,332,333], Carlos F. Castaño[283,284], Jose E. Castelao[334], Aranzazu Castellano Candalija[335], María A. Castillo[324], Francisco C. Ceballos[313], Jessica G. Chaux[287], Walter G. Chaves-Santiago[287,336], Sylena Chiquillo-Gómez[268,269], Marco A. Cid-Lopez[265], Oscar Cienfuegos-Jimenez[36], Rosa Conde-Vicente[337], M. Lourdes Cordero-Lorenzana[338], Dolores Corella[339,340], Almudena Corrales[30,31], Jose L. Cortes-Sanchez[36,341], Marta Corton[37,275], Tatiana X. Costa[342], Raquel Cruz[37,44,45,343], Marina S. Cruz[344], Luisa Cuesta[344], Gabriela C. R. Cunha[345], Gabriela V. da Silva[289], David Dalmau[320,346], Raquel C. S. Dantas-Komatsu[288], M. Teresa Darnaude[347], Raimundo de Andrés[348], Jéssica N. G. de Araújo[349], Carmen de Juan[350], Juan De la Cruz Troca[322,351,352], Carmen de la Horra[323], Ana B. de la Hoz[281], Alba De Martino-Rodríguez[353,354], Julianna Lys de Sousa Alves Neri[355], Victor del Campo-Pérez[356], Juan Delgado-Cuesta[357], Covadonga M. Díaz-Caneja[255,279,285], Anderson Díaz-Pérez[269], Aranzazu Diaz de Bustamante[347], Beatriz Dietl[320], Silvia Diz-de Almeida[37,44], Manoella do Monte Alves[358,359], Elena Domínguez-Garrido[360], Katiusse A. dos Santos[328], Alice M. Duarte[289], Jose Echave-Sustaeta[361], Rocío Eiros[362], César O. Enciso-Olivera[286,287], Gabriela Escudero[363], Pedro Pablo España[364], Gladys Mercedes Estigarribia Sanabria[365], María Carmen Fariñas[33,34,35], Marianne R. Fernandes[366,367], Ramón Fernández[33,368], Lidia Fernandez-Caballero[37,275], Ana Fernández-Cruz[369], María J. Fernandez-Nestosa[370], Uxía Fernández-Robelo[371], Amanda Fernández-Rodríguez[256,313], Marta Fernández-Sampedro[33,34,35], Ruth Fernández-Sánchez[37,275], Tania Fernández-Villa[372], Silvia Fernández Ferrero[260], Yolanda Fernández Martínez[260], Carmen Fernández Capitán[335], Patricia Flores-Pérez[373], Vicente Friaza[322,323], Lácides Fuenmayor-Hernández[269], Marta Fuertes Núñez[260], Victoria Fumadó[375], Ignacio Gadea[375], Lidia Gagliardi[283,284], Manuela Gago-Domínguez[45,46], Natalia Gallego[38], Cristina Galoppo[376], Inés García[37,275], Mercedes García[283,284], Leticia García[283,284], Carlos Garcia-Cerrada[37,250,251], Aitor García-de-Vicuña[281,331], Josefina García-García[300], Irene García-García[309], Carmen García-Ibarbia[33,34,35], Andrés A. García-Montero[377], Ana García-Soidán[300], Elisa García-Vázquez[300], María Carmen García Torrejón[251,379], Emiliano Garza-Frias[36], Angela Gentile[376], Belén Gil-Fournier[380], Javier Gómez-Arrue[353,354], Mario Gómez-Duque[287,336], Luis Gómez Carrera[267], María Gómez García[343], Ángela Gómez Sacristán[381], Anna González-Neira[276], Javier González-Peñas[255,279,285], Manuel Gonzalez-Sagrado[337], Beatriz González Álvarez[353,354], Fernan Gonzalez Bernaldo de Quirós[382], Hugo Gonzalo Benito[383], Oscar Gorgojo-Galindo[384], Miguel Górgolas[318], Florencia Guaragna[376], Genilson P. Guegel[385], Beatriz Guillen-Guio[30], Encarna Guillen-Navarro[300,386,387,388], Pablo Guisado-Vasco[361], Juan F. Gutiérrez-Bautista[389],

Luz D. Gutierrez-Castañeda[287,390], Sarah Heili-Frades[391], Estefania Hernandez[392], Luis D. Hernandez-Ortega[299,393], Guillermo Hernández-Pérez[277], Rebeca Hernández-Vaquero[394], Cristina Hernández Moro[260], Belen Herraez[276], M. Teresa Herranz[300], María Herrera[283,284], María José Herrero[395,396], Antonio Herrero-Gonzalez[397], Juan P. Horcajada[256,263,398,399,400], Natale Imaz-Ayo[281], Maider Intxausti-Urrutibeaskoa[401], María Íñiguez[402], Rafael H. Jacomo[403], Rubén Jara[300], Perez Maria Jazmin[376], Ángel Jiménez[283,284], Pilar Jiménez[389], Ignacio Jiménez-Alfaro[404], María A. Jimenez-Sousa[256,313], Iolanda Jordan[322,405,406], Rocío Laguna-Goya[407,408], Daniel Laorden[267], María Lasa-Lazaro[407,408], María Claudia Lattig[324,409], Ailen Lauriente[376], Anabel Liger Borja[410], Lucía Llanos[411], Amparo López-Bernús[277,278], Esther Lopez-Garcia[322,351,352,412], Rosario Lopez-Rodriguez[37,275], Miguel A. López-Ruz[413,414,415], Eduardo López Granados[37,416,417], Leonardo Lorente[418], José E. Lozano[419], María Lozano-Espinosa[410], Andre D. Luchessi[420], Ignacio Mahillo[31,421,422], Esther Mancebo[407,408], Carmen Mar[364], Cristina Marcelo Calvo[335], Miguel Marcos[277,278], Alba Marcos-Delgado[423], Alicia Marín Candon[309], Pablo Mariscal Aguilar[267], María M. Martín[424], María Dolores Martín[425], Vicente Martín[322,423], Marta Martin-Fernandez[426], Caridad Martín-López[410], José-Ángel Martín-Oterino[277,278], Laura Martin-Pedraza[306], María Martín-Vicente[313], Amalia Martinez[427], Ricardo Martínez[392], Juan José Martínez[37,330], Silvia Martínez[33,35], Eleno Martínez-Aquino[428], Óscar Martínez-González[307], Iciar Martinez-Lopez[429,430], Oscar Martinez-Nieto[296,409], Pedro Martinez-Paz[383], Angel Martinez-Perez[431], Andrea Martínez-Ramas[37,275], Michel F. Martinez-Resendez[36], Violeta Martínez Robles[260], Laura Marzal[37,275], Juliana F. Mazzeu[432,433,434], Jeane F. P. Medeiros[288], Kelliane A. Medeiros[435,436], Francisco J. Medrano[321,322,323], Xose M. Meijome[437,438], Natalia Mejuto-Montero[439], Ana Méndez-Echevarria[293], Humberto Mendoza Charris[269,310], Eleuterio Merayo Macías[440], Fátima Mercadillo[441], Arieh R. Mercado-Sesma[299,393], Pablo Minguez[37,275], Antonio J. J. Molina[322,423], Elena Molina-Roldán[442], Juan Jose Montoya[392], Vitor M. S. Moraes[326], Patricia Moreira-Escriche[350], Xenia Morelos-Arnedo[269,310], Antonio Moreno-Docón[300], Junior Moreno-Escalante[269], Victor Moreno Cuerda[250,251], Alberto Moreno Fernández[335], Rubén Morilla[323,443], Patricia Muñoz García[31,255,444], Pablo Neira[376], Julian Nevado[37,38,445], Israel Nieto-Gañán[378], Joana F. R. Nunes[290], Rocio Nuñez-Torres[276], Antònia Obrador-Hevia[446,447], J. Gonzalo Ocejo-Vinyals[33,35], Virginia Olivar[376], Silviene F. Oliveira[432,433,434,448,449], Lorena Ondo[37,275], Alberto Orfao[271,272,273,274], Luis Ortega[295], Eva Ortega-Paino[33,35], Fernando Ortiz-Flores[33,35], Rocio Ortiz-Lopez[36,451], José A. Oteo[261,402], Harry Pachajoa[452,453], Manuel Pacheco[392], Fredy Javier Pacheco-Miranda[269], Irene Padilla Conejo[260], Sonia Panadero-Fajardo[327], Mara Parellada[255,279,285], Roberto Pariente-Rodríguez[378], Estela Paz-Artal[407,408,454], Germán Peces-Barba[31,455], Miguel S. Pedromingo Kus[456], Celia Perales[375], Patricia Perez[457], César Pérez[458], Gustavo Perez-de-Nanclares[281,331], Felipe Pérez-García[459,460], Patricia Pérez-Matute[402], Alexandra Pérez-Serra[312,314], M. Elena Pérez-Tomás[300], Teresa Perucho[461], Lisbeth A. Pichardo[260], Susana M. T. Pinho[435,462,463], Mel·lina Pinsach-Abuin[312,314], Luz Adriana Pinzón[287,336], Guillermo Pita[276], Francesc Pla-Junca[37,464], Laura Planas-Serra[37,330], Ericka N. Pompa-Mera[465], Gloria L. Porras-Hurtado[392], Aurora Pujol[37,330,466], María Eugenia Quevedo Chávez[268,269], Maria Angeles Quijada[280,467], Inés Quintela[343], Diana Ramirez-Montaño[468], Soraya Ramiro León[380], Pedro Rascado Sedes[394], Delia Recalde[353,354], Emma Recio-Fernández[402], Salvador Resino[256,313], Adriana P. Ribeiro[435,436,463], Carlos S. Rivadeneira-Chamorro[287], Diana Roa-Agudelo[296], Montserrat Robelo Pardo[394], Marilyn Johanna Rodriguez[287], Fernando Rodriguez-Artalejo[322,351,352,412], Marena Rodríguez-Ferrer[269], Carlos Rodriguez-Gallego[32,469], José A. Rodriguez-Garcia[260], María A. Rodriguez-Nicolas[389], Antonio Rodriguez-Nicolas[317], Agustí Rodriguez-Palmero[330,470], Emilio Rodríguez-Ruiz[45,394], Paula A. Rodriguez-Urrego[296], Belén Rodríguez Maya[250], German Ezequiel Rodriguez Novoa[376], Federico Rojo[274,471], Andrea Romero-Coronado[269], Filomeno Rondón García[260], Lidia S. Rosa[472], Antonio Rosales-Castillo[473], Cladelis Rubio[474,475], María Rubio Olivera[283,284], Montserrat Ruiz[37,330], Francisco Ruiz-Cabello[389,414,476], Eva Ruiz-Casares[461], Juan J. Ruiz-Cubillan[33,35], Javier Ruiz-Hornillos[284,477,478], Pablo Ryan[479,480,481], Hector D. Salamanca[286,287], Lorena Salazar-García[324], Giorgina Gabriela Salgueiro Origlia[335], Pedro-Luis Sánchez[278,362], Clara Sánchez-Pablo[362], Olga Sánchez-Pernaute[482], Antonio J. Sánchez López[483], María Concepción Sánchez Prados[267], Javier Sánchez Real[260], Jorge Sánchez Redondo[250,484], Cristina Sancho-Sainz[401], Anna Sangil[308], Arnoldo Santos[458], Ney P. C. Santos[366], Agatha Schlüter[37,330], Sonia Segovia[464,485,486], Alex Serra-Llovich[346], Fernando Sevil Puras[257], Marta Sevilla Porras[37,38], Miguel A. Sicolo[487,488], Vivian N. Silbiger[420], Nayara S. Silva[290], Fabiola T. C. Silva[290], Cristina Silván Fuentes[7], Jordi Solé-Violán[31,489,490], José Manuel Soria[431], Jose V. Sorlí[339,340], Renata R. Sousa[432], Juan Carlos Souto[252], Karla S. C. Souza[328], Vanessa S. Souza[345], John J. Sprockel[287,336], José Javier Suárez-Rama[343], David A. Suarez-Zamora[296], Xiana Taboada-Fraga[439], Eduardo Tamayo[384,491], Alvaro Tamayo-Velasco[492], Juan Carlos Taracido-Fernandez[397], Nathali A. C. Tavares[493], Carlos Tellería[353,354], Jair Antonio Tenorio Castaño[37,38,445], Alejandro Teper[376], Juan Torres-Macho[494], Lilian Torres-Tobar[287], Ronald P. Torres Gutiérrez[456], Jesús Troya[479], Miguel Urioste[441], Juan Valencia-Ramos[495], Agustín Valido[270,496], Juan Pablo Vargas Gallo[497,498], Belén Varón[499], Romero H. T. Vasconcelos[493], Tomas Vega[500], Santiago Velasco-Quirce[501], Valentina Vélez-Santamaría[329,330], Virginia Víctor[283,284], Julia Vidán Estévez[260], Miriam Vieitez-Santiago[33,35], Carlos Vilches[502], Lavinia Villalobos[260], Felipe Villar[455], Judit Villar-Garcia[503,504,505], Cristina Villaverde[37,275], Pablo Villoslada-Blanco[402], Ana Virseda-Berdices[313], Zuleima Yáñez[269], Antonio Zapatero-Gaviria[506], Ruth Zarate[507], Sandra Zazo[471], Miguel López de Heredia[37], Ingrid Mendes[37], Rocío Moreno[37], Esther Sande[37,44,45], Carlos Flores[29,30,31,32], José A. Riancho[33,34,35], Augusto Rojas-Martinez[36], Pablo Lapunzina[37,38,445] & Angel Carracedo[37,44,45,46,343]

[250]Hospital Universitario Mostoles, Medicina Interna, Madrid, Spain. [251]Universidad Francisco de Vitoria, Madrid, Spain. [252]Haemostasis and Thrombosis Unit, Hospital de la Santa Creu i Snt Pau, IIB Sant Pau, Barcelona, Spain. [253]Unit of Infectious Diseases, Hospital Universitario 12 de Octubre, Instituto de Investigación Sanitaria Hospital 12 de Octubre (imas12), Madrid, Spain. [254]Spanish Network for Research in Infectious Diseases (REIPI RD16/0016/0002), Instituto de Salud Carlos III, Madrid, Spain. [255]School of Medicine, Universidad Complutense, Madrid, Spain. [256]Centro de Investigación Biomédica en Red de Enfermedades Infecciosas

(CIBERINFEC), Instituto de Salud Carlos III, Madrid, Spain. [257]Hospital General Santa Bárbara de Soria, Soria, Spain. [258]Pediatric Neurology Unit, Department of Pediatrics, Navarra Health Service Hospital, Pamplona, Spain. [259]Navarra Health Service, NavarraBioMed Research Group, Pamplona, Spain. [260]Complejo Asistencial Universitario de León, León, Spain. [261]Infectious Diseases Department, Hospital Universitario San Pedro, Logroño, Spain. [262]Fundación Institut Guttmann, Institut Universitari de Neurorehabilitació adscrit a la UAB, Hospital de Neurorehabilitació, Barcelona, Spain. [263]Universitat Autònoma de Barcelona (UAB), Barcelona, Spain. [264]Fundació Institut d'Investigació en Ciències de la Salut Germans Trias i Pujol, Barcelona, Spain. [265]Hospital General de Occidente, Guadalajara, Mexico. [266]Microbiology Unit, Hospital Universitario N.S. de Candelaria, Santa Cruz de Tenerife, Spain. [267]Servicio de Neumología, Hospital Universitario La Paz-IDIPAZ, Madrid, Spain. [268]Camino Universitario Adelita de Char, Mired IPS, Barranquilla, Colombia. [269]Facultad de Ciencias de la Salud, Universidad Simón Bolívar, Barranquilla, Colombia. [270]Neumología, Hospital Universitario Virgen Macarena, Seville, Spain. [271]Departamento de Medicina, Universidad de Salamanca, Salamanca, Spain. [272]Centro de Investigación del Cáncer (IBMCC), Universidad de Salamanca, CSIC, Salamanca, Spain. [273]Biomedical Research Institute of Salamanca (IBSAL), Salamanca, Spain. [274]Center for Biomedical Network Research on Cancer (CIBERONC), Instituto de Salud Carlos III, Madrid, Spain. [275]Department of Genetics & Genomics, Instituto de Investigación Sanitaria–Fundación Jiménez Díaz University Hospital— Universidad Autónoma de Madrid (IIS-FJD, UAM), Madrid, Spain. [276]Human Genotyping— CEGEN Unit, Spanish National Cancer Research Centre, Madrid, Spain. [277]Servicio de Medicina Interna, Hospital Universitario de Salamanca-IBSAL, Salamanca, Spain. [278]Universidad de Salamanca, Salamanca, Spain. [279]Department of Child and Adolescent Psychiatry, Institute of Psychiatry and Mental Health, Hospital General Universitario Gregorio Marañón (IiSGM), Madrid, Spain. [280]Clinical Pharmacology Service, Hospital de la Santa Creu i Sant Pau, IIB Sant Pau, Barcelona, Spain. [281]Biocruces Bizkai HRI, Barakaldo, Spain. [282]Cruces University Hospital, Osakidetza, Barakaldo, Spain. [283]Hospital Infanta Elena, Madrid, Spain. [284]Instituto de Investigación Sanitaria–Fundación Jiménez Díaz University Hospital, Universidad Autónoma de Madrid (IIS-FJD, UAM), Madrid, Spain. [285]Centre for Biomedical Network Research on Mental Health (CIBERSAM), Instituto de Salud Carlos III, Madrid, Spain. [286]Fundación Hospital Infantil Universitario de San José, Bogotá, Colombia. [287]Fundación Universitaria de Ciencias de la Salud, Bogotá, Colombia. [288]Programa de Pós-graduação em Ciências da Saúde, Universidade Federal do Rio Grande do Norte, Natal, Brazil. [289]Departamento de Medicina Clínica, Universidade Federal do Rio Grande do Norte, Natal, Brazil. [290]Departamento de Genética e Morfologia, Instituto de Ciências Biológicas, Universidade de Brasília, Brasilia, Brazil. [291]Colégio Marista de Brasília, Brasilia, Brazil. [292]Associação Brasileira de Educação e Cultura, Londrina, Brazil. [293]Servicio de Medicina Interna, Hospital Universitario La Paz-IDIPAZ, Madrid, Spain. [294]Fundació Docència i Recerca Mutua Terrassa, Barcelona, Spain. [295]Spanish National Cancer Research Center, CNIO Biobank, Madrid, Spain. [296]Departamento Patologia y Laboratorios, Fundación Santa Fe de Bogota, Bogotá, Colombia. [297]Hospital General de Occidente, Zapopan, Mexico. [298]Centro Universitario de Tonalá, Universidad de Guadalajara, Tonalá, Mexico. [299]Centro de Investigación Multidisciplinario en Salud, Universidad de Guadalajara, Guadalajara, Mexico. [300]Instituto Murciano de Investigación Biosanitaria (IMIB-Arrixaca), Murcia, Spain. [301]Universidad Católica San Antonio de Murcia (UCAM), Murcia, Spain. [302]Servicio de Medicina Interna-Unidad de Enfermedades Infecciosas, Hospital Universitario de Salamanca-IBSAL, Salamanca, Spain. [303]Department of Internal Medicine, Hospital Universitario de Fuenlabrada, Madrid, Spain. [304]Laboratorio de Vigilancia Molecular Aplicada, Escola Tecnica de Saúde, Pará, Brazil. [305]Genetics Postgraduate Program, Federal University of Pernambuco, Recife, Brazil. [306]Servicio de Alergia, Hospital Universitario Infanta Leonor, Madrid, Spain. [307]Servicio de Medicina Intensiva, Hospital Universitario del Tajo, Toledo, Spain. [308]Hospital Universitario Mutua Terrassa, Barcelona, Spain. [309]Servicio de Farmacología, Hospital Universitario La Paz-IDIPAZ, Madrid, Spain. [310]Alcaldía de Barranquilla, Secretaría de Salud, Barranquilla, Colombia. [311]Xenética Cardiovascular, Instituto de Investigación Sanitaria de Santiago (IDIS), Santiago de Compostela, Spain. [312]Centre for Biomedical Network Research on Cardiovascular Diseases (CIBERCV), Instituto de Salud Carlos III, Madrid, Spain. [313]Unidad de Infección Viral e Inmunidad, Centro Nacional de Microbiología (CNM), Instituto de Salud Carlos III (ISCIII), Madrid, Spain. [314]Cardiovascular Genetics Center, Institut d'Investigació Biomèdica Girona (IDIBGI), Girona, Spain. [315]Medical Science Department, School of Medicine, University of Girona, Girona, Spain. [316]Cardiology Service, Hospital Josep Trueta, Girona, Spain. [317]Institute of Biomedicine of Seville (IBiS), Consejo Superior de Investigaciones Científicas (CSIC), University of Seville, Virgen del Rocio University Hospital, Seville, Spain. [318]Division of Infectious Diseases, Instituto de Investigación Sanitaria–Fundación Jiménez Díaz University Hospital, Universidad Autónoma de Madrid (IIS-FJD, UAM), Madrid, Spain. [319]Intensive Care Unit, Hospital Universitario Insular de Gran Canaria, Las Palmas de Gran Canaria, Spain. [320]Hospital Universitario Mutua Terrassa, Terrassa, Spain. [321]Departemento de Medicina, Hospital Universitario Virgen del Rocío, Universidad de Sevilla, Seville, Spain. [322]Centre for Biomedical Network Research on Epidemiology and Public Health (CIBERESP), Instituto de Salud Carlos III, Madrid, Spain. [323]Instituto de Biomedicina de Sevilla, Seville, Spain. [324]Facultad de Ciencias, Universidad de los Andes, Bogotá, Colombia. [325]Department of Nutrition, University of Fortaleza (UNIFOR), Fortaleza, Brazil. [326]Departamento de Química, Faculdade de Filosofia, Ciências e Letras de Ribeirão Preto, Universidade de São Paulo, São Paulo, Brazil. [327]Andalusian Public Health System Biobank, Granada, Spain. [328]Programa de Pós-Graduação em Ciências Farmacêuticas, Universidade Federal do Rio Grande do Norte, Natal, Brazil. [329]Neuromuscular Unit, Neurology Department, Hospital Universitari de Bellvitge, L'Hospitalet de Llobregat, Barcelona, Spain. [330]Bellvitge Biomedical Research Institute (IDIBELL), Neurometabolic Diseases Laboratory, L'Hospitalet de Llobregat, Barcelona, Spain. [331]Osakidetza, Cruces University Hospital, Barakaldo, Spain. [332]Centre for Biomedical Network Research on Diabetes and Metabolic Associated Diseases (CIBERDEM), Instituto de Salud Carlos III, Madrid, Spain. [333]University of Pais Vasco, UPV/EHU, Bizkaia, Spain. [334]Oncology and Genetics Unit, Instituto de Investigacion Sanitaria Galicia Sur, Xerencia de Xestion Integrada de Vigo-Servizo Galego de Saúde, Vigo, Spain. [335]Hospital Universitario La Paz, Hospital Carlos III, Madrid, Spain. [336]Hospital de San José, Sociedad de Cirugía de Bogota, Bogotá, Colombia. [337]Hospital Universitario Río Hortega, Valladolid, Spain. [338]Servicio de Medicina Intensiva, Complejo Hospitalario Universitario de A Coruña (CHUAC), Sistema Galego de Saúde (SERGAS), A Coruña, Spain. [339]Preventive Medicine Department, Valencia University, Valencia, Spain. [340]Centre for

Biomedical Network Research on Physiopatology of Obesity and Nutrition (CIBEROBN), Instituto de Salud Carlos III, Madrid, Spain. [341]Department of Microgravity and Translational Regenerative Medicine, Otto von Guericke University, Magdeburg, Germany. [342]Maternidade Escola Janário Cicco, Natal, Brazil. [343]Centro Nacional de Genotipado (CEGEN), Universidade de Santiago de Compostela, Santiago de Compostela, Spain. [344]Institute of Psychiatry and Mental Health, Hospital General Universitario Gregorio Marañón (IiSGM), Madrid, Spain. [345]Programa de Pós Graduação em Ciências da Saúde, Faculdade de Medicina, Universidade de Brasília, Brasilia, Brazil. [346]Fundació Docència I Recerca Mutua Terrassa, Terrassa, Spain. [347]Unidad de Genética, Hospital Universitario Mostoles, Madrid, Spain. [348]Internal Medicine Department, Instituto de Investigación Sanitaria–Fundación Jiménez Díaz University Hospital, Universidad Autónoma de Madrid (IIS-FJD, UAM), Madrid, Spain. [349]Universidade Federal do Rio Grande do Norte, Pós-graduação, Rede de Biotecnologia do Nordeste (Renorbio), Natal, Brazil. [350]Servicio de Medicina Interna, Hospital Universitario Severo Ochoa, Madrid, Spain. [351]Department of Preventive Medicine and Public Health, School of Medicine, Universidad Autónoma de Madrid, Madrid, Spain. [352]IdiPaz (Instituto de Investigación Sanitaria Hospital Universitario La Paz), Madrid, Spain. [353]Instituto Aragonés de Ciencias de la Salud (IACS), Zaragoza, Spain. [354]Instituto Investigación Sanitaria Aragón (IIS-Aragon), Zaragoza, Spain. [355]Programa de Pós Graduação em Nutrição, Universidade Federal do Rio Grande do Norte, Natal, Brazil. [356]Preventive Medicine Department, Instituto de Investigacion Sanitaria Galicia Sur, Xerencia de Xestion Integrada de Vigo-Servizo Galego de Saúde, Vigo, Spain. [357]Servicio de Medicina Interna, Hospital Universitario Virgen del Rocío, Seville, Spain. [358]Departamento de Infectologia, Universidade Federal do Rio Grande do Norte, Natal, Brazil. [359]Hospital de Doenças Infecciosas Giselda Trigueiro, Natal, Brazil. [360]Unidad Diagnóstico Molecular, Fundación Rioja Salud, La Rioja, Spain. [361]Hospital Universitario Quironsalud Madrid, Madrid, Spain. [362]Servicio de Cardiología, Hospital Universitario de Salamanca-IBSAL, Salamanca, Spain. [363]Servicio de Medicina Interna, Hospital Universitario Puerta de Hierro, Majadahonda, Spain. [364]Biocruces Bizkaia Health Research Institute, Galdakao University Hospital, Osakidetza, Bizkaia, Spain. [365]Instituto Regional de Investigación en Salud-Universidad Nacional de Caaguazú, Caaguazú, Paraguay. [366]Núcleo de Pesquisas em Oncologia, Universidade Federal do Pará, Belém, Brazil. [367]Departamento de Ensino e Pesquisa, Hospital Ophir Loyola, Belém, Brazil. [368]Fundación Asilo San Jose, Santander, Spain. [369]Unidad de Enfermedades Infecciosas, Servicio de Medicina Interna, Hospital Universitario Puerta de Hierro, Instituto de Investigación Sanitaria Puerta de Hierro—Segovia de Arana, Madrid, Spain. [370]Universidad Nacional de Asunción, Facultad de Politécnica, Paraguay. [371]Urgencias Hospitalarias, Complejo Hospitalario Universitario de A Coruña (CHUAC), Sistema Galego de Saúde (SERGAS), A Coruña, Spain. [372]Grupo de Investigación en Interacciones Gen-Ambiente y Salud (GIIGAS), Instituto de Biomedicina (IBIOMED), Universidad de León, León, Spain. [373]Pediatrics Department, Hospital Universitario Niño Jesús, Madrid, Spain. [374]Unitat de Malalties Infeccioses i Importades, Servei de Pediatría, Infectious and Imported Diseases, Pediatric Unit, Hospital Universitari Sant Joan de Déu, Barcelona, Spain. [375]Microbiology Department, Instituto de Investigación Sanitaria–Fundación Jiménez Díaz University Hospita, Universidad Autónoma de Madrid (IIS-FJD, UAM), Madrid, Spain. [376]Hospital de Niños Ricardo Gutierrez, Buenos Aires, Argentina. [377]University of Salamanca, Biomedical Research Institute of Salamanca (IBSAL), Salamanca, Spain. [378]Department of Immunology, IRYCIS, Hospital Universitario Ramón y Cajal, Madrid, Spain. [379]Servicio de Medicina Intensiva, Hospital Infanta Elena, Madrid, Spain. [380]Servicio de Genética, Hospital Universitario de Getafe, Madrid, Spain. [381]Pneumology Department, Hospital General Universitario Gregorio Marañón (iiSGM), Madrid, Spain. [382]Ministerio de Salud Ciudad de Buenos Aires, Buenos Aires, Argentina. [383]Unidad de Apoyo a la Investigación, Hospital Clinico Universitario de Valladolid, Valladolid, Spain. [384]Departamento de Cirugía, Universidad de Valladolid, Valladolid, Spain. [385]Secretaria Municipal de Saude de Apodi, Natal, Brazil. [386]Sección Genética Médica, Servicio de Pediatría, Hospital Clínico Universitario Virgen de la Arrixaca, Servicio Murciano de Salud, Murcia, Spain. [387]Departamento Cirugía, Pediatría, Obstetricia y Ginecología, Facultad de Medicina, Universidad de Murcia (UMU), Murcia, Spain. [388]Grupo Clínico Vinculado, Centre for Biomedical Network Research on Rare Diseases (CIBERER), Instituto de Salud Carlos III, Madrid, Spain. [389]Servicio de Análisis Clínicos e Inmunología, Hospital Universitario Virgen de las Nieves, Granada, Spain. [390]Hospital Universitario Centro Dermatológico Federico Lleras Acosta, Bogotá, Colombia. [391]Intermediate Respiratory Care Unit, Department of Pneumology, Instituto de Investigación Sanitaria–Fundación Jiménez Díaz University Hospital, Universidad Autónoma de Madrid (IIS-FJD, UAM), Madrid, Spain. [392]Clinica Comfamiliar Risaralda, Pereira, Colombia. [393]Centro Universitario de Tonalá, Universidad de Guadalajara, Guadalajara, Mexico. [394]Unidad de Cuidados Intensivos, Hospital Clínico Universitario de Santiago (CHUS), Sistema Galego de Saúde (SERGAS), Santiago de Compostela, Spain. [395]Plataforma de Farmacogenética, IIS La Fe, Valencia, Spain. [396]Departamento de Farmacología, Universidad de Valencia, Valencia, Spain. [397]Data Analysis Department, Instituto de Investigación Sanitaria–Fundación Jiménez Díaz University Hospital, Universidad Autónoma de Madrid (IIS-FJD, UAM), Madrid, Spain. [398]Infectious Diseases Service, Hospital del Mar, Barcelona, Spain. [399]Institut Hospital del Mar d'Investigacions Mèdiques (IMIM), Barcelona, Spain. [400]CEXS-Universitat Pompeu Fabra, Spanish Network for Research in Infectious Diseases (REIPI), Barcelona, Spain. [401]Biocruces Bizkaia Health Research Institute, Basurto University Hospital, Osakidetza, Spain. [402]Infectious Diseases, Microbiota and Metabolism Unit, Center for Biomedical Research of La Rioja (CIBIR), Logroño, Spain. [403]Sabin Medicina Diagnóstica, São Paulo, Brazil. [404]Opthalmology Department, Instituto de Investigación Sanitaria–Fundación Jiménez Díaz University Hospital, Universidad Autónoma de Madrid (IIS-FJD, UAM), Madrid, Spain. [405]Pediatric Critical Care Unit, Hospital Sant Joan de Deu, Barcelona, Spain. [406]Paediatric Intensive Care Unit, Agrupación Hospitalaria Clínic-Sant Joan de Déu, Esplugues de Llobregat, Barcelona, Spain. [407]Department of Immunology, Hospital Universitario 12 de Octubre, Madrid, Spain. [408]Transplant Immunology and Immunodeficiencies Group, Instituto de Investigación Sanitaria Hospital 12 de Octubre (imas12), Madrid, Spain. [409]SIGEN Alianza Universidad de los Andes, Fundación Santa Fe de Bogotá, Bogotá, Colombia. [410]Medicina Intensiva, Hospital General de Segovia, Segovia, Spain. [411]Clinical Trials Unit, Instituto de Investigación Sanitaria–Fundación Jiménez Díaz University Hospital, Universidad Autónoma de Madrid (IIS-FJD, UAM), Madrid, Spain. [412]IMDEA-Food Institute, CEI UAM+CSIC, Madrid, Spain. [413]Servicio de Enfermedades Infecciosas, Hospital Universitario Virgen de las Nieves, Granada, Spain. [414]Instituto de Investigación Biosanitaria de Granada (ibs.GRANADA), Granada, Spain. [415]Departamento de Medicina, Universidad de Granada, Granada, Spain.

[416]Servicio de Inmunología, Hospital Universitario La Paz-IDIPAZ, Madrid, Spain. [417]Lymphocyte Pathophysiology in Immunodeficiencies Group, La Paz Institute for Health Research (IdiPAZ), Madrid, Spain. [418]Intensive Care Unit, Hospital Universitario de Canarias, La Laguna, Spain. [419]Dirección General de Salud Pública, Consejería de Sanidad, Junta de Castilla y León, Valladolid, Spain. [420]Departamento de Analises Clinicas e Toxicologicas, Universidade Federal do Rio Grande do Norte, Natal, Brazil. [421]Epidemiology, Fundación Jiménez Díaz, Madrid, Spain. [422]Department of Medicine, Universidad Autónoma de Madrid, Madrid, Spain. [423]Instituto de Biomedicina (IBIOMED), Universidad de León, León, Spain. [424]Intensive Care Unit, Hospital Universitario N. S. de Candelaria, Santa Cruz de Tenerife, Spain. [425]Preventive Medicine Department, Instituto de Investigación Sanitaria–Fundación Jiménez Díaz University Hospital, Universidad Autónoma de Madrid (IIS-FJD, UAM), Madrid, Spain. [426]Departamento de Medicina, Universidad de Valladolid, Valladolid, Spain. [427]Servicio de Medicina Intensiva, Hospital Universitario Infanta Leonor, Madrid, Spain. [428]Servicio de Medicina Interna, Sanatorio Franchin, Buenos Aires, Argentina. [429]Unidad de Genética y Genómica Islas Baleares, Hospital Universitario Son Espases, Islas Baleares, Spain. [430]Unidad de Diagnóstico Molecular y Genética Clínica, Hospital Universitario Son Espases, Islas Baleares, Spain. [431]Genomics of Complex Diseases Unit, Research Institute of Hospital de la Santa Creu i Sant Pau, IIB Sant Pau, Barcelona, Spain. [432]Faculdade de Medicina, Universidade de Brasília, Brasília, Brazil. [433]Programa de Pós-Graduação em Ciências Médicas, Universidade de Brasília, Brasília, Brazil. [434]Programa de Pós-Graduação em Ciências da Saúde, Universidade de Brasília, Brasília, Brazil. [435]Hospital das Forças Armadas, Brasília, Brazil. [436]Exército Brasileiro, Cruzeiro, Brazil. [437]Hospital El Bierzo, Gerencia de Asistencia Sanitaria del Bierzo (GASBI), Gerencia Regional de Salud (SACYL), Ponferrada, Spain. [438]Grupo INVESTEN, Instituto de Salud Carlos III, Madrid, Spain. [439]Unidad de Cuidados Intensivos, Complejo Universitario de A Coruña (CHUAC), Sistema Galego de Saúde (SERGAS), A Coruña, Spain. [440]Unidad Cuidados Intensivos, Hospital El Bierzo, León, Spain. [441]Familial Cancer Clinical Unit, Spanish National Cancer Research Centre, Madrid, Spain. [442]Instituto de Investigación Sanitaria San Carlos (IdISSC), Hospital Clínico San Carlos (HCSC), Madrid, Spain. [443]Departamento de Enfermería, Universidad de Sevilla, Seville, Spain. [444]Hospital General Universitario Gregorio Marañón (IiSGM), Madrid, Spain. [445]ERN-ITHACA-European Reference Network on Rare Congenital Malformations and Rare Intellectual Disability, Brussels, Belgium. [446]Unidad de Genética y Genómica Islas Baleares, Unidad de Diagnóstico Molecular y Genética Clínica, Hospital Universitario Son Espases, Islas Baleares, Spain. [447]Instituto de Investigación Sanitaria Islas Baleares (IdISBa), Islas Baleares, Spain. [448]Programa de Pós-Graduação em Biologia Animal, Universidade de Brasília, Brasília, Brazil. [449]Programa de Pós-Graduação Profissional em Ensino de Biologia, Universidade de Brasília, Brasília, Brazil. [450]Anatomía Patológica, Instituto de Investigación Sanitaria San Carlos (IdISSC), Hospital Clínico San Carlos (HCSC), Madrid, Spain. [451]Tecnológico de Monterrey, Monterrey, Mexico. [452]Centro de Investigación en Anomalías Congénitas y Enfermedades Raras (CIACER), Universidad Icesi, Cali, Colombia. [453]Departamento de Genetica, Fundación Valle del Lili, Cali, Colombia. [454]Department of Immunology, Ophthalmology and ENT, Universidad Complutense de Madrid, Madrid, Spain. [455]Department of Neumology, Instituto de Investigación Sanitaria–Fundación Jiménez Díaz University Hospital, Universidad Autónoma de Madrid (IIS-FJD, UAM), Madrid, Spain. [456]Hospital Nuestra Señora de Sonsoles, Ávila, Spain. [457]Inditex, A Coruña, Spain. [458]Intensive Care Department, Instituto de Investigación Sanitaria–Fundación Jiménez Díaz University Hospital, Universidad Autónoma de Madrid (IIS-FJD, UAM), Madrid, Spain. [459]Servicio de Microbiología Clínica, Hospital Universitario Príncipe de Asturias, Madrid, Spain. [460]Departamento de Biomedicina y Biotecnología, Facultad de Medicina y Ciencias de la Salud, Universidad de Alcalá de Henares, Madrid, Spain. [461]GENYCA, Madrid, Spain. [462]Marinha do Brasil, Brasil, Brazil. [463]Universidade de Brasília, Brasília, Brazil. [464]Neuromuscular Diseases Unit, Department of Neurology, Hospital de la Santa Creu i Sant Pau, Universitat Autònoma de Barcelona, Barcelona, Spain. [465]Unidad de Investigación Médica en Enfermedades Infecciosas y Parasitarias, Instituto Mexicano del Seguro Social (IMSS), Centro Médico Nacional Siglo XXI, Mexico City, Mexico. [466]Catalan Institution of Research and Advanced Studies (ICREA), Barcelona, Spain. [467]Drug Research Centre, Institut d'Investigació Biomèdica Sant Pau, IIB-Sant Pau, Barcelona, Spain. [468]Departamento de Genetica, Clinica imbanaco, Cali, Colombia. [469]Department of Immunology, Hospital Universitario de Gran Canaria Dr Negrín, Las Palmas de Gran Canaria, Spain. [470]Pediatrics Department, Universitary Hospital Germans Trias i Pujol, Badalona, Spain. [471]Department of Pathology, Biobank, Instituto de Investigación Sanitaria–Fundación Jiménez Díaz University Hospital, Universidad Autónoma de Madrid (IIS-FJD, UAM), Madrid, Spain. [472]Faculdade de Ciências da Saúde, Universidade de Brasília, Brasília, Brazil. [473]Servicio de Medicina Interna, Hospital Universitario Virgen de las Nieves, Granada, Spain. [474]Grupo de Ciencias Básicas en Salud (CBS), Fundación Universitaria de Ciencias de la Salud, Bogotá, Colombia. [475]Sociedad de Cirugía de Bogotá, Hospital de San José, Bogotá, Colombia. [476]Departamento Bioquímica, Biología Molecular e Inmunología III, Universidad de Granada, Granada, Spain. [477]Allergy Unit, Hospital Infanta Elena, Madrid, Spain. [478]Faculty of Medicine, Universidad Francisco de Vitoria, Madrid, Spain. [479]Hospital Universitario Infanta Leonor, Madrid, Spain. [480]Complutense University of Madrid, Madrid, Spain. [481]Gregorio Marañón Health Research Institute (IiSGM), Madrid, Spain. [482]Reumathology Service, Instituto de Investigación Sanitaria–Fundación Jiménez Díaz University Hospital, Universidad Autónoma de Madrid (IIS-FJD, UAM), Madrid, Spain. [483]Biobank, Puerta de Hierro-Segovia de Arana Health Research Institute, Madrid, Spain. [484]Universidad Rey Juan Carlos, Madrid, Spain. [485]The John Walton Muscular Dystrophy Research Centre, Newcastle University and Newcastle Hospitals NHS Foundation Trust, Newcastle upon Tyne, UK. [486]Neuromuscular Unit, Neuropediatrics Department, Institut de Recerca Sant Joan de Déu, Hospital Sant Joan de Déu, Barcelona, Spain. [487]Casa de Saúde São Lucas, Natal, Brazil. [488]Hospital Rio Grande, Natal, Brazil. [489]Intensive Care Unit, Hospital Universitario de Gran Canaria Dr Negrín, Las Palmas de Gran Canaria, Spain. [490]Universidad Fernando Pessoa Canarias, Las Palmas de Gran Canaria, Spain. [491]Servicio de Anestesiología y Reanimación, Hospital Clinico Universitario de Valladolid, Valladolid, Spain. [492]Servicio de Hematologia y Hemoterapia, Hospital Clinico Universitario de Valladolid, Valladolid, Spain. [493]Hospital Universitario Lauro Wanderley, João Pessoa, Brazil. [494]Servicio de Medicina Interna, Hospital Universitario Infanta Leonor, Madrid, Spain. [495]University Hospital of Burgos, Burgos, Spain. [496]Universidad de Sevilla, Seville, Spain. [497]Fundación Santa Fe de Bogota, Instituto de Servicios Medicos de Emergencia y Trauma, Bogotá, Colombia. [498]Universidad de los Andes, Bogotá, Colombia. [499]Quironprevención, A Coruña, Spain. [500]Junta de Castilla y León, Consejería de Sanidad, Valladolid, Spain. [501]Gerencia Atención Primaria de Burgos, Burgos, Spain. [502]Immunogenetics–Histocompatibility group, Servicio de Inmunología, Instituto de Investigación Sanitaria Puerta de Hierro, Segovia de Arana, Madrid, Spain. [503]Department of Infectious Diseases, Hospital del Mar, Barcelona, Spain. [504]IMIM—Hospital del Mar Medical Research Institute, Institut Hospital del Mar d'Investigacions Mediques, Barcelona, Spain. [505]Department of Medicine, Universitat Autònoma de Barcelona, Barcelona, Spain. [506]Consejería de Sanidad, Comunidad de Madrid, Madrid, Spain. [507]Centro para el Desarrollo de la Investigación Científica, Asunción, Paraguay.

## ISARICC Investigators

### Co-Investigators

J. Kenneth Baillie[1,2,3,11], Peter J. M. Openshaw[12,26], Malcolm G. Semple[16,17], Beatrice Alex[508], Petros Andrikopoulos[509,510], Benjamin Bach[508], Wendy S. Barclay[511], Debby Bogaert[18], Meera Chand[512], Kanta Chechi[509,513], Graham S. Cooke[514], Ana da Silva Filipe[515], Thushan de Silva[516], Annemarie B. Docherty[11,517], Gonçalo dos Santos Correia[518,519], Marc-Emmanuel Dumas[509,510,520,521], Jake Dunning[12,522], Tom Fletcher[523], Christopher A. Green[524], William Greenhalf[525], Julian Griffin[509], Rishi K. Gupta[526], Ewen M. Harrison[517], Antonia Y. W. Ho[515,527], Karl Holden[528], Peter W. Horby[21], Samreen Ijaz[529], Say Khoo[530], Paul Klenerman[531,532], Andrew Law[3], Matthew Lewis[518,519], Sonia Liggi[509], Wei Shen Lim[533], Lynn Maslen[518,519], Alexander J. Mentzer[534,535], Laura Merson[536], Alison M. Meynert[517], Shona C. Moore[537], Mahdad Noursadeghi[538], Michael Olanipekun[509,510], Anthonia Osagie[509,510], Massimo Palmarini[515], Carlo Palmieri[539,540], William A. Paxton[537,541], Georgios Pollakis[537,541], Nicholas Price[542,543], Andrew Rambaut[544], David L. Russell[18], Clark D. Russell[18], Vanessa Sancho-Shimizu[545], Caroline Sands[518,519], Janet T. Scott[515,546], Louise Sigfrid[536], Tom Solomon[16,547], Shiranee Sriskandan[514,548], David Stuart[8], Charlotte Summers[19], Olivia V. Swann[549], Zoltan Takats[509,550], Panteleimon Takis[518,519], Richard S. Tedder[551,552,553], A. A. Roger Thompson[554], Emma C. Thomson[515], Ryan S. Thwaites[12], Lance C. W. Turtle[16,555], Maria Zambon[522] & Gail Carson[536]

### Data Analysis Team

Thomas M. Drake[517], Cameron J. Fairfield[517], Stephen R. Knight[517], Kenneth A. Mclean[517], Derek Murphy[517], Lisa Norman[517], Riinu Pius[517] & Catherine A. Shaw[517]

### Data Architecture Team

Marie Connor[556], Jo Dalton[556], Carrol Gamble[556], Michelle Girvan[556], Sophie Halpin[556], Janet Harrison[556], Clare Jackson[556], Laura Marsh[556], Stephanie Roberts[556], Egle Saviciute[556], Sara Clohisey[3], Ross Hendry[3], Susan Knight[557], Eva Lahnsteiner[557], Andrew Law[3], Gary Leeming[558], Lucy Norris[559], James Scott-Brown[508], Sarah Tait[557] & Murray Wham[2]

### Data Analysis and Management Team

James Lee[560], Daniel Plotkin[560] & Seán Keating[11]

### Project Administration Team

Cara Donegan[561] & Rebecca G. Spencer[561]

### Project Management Team

Chloe Donohue[556], Fiona Griffiths[3], Hayley Hardwick[16] & Wilna Oosthuyzen[3]

[508]School of Informatics, University of Edinburgh, Edinburgh, UK. [509]Section of Biomolecular Medicine, Division of Systems Medicine, Department of Metabolism, Digestion and Reproduction, Sir Alexander Fleming Building, Imperial College London, London, UK. [510]Section of Genomic and Environmental Medicine, Respiratory Division, National Heart and Lung Institute, London, UK. [511]Section of Molecular Virology, Imperial College London, London, UK. [512]Antimicrobial Resistance and Hospital Acquired Infection Department, Public Health England, London, UK. [513]Department of Epidemiology and Biostatistics, School of Public Health, Faculty of Medicine, Imperial College London, London, UK. [514]Department of Infectious Disease, Imperial College London, London, UK. [515]MRC-University of Glasgow Centre for Virus Research, Glasgow, UK. [516]The Florey Institute for Host-Pathogen Interactions, Department of Infection, Immunity and Cardiovascular Disease, University of Sheffield, Sheffield, UK. [517]Centre for Medical Informatics, The Usher Institute, University of Edinburgh, Edinburgh, UK. [518]National Phenome Centre, Department of Metabolism, Digestion and Reproduction, Imperial College London, London, UK. [519]Section of Bioanalytical Chemistry, Department of Metabolism, Digestion and Reproduction, Imperial College London, London, UK. [520]European Genomic Institute for Diabetes, CNRS UMR 8199, INSERM UMR 1283, Institut Pasteur de Lille, Lille University Hospital, University of Lille, Lille, France. [521]McGill University and Genome Quebec Innovation Centre, Montréal, Quebec, Canada. [522]National Infection Service, Public Health England, London, UK. [523]Liverpool School of Tropical Medicine, Liverpool, UK. [524]Institute of Microbiology and Infection, University of Birmingham, Birmingham, UK. [525]Department of Molecular and Clinical Cancer Medicine, University of Liverpool, Liverpool, UK. [526]Institute for Global Health, University College London, London, UK. [527]Department of Infectious Diseases, Queen Elizabeth University Hospital, Glasgow, UK. [528]University of Liverpool, Liverpool, UK. [529]Virology Reference Department, National Infection Service, Public Health England, London, UK. [530]Department of Pharmacology, University of Liverpool, Liverpool, UK. [531]Nuffield Department of Medicine, Peter Medawar Building for Pathogen Research, University of Oxford, Oxford, UK. [532]Translational Gastroenterology Unit, Nuffield Department of Medicine, University of Oxford, Oxford, UK. [533]Nottingham University Hospitals NHS Trust, Nottingham, UK. [534]Nuffield Department of Medicine, John Radcliffe Hospital, Oxford, UK. [535]Department of Microbiology/Infectious Diseases, Oxford University Hospitals NHS Foundation Trust, John Radcliffe Hospital, Oxford, UK. [536]ISARIC Global Support Centre, Centre for Tropical Medicine and Global Health, Nuffield Department of Medicine, University of Oxford, Oxford, UK. [537]Institute of Infection, Veterinary and Ecological Sciences, University of Liverpool, Liverpool, UK. [538]Division of Infection and Immunity, University College London, London, UK. [539]Molecular and Clinical Cancer Medicine, Institute of Systems, Molecular and Integrative

Biology, University of Liverpool, Liverpool, UK. [540]Clatterbridge Cancer Centre NHS Foundation Trust, Liverpool, UK. [541]NIHR Health Protection Research Unit in Emerging and Zoonotic Infections, Liverpool, UK. [542]Centre for Clinical Infection and Diagnostics Research, Department of Infectious Diseases, School of Immunology and Microbial Sciences, King's College London, London, UK. [543]Department of Infectious Diseases, Guy's and St Thomas' NHS Foundation Trust, London, UK. [544]Institute of Evolutionary Biology, University of Edinburgh, Edinburgh, UK. [545]Department of Pediatrics and Virology, St Mary's Medical School Building, Imperial College London, London, UK. [546]NHS Greater Glasgow & Clyde, Glasgow, UK. [547]Walton Centre NHS Foundation Trust, Liverpool, UK. [548]MRC Centre for Molecular Bacteriology and Infection, Imperial College London, London, UK. [549]Department of Child Life and Health, University of Edinburgh, Edinburgh, UK. [550]National Phenome Centre, Division of Systems Medicine, Department of Metabolism, Digestion and Reproduction, Imperial College London, London, UK. [551]Blood Borne Virus Unit, Virus Reference Department, National Infection Service, Public Health England, London, UK. [552]Transfusion Microbiology, National Health Service Blood and Transplant, London, UK. [553]Department of Medicine, Imperial College London, London, UK. [554]Department of Infection, Immunity and Cardiovascular Disease, University of Sheffield, Sheffield, UK. [555]Tropical & Infectious Disease Unit, Royal Liverpool University Hospital, Liverpool, UK.

[556]Liverpool Clinical Trials Centre, University of Liverpool, Liverpool, UK. [557]Public Health Scotland, Edinburgh, UK. [558]Centre for Health Informatics, Division of Informatics, Imaging and Data Science, School of Health Sciences, Faculty of Biology, Medicine and Health, University of Manchester, Manchester Academic Health Science Centre, Manchester, UK. [559]EPCC, University of Edinburgh, Edinburgh, UK. [560]ISARIC, Global Support Centre, COVID-19 Clinical Research Resources, Epidemic diseases Research Group, Oxford (ERGO), University of Oxford, Oxford, UK. [561]Institute of Infection, Veterinary and Ecological Sciences, Faculty of Health and Life Sciences, University of Liverpool, Liverpool, UK.

**The 23andMe COVID-19 Team**

**Janie F. Shelton[562], Anjali J. Shastri[562], Chelsea Ye[562], Catherine H. Weldon[562], Teresa Filshtein-Sonmez[562], Daniella Coker[562], Antony Symons[562], Jorge Esparza-Gordillo[563], Stella Aslibekyan[562] & Adam Auton[562]**

[562]23andMe, Sunnyvale, CA, USA. [563]Human genetics R&D, GSK Medicines Research Centre, Target Sciences R&D, Stevenage, UK.

# Methods

## Hospitalization meta-analysis

The hospitalized phenotype includes patients who were hospitalized with a laboratory-confirmed SARS-Cov2 infection. In this analysis we included GenOMICC, GenOMICC Brazil, GenOMICC Saudi Arabia, ISARIC4C, HGIv6 B2 phenotype with subtraction of GenOMICC data, SCOURGE hospitalized versus population and mild cases, and 23andMe broad respiratory phenotype. A summary description of each analysis is given above, a table with the included studies can be found in Supplementary Table 14 and an extended description can be found in Supplementary Table 1.

## Critical illness meta-analysis

The critically ill COVID-19 group included patients who were hospitalized owing to symptoms associated with laboratory-confirmed SARS-CoV-2 infection and who required respiratory support or whose cause of death was associated with COVID-19. In the critical illness analysis, we included GenOMICC, patients with critical illness from ISARIC4C, HGIv6 phenotype A2 with subtraction of GenOMICC data, SCOURGE severity grades 3 and 4 versus population controls, and 23andMe respiratory support phenotype. A summary description of each analysis can be found above, a table with the included studies can be found in Supplementary Table 13 and an extended description can be found in Supplementary Table 1.

## Meta-analyses

All meta-analyses across studies were performed using a fixed-effect inverse-variance weighting method and control for population stratification in the METAL software[23]. Allele frequency was calculated as the average frequency across studies with the METAL option AVERAGE-FREQ. $P$ values for heterogeneity in effect sizes between studies were calculated using a Cochran's $Q$-test implemented in METAL. For variants in the same position with different REF and ALT alleles across studies, the GenoMICC variant in the European population was selected and the rest were removed. Finally, variants with switched ALT and REF alleles between HGIv6 and GenOMICC were also removed on the basis of differences in allele frequency of the alternative allele. Variants were annotated to the closest genes using dbsnp v.b151 GRCh38p7 and bionrRt R package (v.2.46.3)[24]. As each single-nucleotide polymorphism (SNP) of the meta-analysis can be present in different subsets of cohorts, there may be large differences in $P$ values in SNPs with a high level of linkage disequilibrium, which may have an effect on downstream analyses. For this reason, variants that were not present in one of the three biggest studies—GenOMICC European ancestry, HGIv6 or SCOURGE—were filtered out from post-GWAS analysis.

## Conditional analysis

We performed a step-wise conditional analysis to find independent signals. As European-specific data are not available in some cohorts but European ancestry is largely predominant (87.2% of cases with critical illness), we performed the conditional analysis using a European reference panel and the meta-analysis results of the whole cohort. To perform the conditional analysis, we used the GCTA (v.1.9.3) --cojo-slct function[25]. The parameters for the function were $P = 5 \times 10^{-8}$, a distance of 10,000 kb and a co-linear threshold of 0.9 (ref. 26), and the reference population for the conditional analysis was individuals of European ancestry with whole-genome sequence available in the GenOMICC study and whole genomes from the 100,000 Genomics England project[2].

## Credible set fine-mapping

We performed fine-mapping using the SuSiE model[27] to construct credible sets for the independent signals identified using conditional analysis. As for conditional analysis, we used a European reference panel and the meta-analysis results of the whole critical illness cohort.

We performed analyses in 1 Mb windows centred on the lead variants identified through conditional analysis. In cases in which windows for multiple variants overlapped, they were joined into a single window. For each window, we fitted the SuSiE summary statistics model setting the expected number of independent signals to the number of identified though conditional analysis. Models for three windows did not converge in 500 iterations and have been excluded. As a reference, we used the publically available linkage disequilibrium information for non-Finish Europeans from the GNOMAD 2.1.1 release. Full data for all variants included in credible sets are included in Supplementary Table 5.

## Gene-level analysis

We performed an analysis summarizing the genetic associations at the gene level using the mBAT-combo method[28]. We used the COVID 'all critical cohorts' meta-analysis (GenOMICC, HGIv6 phenotype A2, SCOURGE and 23andMe) summary statistics. As this is a trans-ethnic meta-analysis, we used a mixed ancestry linkage disequilibrium reference panel, consisting of 3,202 1000 Genomes phase 3 samples. We considered a list of protein-coding genes with unique ensemble gene ID based on the release from GENCODE (v.40) for hg38, which can be found on the mBAT-combo website (https://yanglab.westlake.edu.cn/software/gcta/#mBAT-combo). A gene region was taken to span 50 kb upstream to 50 kb downstream of the gene's untranslated regions.

## Sex-stratified meta-analysis

To test for differences in genetic effects, we performed sex-stratified GWAS of the COVID-19 critical illness phenotype in the European ancestry GenOMICC WGS and genotyped cohorts and SCOURGE. We then performed a meta-analysis for each sex following the same methods as for the main analysis. We tested for differences in effects between the meta-analyses of the two sexes following previously described methods[29].

## Mendelian randomization

GSMR[5] was performed. We used the COVID 'all critical cohorts' meta-analysis (GenOMICC, HGIv6 phenotype A2, SCOURGE and 23andMe) as the outcome, protein expression quantitative-trait loci (pQTLs) from ref. 30 and RNA expression quantitative-trait loci (eQTLs) from eQTLgen[31] (2019-12-23 data release) as exposures, and 10,000 individuals of European ancestry randomly sampled from the UK Biobank as the linkage disequilibrium reference cohort (50,000 for linkage disequilibrium to missense variant plots). GSMR was performed for all exposures for which we were able to identify two or more suitable SNPs. SNPs were chosen to meet the following criteria: (1) SNP to exposure association $P < 5 \times 10^{-8}$; (2) linkage disequilibrium clumping lead SNPs only ($\pm 1$ Mb, $r^2 < 0.05$); (3) SNP not removed by HEIDI-outlier filtering (for the removal of SNPs with evidence of horizontal pleiotropy) at the default threshold value of 0.01. eQTLGen effect sizes and standard errors were estimated as described in supplementary note 2 of ref. 32. We considered as significant those exposure–outcome pairs with FDR < 0.05.

## TWAS analysis

To perform TWAS analysis in GTExv8 tissues[33], we used the MetaXcan framework and the GTExv8 eQTL and sQTL MASHR-M models available for download online (http://predictdb.org/) and the 'all critical cohorts' meta-analysis. We first calculated individual TWAS for whole blood and lungs using the S-PrediXcan function[34,35]. We next performed a metaTWAS including data from all tissues to increase the statistical power using s-MultiXcan[36]. We applied Bonferroni correction to the results to choose significant genes and introns for each analysis.

## Monocyte gene expression

To detect eQTLs, untreated primary monocytes were prepared from 174 healthy individuals of Northern European (British) ancestry recruited

through the Oxford Biobank. Poly(A) RNA was paired-end 100 bp sequenced in the Oxford Genome Centre using the Illumina HiSeq-4000 machines (median = 47,735,438 reads per sample). Reads were aligned to CRGh38/hg38 using HISAT2 with the default parameters. High mapping quality reads were selected on the basis of MAPQ score using bamtools. Duplicate reads were marked and removed using picard (v.1.105). Samtools was used to pass through the mapped reads and calculate statistics. Read count information was generated using HTSeq and normalized using DESeq2. Sample contamination and swaps were detected by comparing the imputed SNP-array genotypes with genotypes called from RNA-seq using verifyBamID. Genotyping was performed with Illumina HumanOmniExpress with coverage of 733,202 separate markers. Genotypes were pre-phased with SHAPEIT2, and missing genotypes were imputed with PBWT. Poly(A) RNA was paired-end sequenced at the Oxford Genome Centre using the Illumina HiSeq-4000 machines. vcftools (v0.1.12b) was applied on genetic variation data in the form of variant call format (VCF) files to filter out indels and SNPs with a minor allele frequency of less than 0.04.

TWAS analysis for monocyte data was performed using genotyping and monocyte RNA-sequencing data from 174 individuals. Using a region of 500 kb around each gene, we calculated gene expression models using the Fusion R package[37]. For each gene, three models were calculated adding as covariates the two first principal components calculated from the genotype: blup, elastic networks and lasso. The model with a better $r^2$ between predicted and measured expression in a fivefold cross-validation was chosen. Then SNP genetic heritability was calculated for the 500 kb region for each gene and those genes with a nominal significant SNP heritability estimate ($P \leq 0.01$) were chosen for the TWAS analysis. Summary statistics for the 'all critical cohorts' meta-analysis and the best model for each gene were then used to perform the TWAS.

### Colocalization

Significant genes in the TWAS and metaTWAS were selected for a colocalization analysis using the coloc R package. The lead SNPs and a region of 200 Mb around the gene were used to colocalize with significant genes in the TWAS with eQTL summary statistics data on the region from GTExv8 lung, GTExv8 whole blood, eQTLgen or monocyte eqtl. As in our previous analysis[2], we first performed a sensitivity analysis of the posterior probability of colocalization (PPH4) on the prior probability of colocalization (P12), going from P12 = $10^{-8}$ to P12 = $10^{-4}$, with the default threshold being P12 = $10^{-5}$. eQTL signal and GWAS signals were deemed to colocalize if these two criteria were met: (1) at P12 = $5 \times 10^{-5}$ the probability of colocalization PPH4 > 0.5; and (2) at P12 = $10^{-5}$ the probability of independent signal (PPH3) was not the main hypothesis (PPH3 < 0.5). These criteria were chosen to allow eQTLs with weaker $P$ values, owing to lack of power in GTEx v.8, to be colocalized with the signal when the main hypothesis using small priors was that there was not any signal in the eQTL data.

### Effect comparison

We compared the estimates of effect sizes between the individual GWASs used in the meta-analysis, for all variants that were genome-wide significant in at least one of the individual GWASs. To this end, we regressed the effects obtained using critical illness and hospitalization in the SCOURGE and 23andMe cohorts, as well as the HGI meta-analyses on the effect estimates obtained using the GenOMICC cohort. To account for estimation errors present in both the dependent and independent variables of the regression we used orthogonal distance regression[38].

### Weight of studies

To calculate the weight of GenOMICC, we downloaded the leave-one-out data of HGIv7. As the meta-analysis is performed using a variance-weighted method, we can recover the variance for each SNP as $v = \frac{1}{\text{s.e.}^2}$,

for the meta-analysis of all of the cohorts and for each one of the leave-one-out analysis. The total weight is $w_{\text{tot}} = \frac{1}{v}$ and the weight leaving out a specific study is $w_{\text{loo}} = \frac{1}{v_{\text{loo}}}$. The weight of a cohort is then $w_{\text{tot}} - w_{\text{loo}}$. We calculated the weight for each the significant SNPs in our analysis for each study and normalized it using the total weight. Finally, we calculated the mean and s.d. from the significant SNPs for each cohort.

### Forest plots

To compare effects between cohorts, we first performed a trans-ancestry meta-analysis for GenOMICC and 23andMe using METAL[23]. Then, we used the metagen and forest functions of the meta R package to produce forest plots for critical illness and hospitalization separately.

### Reporting summary

Further information on research design is available in the Nature Portfolio Reporting Summary linked to this article.

### Data availability

Downloadable summary data are available through the GenOMICC data site (https://genomicc.org/data). Summary statistics are available, but without the 23andMe summary statistics, except for the 10,000 most significant hits, for which full summary statistics are available. The full GWAS summary statistics for the 23andMe discovery dataset will be made available through 23andMe to qualified researchers under an agreement with 23andMe that protects the privacy of the 23andMe participants. For further information and to apply for access to the data, see the 23andMe website (https://research.23andMe.com/dataset-access/). All individual-level genotype and whole-genome sequencing data (for both academic and commercial uses) can be accessed through the UKRI/HDR UK Outbreak Data Analysis Platform (https://odap.ac.uk). A restricted dataset for a subset of GenOMICC participants is also available through the Genomics England data service. Monocyte RNA-seq data are available under the title 'Monocyte gene expression data' within the Oxford University Research Archives (https://doi.org/10.5287/ora-ko7q2nq66). Sequencing data will be made freely available to organizations and researchers to conduct research in accordance with the UK Policy Framework for Health and Social Care Research through a data access agreement. Sequencing data have been deposited at the European Genome–Phenome Archive (EGA), which is hosted by the EBI and the CRG, under accession number EGAS00001007111.

### Code availability

Code to calculate the imputation of $P$ values on the basis of SNPs in linkage disequilibrium is available at GitHub (https://github.com/baillielab/GenOMICC_GWAS).

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

**Acknowledgements** We thank the patients and their loved ones who volunteered to contribute to this study at one of the most difficult times in their lives, and the research staff in every intensive care unit who recruited patients at personal risk during the most extreme conditions ever witnessed in most hospitals. GenOMICC was funded by Sepsis Research (the Fiona Elizabeth Agnew Trust), the Intensive Care Society, a Wellcome Trust Senior Research Fellowship (to J.K.B., 223164/Z/21/Z), the Department of Health and Social Care (DHSC), Illumina, LifeArc, the Medical Research Council, UKRI, a BBSRC Institute Program Support Grant to the Roslin Institute (BBS/E/D/20002172, BBS/E/D/10002070 and BBS/E/D/30002275) and UKRI grants MC_PC_20004, MC_PC_19025, MC_PC_1905 and MRNO2995X/1. A.D.B. acknowledges funding from the Wellcome PhD training fellowship for clinicians (204979/Z/16/Z), the Edinburgh Clinical Academic Track (ECAT) programme. This research is supported in part by the Data and Connectivity National Core Study, led by Health Data Research UK in partnership with the Office for National Statistics and funded by UK Research and Innovation (grant MC_PC_20029). Laboratory work was funded by a Wellcome Intermediate Clinical Fellowship to B.F. (201488/Z/16/Z). We acknowledge the staff at NHS Digital, Public Health England and the Intensive Care National Audit and Research Centre who provided clinical data on the participants; and the National Institute for Healthcare Research Clinical Research Network (NIHR CRN) and the Chief Scientist's Office (Scotland), who facilitate recruitment into research studies in NHS hospitals, and to the global ISARIC and InFACT consortia. GenOMICC genotype controls were obtained using UK Biobank Resource under project 788 funded by Roslin Institute Strategic Programme Grants from the BBSRC (BBS/E/D/10002070 and BBS/E/D/30002275) and Health Data Research UK (HDR-9004 and HDR-9003). UK Biobank data were used in the GSMR analyses presented here under project 66982. The UK Biobank was established by the Wellcome Trust medical charity, Medical Research Council, Department of Health, Scottish Government and the Northwest Regional Development Agency. It has also had funding from the Welsh Assembly Government, British Heart Foundation and Diabetes UK. The work of L.K. was supported by an RCUK Innovation Fellowship from the National Productivity Investment Fund (MR/R026408/1). J.Y. is supported by the Westlake Education Foundation. SCOURGE is funded by the Instituto de Salud Carlos III (COV20_00622 to A.C., PI20/00876 to C.F.), European Union (ERDF) 'A way of making Europe', Fundación Amancio Ortega, Banco de Santander (to A.C.), Cabildo Insular de Tenerife (CGIEU0000219140 'Apuestas científicas del ITER para colaborar en la lucha contra la COVID-19' to C.F.) and Fundación Canaria Instituto de Investigación Sanitaria de Canarias (PIFIISC20/57 to C.F.). We also acknowledge the contribution of the Centro Nacional de Genotipado (CEGEN) and Centro de Supercomputación de Galicia (CESGA) for funding this project by providing supercomputing infrastructures. A.D.L. is a recipient of fellowships from the National Council for Scientific and Technological Development (CNPq)-Brazil (309173/2019-1 and 201527/2020-0). We thank the members of the Banco Nacional de ADN and the GRA@CE cohort group; and the research participants and employees of 23andMe for making this work possible. A full list of contributors who have provided data that were collated in the HGI project, including previous iterations, is available online (https://www.covid19hg.org/acknowledgements).

**Author contributions** E.P.-C., K. Rawlik, K.M., S.K., C.P.P., J.F.W., V.V., M.A., A.D.L., E.J.P., R.C., A.C., A.F., L.M., K. Rowan, A.C.P., A.L., S.C.H. and J.K.B. contributed to design. E.P.-C., K. Rawlik, A.D.B., T.Q., Y.W., I.N., G.A.M., M.Z., L.K., A.K., A.R., T.M., J.Y., A.L., B.F., S.C.H. and J.K.B. contributed to data analysis. E.P.-C., K. Rawlik, I.N., A.K., A.R., J.M., C.D.R., A.L., B.F. and S.C.H. contributed to bioinformatics. E.P.-C., K. Rawlik, I.N., G.A.M., M.Z., A.K., J.M., C.D.R., R.T., D. McAuley, A.N., M.G.S., B.F., S.C.H. and J.K.B. contributed to writing and reviewing the manuscript. I.N., F.G., W.O., K.M., S.K., D. Maslove, A.N., M.G.S., J.K., M.S.-H., C.S., C.H., P.H., L.L., D. McAuley, H.M., P.J.M.O., C.B., T.W., A.T., C.F., J.A.R., A.R.-M., P.L., C.P.P., A.F., L.M., K. Rowan, A.L., B.F. and S.C.H. contributed to oversight. F.G. and W.O. contributed to project management. F.G., W.O. and J.K.B. contributed to ethics and governance. K.M., A.F. and L.M. contributed to sample handling and sequencing. C.P.P., K. Rowan, S.C.H. and J.K.B. contributed to conception. C.P.P., J.F.W., V.V., M.A., A.D.L., E.J.P., R.C., A.C., K. Rowan and A.C.P. contributed to reviewing the manuscript. K. Rowan and A.L. contributed to clinical data management. J.K.B. contributed to scientific leadership.

**Competing interests** The authors declare no competing interests.

**Additional information**
**Correspondence and requests for materials** should be addressed to J. Kenneth Baillie.

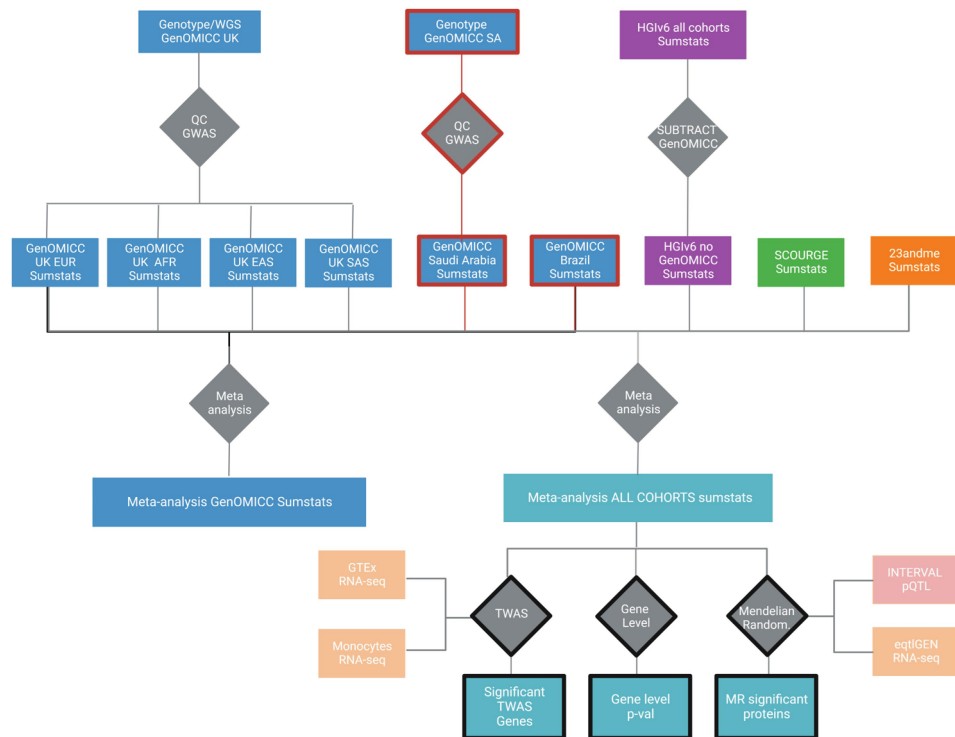

**Extended Data Fig. 1 | Pipeline of meta-analysis and post-GWAS analyses.** Red border indicates that the data is only available for the hospitalized phenotype, while a black border indicates that the analysis was performed for the critical illness phenotype.

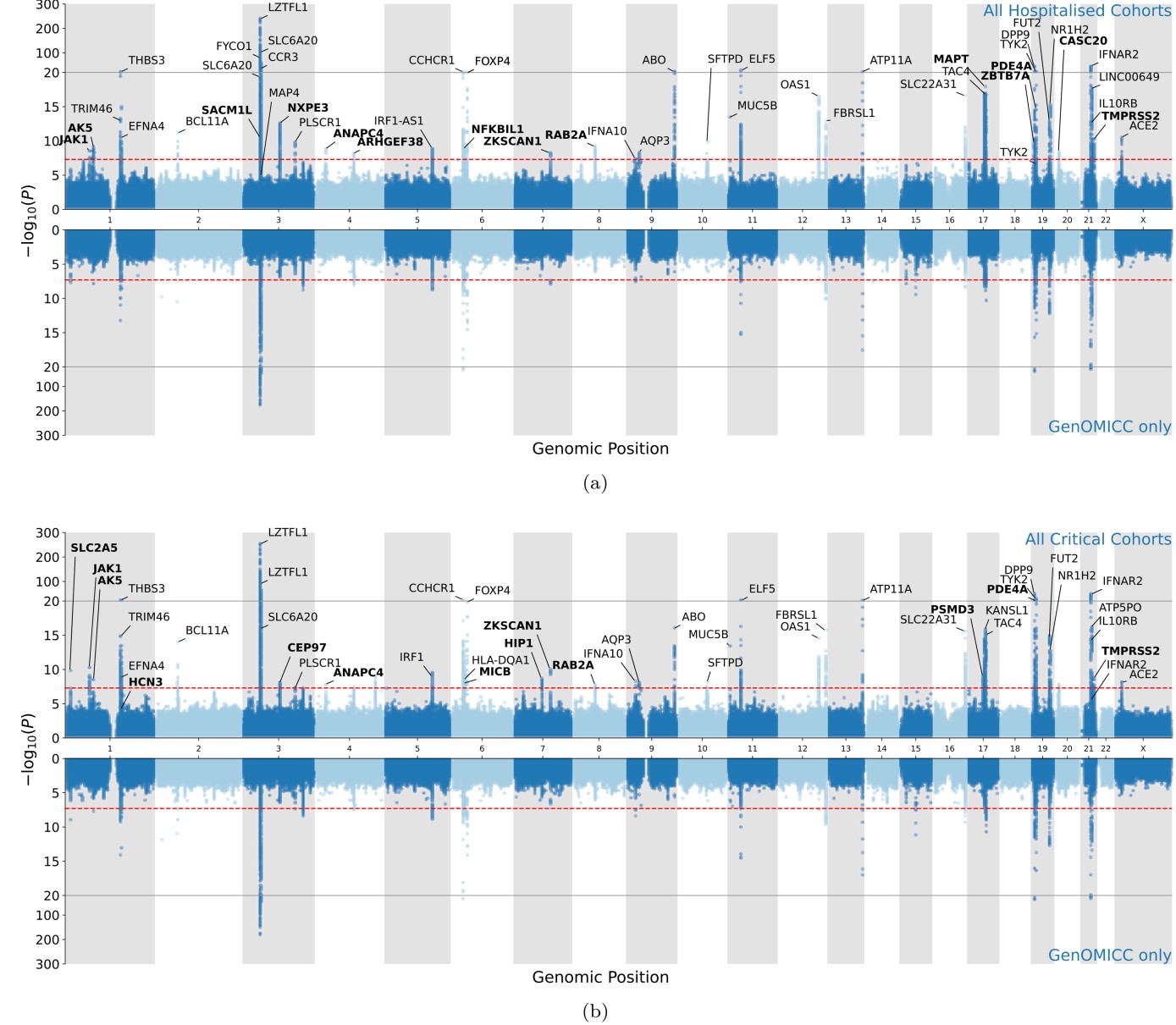

**Extended Data Fig. 2 | Miami plots.** Meta-analysis results are shown for a) critical and b) hospitalized phenotypes. In each plot results obtained using all cohorts are shown at the top and using GenOMICC data only at the bottom. Independent lead variants in the analyses of all cohorts are annotated with associated genes. Genome-wide significant associations that have not been previously reported are indicated in bold.

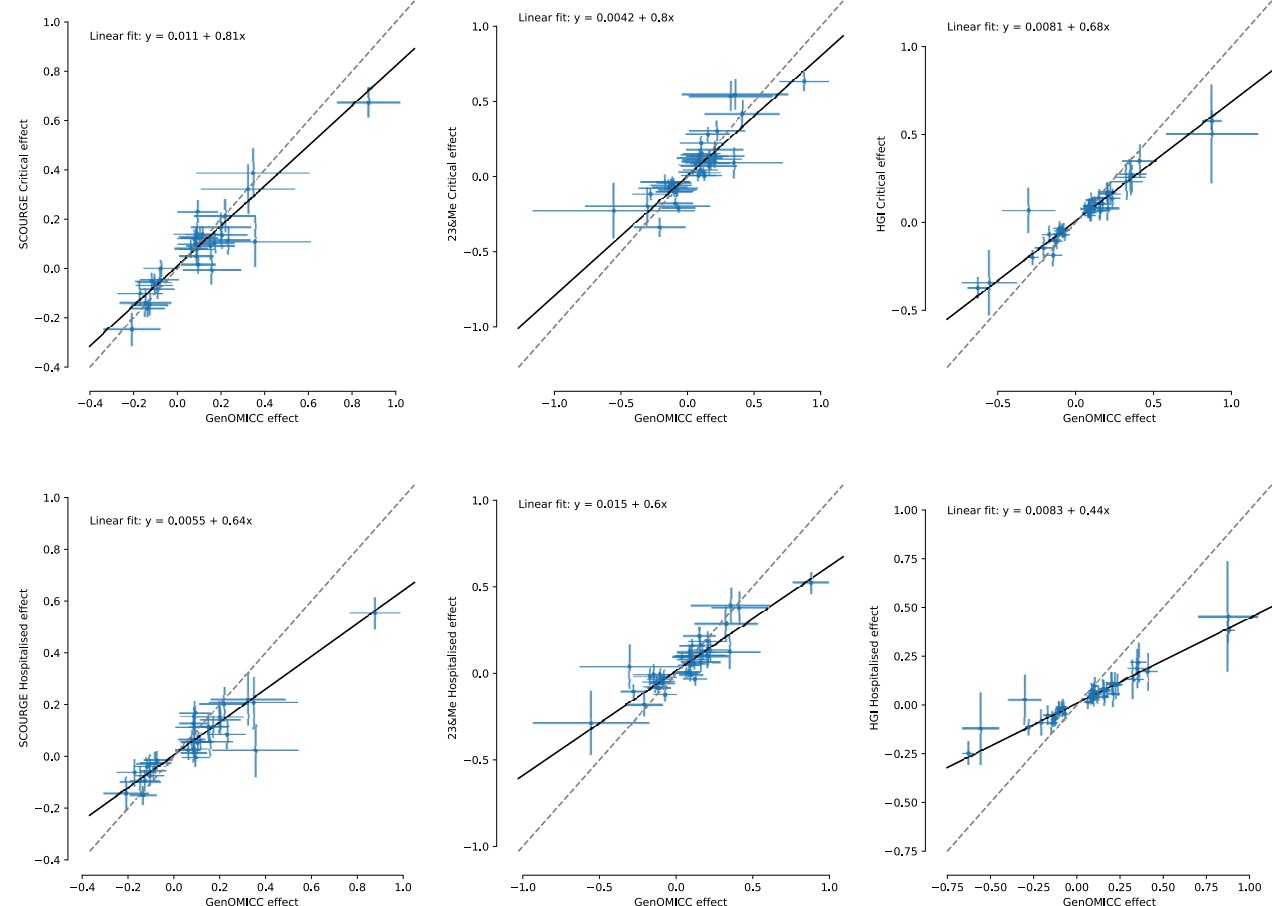

**Extended Data Fig. 3 | Comparison of effect size estimates.** GenOMICC is compared with the critical and hospitalized phenotype definitions in the SCOURGE, 23andMe, and HGI analyses. The black line indicates the best linear fit, given by the equation in each plot, obtained using Orthogonal Distance Regression to account for estimation errors in both sets of effects in the comparison.

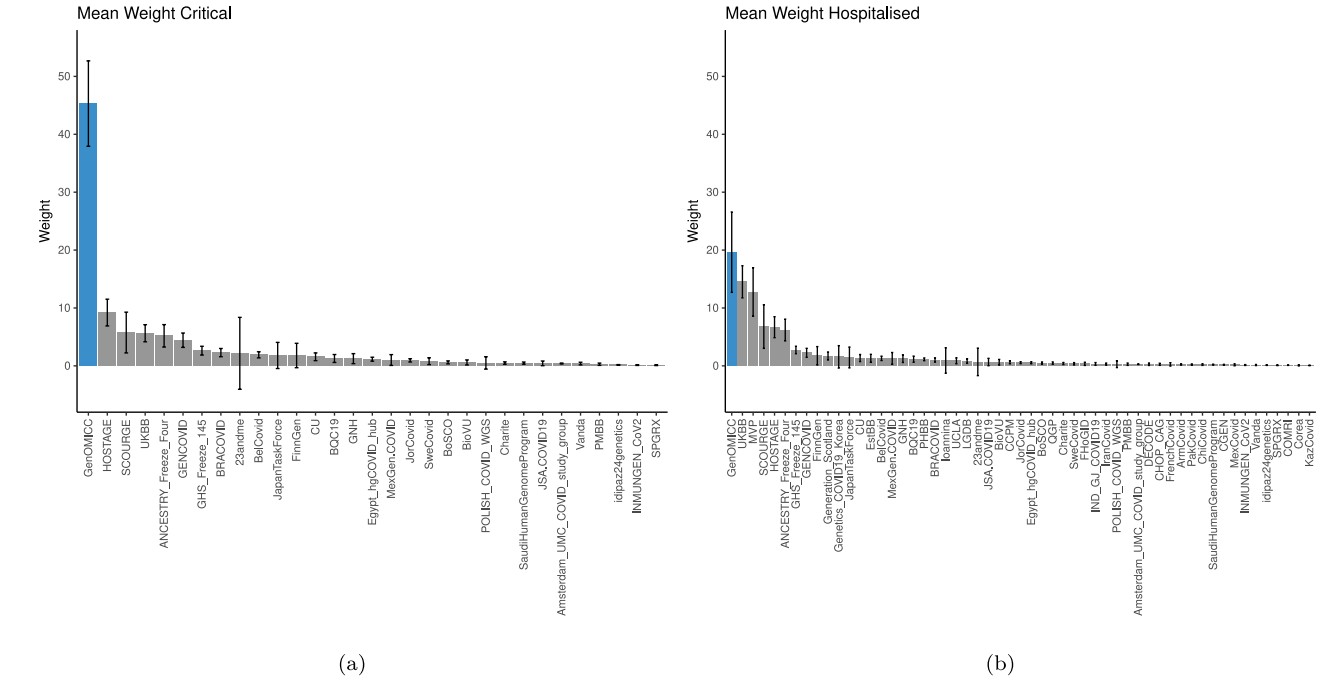

(a)

(b)

**Extended Data Fig. 4 | Study weightings for (a) critical and (b) hospitalized COVID-19.** Mean +/− standard deviation of weights assigned to each data source in meta-analyses for all significant SNPs.

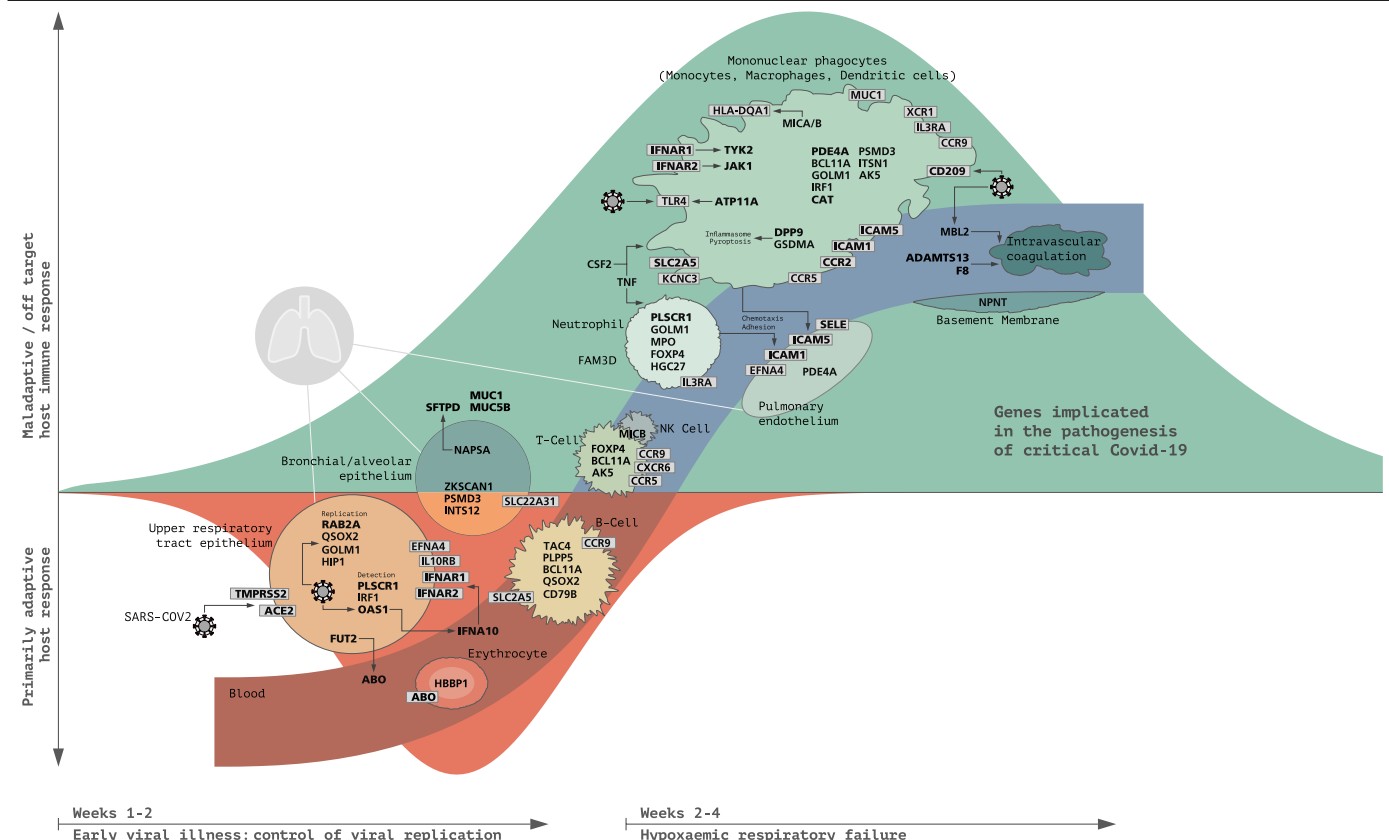

**Extended Data Fig. 5 | Cartoon showing postulated roles for genes and mediators implicated in the pathogenesis of critical COVID-19 by GenOMICC GWAS, TWAS and Mendelian randomization.** Postulated roles for genetic variants are shown in a highly simplified format to illustrate potential roles in pathogenesis, with the shaded background indicating the hypothetical impact of the host immune response over time[17]. Host immune processes are divided into those that are thought to play a role in controlling viral replication early in disease (orange section, showing "adaptive" response), and those implicated in driving hypoxaemic respiratory failure later in disease (green section, showing "maladaptive" response). Bold type gene names indicate a higher level of confidence in both the gene identification and the biological role.

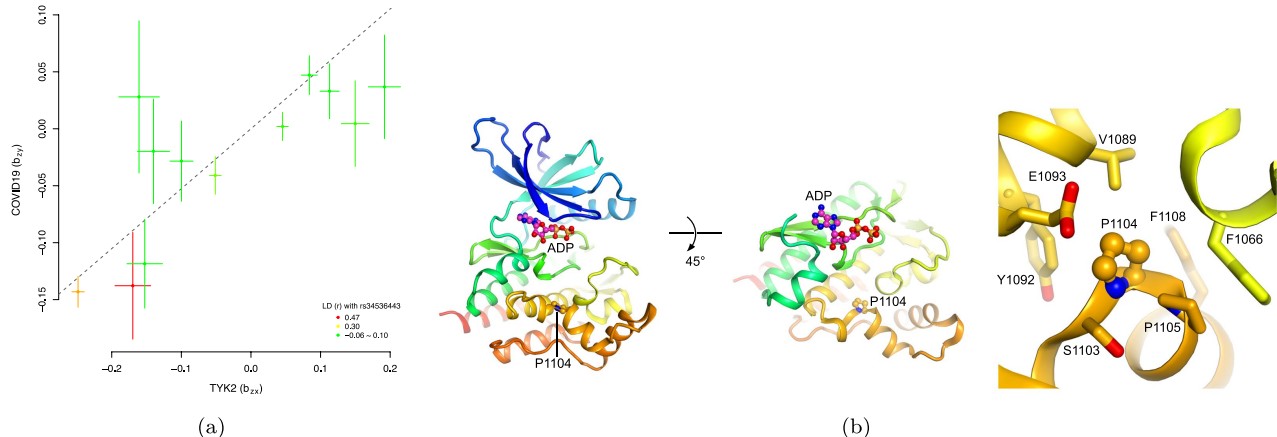

(a)    (b)

**Extended Data Fig. 6 | Functional genomics analyses for TYK2.** (a) Effect size plot for effect of multiple variants on TYK2 expression (eQTLgen, x-axis) against increasing susceptibility to critical COVID-19 ($\beta_{xy} = 0.53$; $P_{xy} = 1.2 \times 10^{-23}$). Colour shows linkage disequilibrium (LD) with the missense variant rs34536443. (b) Crystal structure of TYK2 kinase domain (Protein Data Bank ID 4GVJ[39]) in two views that differ by a 45° rotation around a horizontal axis. The side chain of P1104 is shown as connected spheres with a nitrogen atom coloured in blue. Carbon, oxygen, nitrogen and phosphorus atoms of ATP are shown as magenta, red, blue and orange connected spheres, respectively. The N-terminal region of the kinase domain is not shown in the second view for clarity. The right-most panel shows a close view of P1104 and neighbouring residues with their side chains shown as sticks. Numbering of residues corresponds to UniProtKB entry P29597. P1104 is in the catalytic kinase domain and proximal to the ATP-binding site; TYK2 P1104A is catalytically impaired[40].

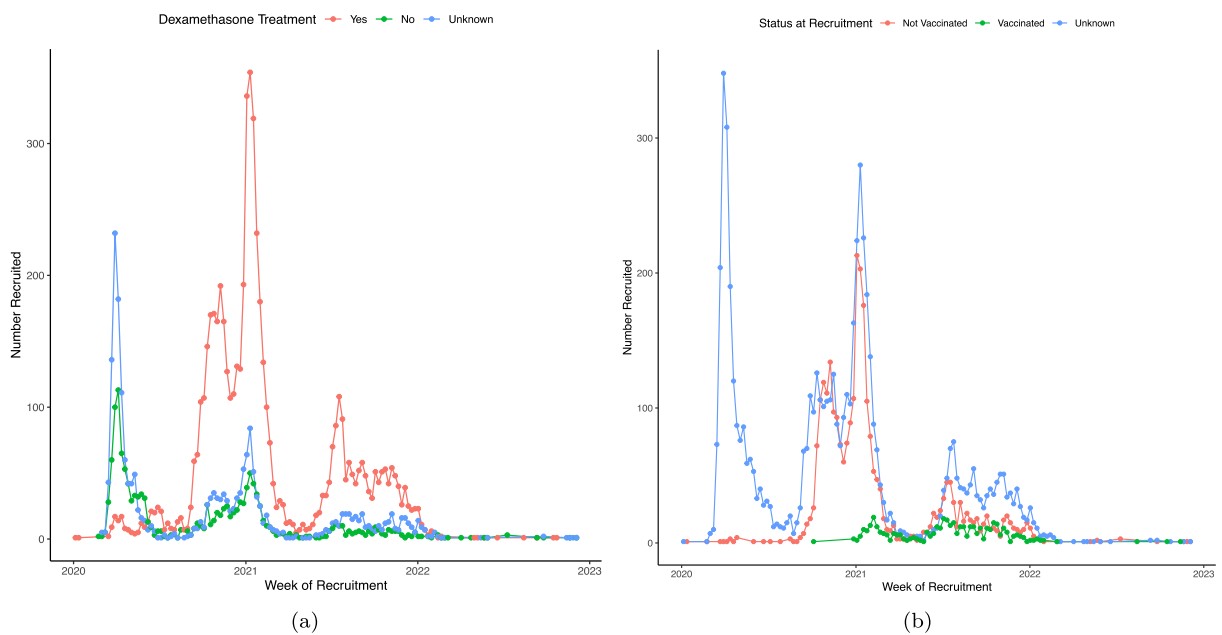

**Extended Data Fig. 7 | Steroid treatment and vaccination status.** Data are shown for a subset of GenOMICC cases who were also recruited to the ISARIC4C study in the UK.

## Extended Data Table 1 | pQTL GSMR results table

| Exposure | b | se | p | $N_{snp}$ |
|---|---|---|---|---|
| FAM3D[†] | 0.17 | 0.016 | $4.82 \times 10^{-24}$ | 6 |
| GOLM1[†] | 0.24 | 0.026 | $2.43 \times 10^{-20}$ | 3 |
| ICAM5[†] | -0.17 | 0.022 | $8.30 \times 10^{-14}$ | 8 |
| QSOX2 | 0.12 | 0.017 | $1.26 \times 10^{-13}$ | 12 |
| ABO[†] | 0.1 | 0.014 | $2.51 \times 10^{-13}$ | 16 |
| CD209[†] | 0.13 | 0.018 | $2.53 \times 10^{-12}$ | 8 |
| ICAM1[†] | 0.069 | 0.01 | $3.05 \times 10^{-11}$ | 15 |
| SELE[†] | -0.095 | 0.015 | $1.42 \times 10^{-10}$ | 8 |
| CREB3L4 | 0.15 | 0.025 | $1.12 \times 10^{-9}$ | 4 |
| PDGFRL[†] | 0.071 | 0.013 | $1.98 \times 10^{-8}$ | 16 |
| ICAM5[†] | -0.098 | 0.018 | $2.72 \times 10^{-8}$ | 6 |
| MPO | -0.14 | 0.038 | $2.67 \times 10^{-4}$ | 4 |
| TLR4.LY96[†] | -0.082 | 0.024 | $7.32 \times 10^{-4}$ | 5 |
| ADAMTS13 | 0.092 | 0.027 | $8.19 \times 10^{-4}$ | 5 |
| MBL2 | -0.042 | 0.013 | $1.34 \times 10^{-3}$ | 15 |

Proteins significantly linked to COVID-19 severity (false discovery rate (FDR) <0.05). Exposure: protein name; b: effect-size estimate of the protein on COVID-19 severity from GSMR; se: standard-error of b; p: p-value of the GSMR result; N: Number of independent SNPs included in the analysis. [†]: indicates proteins with GSMR evidence previously presented in Kousathanas et al.[2].

## Extended Data Table 2 | Replication table

| chr.pos.b38 | rsid | Gene | SCGG Jul-2020 | GenOMICC1 Aug-2020 | HGI1 May-2021 | GenOMICC2 Sep-2021 | HGI2 Nov-2021 | GenOMICC3 March-2022 |
|---|---|---|---|---|---|---|---|---|
| 1:9067157 | rs2478868 | SLC2A5 | | | | | | ✓ |
| 1:64948270 | rs12046291 | JAK1 | | | | | | ✓ |
| 1:77501822 | rs71658797 | AK5 | | | | | | ✓ |
| 1:155066988 | rs114301457 | EFNA4 | | | | ✓ | | ✓ |
| 1:155175305 | rs7528026 | TRIM46 | | | | ✓ | ✓ | ✓ |
| 1:155197995 | rs41264915 | THBS3 | | | ✓ | ✓ | ✓ | ✓ |
| 1:155278322 | rs11264349 | HCN3 | | | | | | ✓ |
| 2:60480453 | rs1123573 | BCL11A | | | | ✓ | | ✓ |
| 3:45796521 | rs2271616 | SLC6A20 | | | ✓ | ✓ | ✓ | ✓ |
| 3:45818159 | rs17713054* | LZTFL1 | ✓ | ✓ | ✓ | ✓ | ✓ | ✓ |
| 3:45873093 | rs35482426 | LZTFL1 | ✓ | ✓ | ✓ | ✓ | ✓ | ✓ |
| 3:101790631 | rs11706494 | NXPE3 | | | | | | ✓ |
| 3:146522652 | rs343314 | PLSCR1 | | | | ✓ | | ✓ |
| 4:25446871 | rs7664615 | ANAPC4 | | | | | | ✓ |
| 4:105673359 | rs72670002 | ARHGEF38 | | | | | | ✓ |
| 4:167824478 | rs1073165 | . | | | | | | ✓ |
| 5:132422622 | rs2269821 | IRF1-AS1 | | | | ✓ | | ✓ |
| 6:31153455 | rs111837807 | CCHCR1 | | ✓ | ✓ | | ✓ | ✓ |
| 6:31571991 | rs2071590 | LTA | | | | | | ✓ |
| 6:32702687 | rs2858305 | HLA-DQA1 | | | | ✓ | | ✓ |
| 6:41522644 | rs41435745 | FOXP4 | | | ✓ | ✓ | ✓ | ✓ |
| 7:75623396 | rs1179620 | HIP1 | | | | | | ✓ |
| 7:100032719 | rs2897075 | ZKSCAN1 | | | | | | ✓ |
| 8:60532539 | rs13276831 | RAB2A | | | | | | ✓ |
| 9:21206606 | rs28368148 | IFNA10 | | | | ✓ | | ✓ |
| 9:33425186 | rs60840586 | AQP3 | | | | | | ✓ |
| 9:133271182 | rs879055593 | ABO | ✓ | | ✓ | | ✓ | ✓ |
| 10:79946568 | rs721917 | SFTPD | | | | | ✓ | ✓ |
| 11:1219991 | rs35705950 | MUC5B | | | | | ✓ | ✓ |
| 11:34482745 | rs61882275 | ELF5 | | | | ✓ | ✓ | ✓ |
| 12:112919637 | rs2660 | OAS1 | | ✓ | ✓ | | ✓ | ✓ |
| 12:132481571 | rs11614702 | FBRSL1 | | | | ✓ | ✓ | ✓ |
| 13:112881427 | rs12585036 | ATP11A | | | | ✓ | | ✓ |
| 16:89196249 | rs117169628 | SLC22A31 | | | | ✓ | ✓ | ✓ |
| 17:40003082 | rs12941811 | PSMD3 | | | | | | ✓ |
| 17:46085231 | rs8080583 | KANSL1 | | | ✓ | ✓ | ✓ | ✓ |
| 17:49863303 | rs77534576 | TAC4 | | | ✓ | ✓ | ✓ | ✓ |
| 19:4717660 | rs12610495 | DPP9 | | ✓ | ✓ | ✓ | ✓ | ✓ |
| 19:10352442 | rs34536443 | TYK2 | | ✓ | ✓ | ✓ | ✓ | ✓ |
| 19:10414696 | rs142770866 | PDE4A | | | | | | ✓ |
| 19:48702915 | rs516246 | FUT2 | | | | ✓ | | ✓ |
| 19:50374423 | rs35463555 | NR1H2 | | | | | ✓ | ✓ |
| 20:6489447 | rs2326788 | CASC20 | | | | | | ✓ |
| 21:33229937 | rs188401375 | IFNAR2 | | ✓ | ✓ | ✓ | ✓ | ✓ |
| 21:33237639 | rs9636867 | IFNAR2 | | ✓ | ✓ | ✓ | ✓ | ✓ |
| 21:33287378 | rs8178521 | IL10RB | | | | ✓ | ✓ | ✓ |
| 21:33980963 | rs76608815 | ATP5PO | | | | ✓ | | ✓ |
| 21:41479527 | rs915823 | TMPRSS2 | | | | | | ✓ |
| 23:15523993 | rs35697037 | ACE2 | | | | | ✓ | ✓ |

Each report of genome-wide significant associations with severe COVID-19 is shown, with associations that were first reported by the GenOMICC consortium are highlighted in blue.

# Reporting Summary

## Statistics

For all statistical analyses, confirm that the following items are present in the figure legend, table legend, main text, or Methods section.

| n/a | Confirmed | |
|---|---|---|
| ☐ | ☒ | The exact sample size (*n*) for each experimental group/condition, given as a discrete number and unit of measurement |
| ☐ | ☒ | A statement on whether measurements were taken from distinct samples or whether the same sample was measured repeatedly |
| ☐ | ☒ | The statistical test(s) used AND whether they are one- or two-sided<br>*Only common tests should be described solely by name; describe more complex techniques in the Methods section.* |
| ☐ | ☒ | A description of all covariates tested |
| ☐ | ☒ | A description of any assumptions or corrections, such as tests of normality and adjustment for multiple comparisons |
| ☐ | ☒ | A full description of the statistical parameters including central tendency (e.g. means) or other basic estimates (e.g. regression coefficient) AND variation (e.g. standard deviation) or associated estimates of uncertainty (e.g. confidence intervals) |
| ☐ | ☒ | For null hypothesis testing, the test statistic (e.g. *F*, *t*, *r*) with confidence intervals, effect sizes, degrees of freedom and *P* value noted<br>*Give P values as exact values whenever suitable.* |
| ☒ | ☐ | For Bayesian analysis, information on the choice of priors and Markov chain Monte Carlo settings |
| ☒ | ☐ | For hierarchical and complex designs, identification of the appropriate level for tests and full reporting of outcomes |
| ☐ | ☒ | Estimates of effect sizes (e.g. Cohen's *d*, Pearson's *r*), indicating how they were calculated |

*Our web collection on statistics for biologists contains articles on many of the points above.*

## Software and code

Policy information about availability of computer code

| | |
|---|---|
| Data collection | Illumina i-scan platform, GenomeStudio Analysis software v2.0.3, GSAMD-24v3-0-EA_20034606_A1.bpm manifest and cluster file provided by manufacturer for GenOMICC UK genotyping. Affymetrix Saub1 chip (Saudi Arabia) and Axiom Analysis suite 5.1.1.1 manifest and cluster file provided by manufacturer for GenOMICC Saudi Arabia. Other datasets were collected previously and Summary statistics used. In order to detect eQTLs, untreated primary monocytes were prepared from 174 healthy individuals of Northern European (British) ancestry recruited via the Oxford biobank. Poly-A RNA was paired-end 100bp sequenced in the Oxford Genome Centre using Illumina Hiseq-4000 machines (median = 47,735,438 reads per sample). Genotyping was performed with Illumina HumanOmniExpress with coverage of 733,202 separate markers. |
| Data analysis | GenomeStudio v2.03, Plink 1.9, Plink 2.0, King 2.1, R v3.6.3, python v3.7, GATK 4.0, USC liftover,  GCTA v1.92 ,REGENIE v3.1.2, metal (2018-08-28),  BCFtools 1.9, QCtools 1.3, FlashPCA2,  admixture,1.3.0 SMR/HEIDI v1.03, MetaXcan v0.6.5, MiniMac4 v1.0, MetaSubtract package v1.60, Rv4.1.0, FUSION (commit e1ba5f7), python v3.10, HISAT2, bamtools, picard v.1.105, verifyBAMID,  samtools, SHAPEIT2, PBWT, vcftools v.0.1.12b, HTseq, Deseq2. |

For manuscripts utilizing custom algorithms or software that are central to the research but not yet described in published literature, software must be made available to editors and reviewers. We strongly encourage code deposition in a community repository (e.g. GitHub). See the Nature Portfolio guidelines for submitting code & software for further information.

# Data

Policy information about availability of data

All manuscripts must include a data availability statement. This statement should provide the following information, where applicable:

- Accession codes, unique identifiers, or web links for publicly available datasets
- A description of any restrictions on data availability
- For clinical datasets or third party data, please ensure that the statement adheres to our policy

Downloadable summary data are available through the GenOMICC data site https://genomicc.org/data. Summary statistics will be available without including 23andme summary statistics, except for the 10,000 most significant hits which will have full summary statistics available. The full GWAS summary statistics for the 23andMe discovery data set will be made available through 23andMe to qualified researchers under an agreement with 23andMe that protects the privacy of the 23andMe participants. Please visit https://research.23andMe.com/dataset-access/ for more information and to apply to access the data.

All individual-level genotype and whole genome sequence data (for both academic and commercial uses) can be accessed through the UKRI/HDR UK Outbreak Data Analysis Platform https://odap.ac.uk. A restricted dataset for a subset of GenOMICC participants is also available through the Genomics England data service.

Monocyte RNA-seq data is available under the title "Monocyte gene expression data" within the Oxford University Research Archives. DOI: 10.5287/ora-ko7q2nq66 (http://dx.doi.org/10.5287/ora-ko7q2nq66)

# Human research participants

Policy information about studies involving human research participants and Sex and Gender in Research.

| Reporting on sex and gender | Sex was asked in the study, and then confirmed by genotype. When there was a discordance between self-reported sex and genotyped sex the sample was removed from the study.<br>Sex-specific analysis were performed for both males and females but did not give any significant results<br>Overall the study included 45472 cases from which ~40% are female. As data comes from summary statistics from other analysis, it has not been possible to calculate the exact number of females and males. |
|---|---|
| Population characteristics | Cases have tested positive for Covid-19 and needed hospitalisation or ICU admission. Controls come from different sources, people which experienced mild (non-hospitalised) Covid-19 or population controls from different Biobanks. In GenOMICC Brazil, mild cases were selected from serological studies of SARS-COV2 infection and PCR test results among health professionals and the general population. In GenOMICC Saudi Arabia, mild controls were selected after a positive PCR test. SCOURGE population controls were extracted from Spanish DNA Biobank and the GR@CE consortium. Participants in 23and me analysis provided informed consent and answered surveys online according to 23andme human subjects research protocol.<br>Untreated primary monocytes were prepared from 174 healthy individuals from British ancestry via the Oxford Biobank |
| Recruitment | Cases were recruited by different studies in hospitals. All participants gave informed consent. Mild controls were recruited on the basis of having experienced mild or assymptomatic Covid-19.<br>For GenOMICC UK and ISARIC4C population controls were used from UK Biobank (project 788), or 100,000 genomes from genomics england. SCOURGE project used controls from Spanish DNA Biobank and the GR@CE consortium |
| Ethics oversight | GenOMICC Scotland: Scotland A Research Ethics Committee 15/SS/0110.<br>GenOMICC England/Wales/Northern Ireland Coventry and Warwickshire Research Ethics Committtee 19/WM/0247.<br>GenOMICC Brazil (BraCovid) National Research Ethics Committee (CONEP) and Ethics Committee for the Analysis of Research Projects at HC FMUSP (CAPPesq) 5025/20/054.<br>GenOMICC Saudi Arabia IRB at King Abdullah International Medical Research Center.<br>ISARIC4C England/Wales/Northern Ireland South Central Oxford C Research Ethics Committee 13/SC/0149.<br>ISARIC4C Scotland Scotland A Research Ethics Committee 20/SS/0028.<br>SCOURGE Galician Ethical Committee 2020/197.<br>23andme Ethical and Independent Review Services \url{http://www.eandireview.com}.<br>Covid-19 HGI Multiple ethics committees (https://www.covid19hg.org/).<br>Oxford biobank approved by South Central - Oxford C Research Ethics Committee, reference 18/SC/0588 |

Note that full information on the approval of the study protocol must also be provided in the manuscript.

# Field-specific reporting

Please select the one below that is the best fit for your research. If you are not sure, read the appropriate sections before making your selection.

☒ Life sciences    ☐ Behavioural & social sciences    ☐ Ecological, evolutionary & environmental sciences

For a reference copy of the document with all sections, see nature.com/documents/nr-reporting-summary-flat.pdf

# Life sciences study design

All studies must disclose on these points even when the disclosure is negative.

| | |
|---|---|
| Sample size | cases 45472, controls 2929541 |
| Data exclusions | no exclusions |
| Replication | As we meta-analysed all public data available for Covid-19, to verify replicability of the findings we performed a heterogeneity test between studies, using a Cochran's Q-test |
| Randomization | Not relevant to the study |
| Blinding | Not relevant to the study |

# Reporting for specific materials, systems and methods

We require information from authors about some types of materials, experimental systems and methods used in many studies. Here, indicate whether each material, system or method listed is relevant to your study. If you are not sure if a list item applies to your research, read the appropriate section before selecting a response.

## Materials & experimental systems

| n/a | Involved in the study |
|---|---|
| ☒ | ☐ Antibodies |
| ☒ | ☐ Eukaryotic cell lines |
| ☒ | ☐ Palaeontology and archaeology |
| ☒ | ☐ Animals and other organisms |
| ☒ | ☐ Clinical data |
| ☒ | ☐ Dual use research of concern |

## Methods

| n/a | Involved in the study |
|---|---|
| ☒ | ☐ ChIP-seq |
| ☒ | ☐ Flow cytometry |
| ☒ | ☐ MRI-based neuroimaging |

