## [Peer Review File · Nature]

Manuscript Title: GWAS and meta-analysis identifies 49 genetic variants underlying critical Covid-19

Reviewer Reports on the Initial Version:

Referees' comments:

Referee #1 (Remarks to the Author):

This is a follow up to two papers from the same consortium that were published in Nature in December 2020 (Pairo-Castineira et al.) and March 2022 (Kousathanas et al.).

Using >20K cases, the authors present the largest meta-analysis of the human genetic determinants of critical illness in COVID-19 (the "Matters arising" by the COVID-19 Host Genetics Initiative, published in August 2022 in Nature, included 9376 critical cases). As such, it has unprecedented power and indeed, the analyses reveal novel host factors involved in host-pathogen interactions and disease progression, some of which representing potential therapeutic targets.

However, the paper has several weaknesses:

- The way the case numbers are presented is confusing. The introduction mentions a total number of 21519 critical cases. However, Suppl. Table 4, which lists the cohorts included in the "critically ill" meta-analysis, includes 24216 patients. I understand the need for brevity in this format, still exclusion criteria and potential overlap between analyses should be clarified.
- Most study participants were genotyped and not sequenced: only 7491 of the 24216 cases are part of a WGS dataset (column 2 of Suppl. Table 4). Those were all analyzed in the second paper of the GenOMICC study (Kousathanas et al.). This means that the new analysis will only be able to identify new association with relatively common genetic variants, present on - or imputable from - genotyping chips, and not with rare and ultra-rare variants, which are more likely to have functional consequence and big effect sizes. This limitation should be clearly explained.
- Page 4, "These variants are, by definition, molecular changes that alter clinical outcomes in Covid-19". This is definitely not true. The 49 reported lead variants show a statistical association with severe COVID-19 status but haven't been demonstrated to be causal.
- A couple of "particularly interesting" results from the transcriptomic analysis are presented, but it is unclear how they were chosen. If I understand the rationale to select potential drug targets, I don't see any specific reason to insist on genes implicated in viral entry rather than on any other gene subset. Also, please explain why, for the JAK1 signal, "the direction of effect is not clear from genetic data".
- A comparison of the results obtained here and by the COVID-19 Host Genetics Initiative (Nature, Aug 2022) would be very helpful – but might be beyond the scope of this short report.
- Several references are wrongly formatted or incomplete

Referee #2 (Remarks to the Author):

Baillie and team provide a global update to their original work evaluating the host genetics of critical COVID-19. They identify 49 genome wide significant associations and use gene expression and protein expression to further understand potential drug targets. This continues to be important work that focuses on those at the more severe end of the spectrum, hospitalization with covid-19. However, there are several comments noted that need expanded explanation so that the proper conclusions can be drawn from this work.

1) the use of admixed populations (like Brazilians) is a welcome addition, but all of the PCA plots

are of everyone together. What we need to see is the case-control match per site. The subtlety of local genetic variation that can affect results can be lost when you put multiple ancestral populations together. For a specific site where the Brazilians or other non-white populations are contributing we need to see the allele frequencies in the different populations. But, we should be seeing those for everyone regardless as there is not homogeneity in European alleles.

2) The authors reference heterogeneity. But, it's not clear what that means. Does that mean heterogeneity of the effect size? If so, how is that considered across different allele frequencies? Or is this classic heterogeneity test used in meta-analysis to determine differences in the direction of the beta?

3) Table 1. This lists all the reported variants and the citation indicates when the genetic variants was first identified. But, this table should provide more. Are these 49 novel genetic findings for this paper (also...supplementary title says 41)? How many are new to this paper and GenOMICCnew? And of those that are not novel, how many does GenOMICC2 replicate/validate? For example, is EFNA4 only seen in GenOMICC but not in GenOMICnew? Some way to differentiate would be helpful so we can better understand what is new, what is replicated and what is validated.

4) It's important not only that we are identifying new variants/genetic regions but that we are fine mapping those that have been identified through the use of global ancestrally different populations. However, that is not addressed and seems like a missed opportunity to identify causal alleles or putative causal alleles for these genes.

5) The Miami plot is difficult to follow (both in main paper and supplementary) given the density of the plot. For example, it's clear that on chromosome 5 IRF1 is seen in both, but on chromosome 9, ABO is not. This likely is due to differences in the population (for ABO). But then when we look at chromosome 17 which variant are we looking at and which gene? There are multiple noted here but it's not clear which are seen in GenOMICC only. What does bolding mean?

6) no sample sizes are indicated in the table 1 or figure 1 or text. This makes interpretation really hard.

7) in methods. variants were filtered out if not present in the 3 largest studies. This can be problematic and needs to be explained. Was this solely done for power and thus everything in this expanded global study is focused only on European alleles? I raise this because then it should be explicit in the text, and then explained that all the additional genetic evaluations were done with only European ancestry as described (meta, conditional etc). It seems disappointing that the populations were expanded but the approach is still euro centric.

8) Methods: was a fixed effect or random effect meta analysis performed given the variation in cohorts?

9) methods: how was pca done for Brazilians and South Asians? as noted above, we need to see this. Also, an inadvertent line is left in that needs to be removed from this section (clearly intended for coauthors and not reviewers).

10) a Flow chart of the methods of the study would be valuable, as well as an expanded table that shows where the samples came from, sample sizes by case/control status, country of origin, genotyping platform, sex (with numbers by case/control not overall, so sex stratified is interpretable), and if they were a part of GenOMICC or GenOMICCnew.

11) Finally, and most importantly. With the passage of time, treatments and vaccines have been made available and this complicates the overall evaluation of host genetics. Please indicate if the cases or controls have been vaccinated or treated. There is also no summary on comorbidities in

these controls or cases. In the original GenOMICC paper there is at least a brief summary and at the least that should be done here. This matters because: a) comorbidities in the controls suggest that the populations are better matched. A better study would be to propensity match the cases to the controls not just considering ancestry, age and sex but also comorbidities. b) vaccines change the immunity profile of the patients and offer a layer of protection. reporting on if the cases were vaccinated would be the first step to understand if hospitalization was post vaccine immunity. And then of course which vaccines were present per country. c) treatments. If a similarly matched (by age and sex) patient is provided Paxlovid or monoclonal antibodies that keeps them from needing to be hospitalized, that has less to do with genetics than the medicine. And this is why ancestry and access and country become issues again because much of the treatment and vaccine differed by country and not everyone had access equity. Much of this could be overlooked in the early days, but it's very hard to overlook it now and to attribute all of these findings to genetics when they could be spurious or confounded associations.

Author Rebuttals to Initial Comments:

Referee #1 (Remarks to the Author):

Using >20K cases, the authors present the largest meta-analysis of the human genetic determinants of critical illness in COVID-19 (the “Matters arising” by the COVID-19 Host Genetics Initiative, published in August 2022 in Nature, included 9376 critical cases). As such, it has unprecedented power and indeed, the analyses reveal novel host factors involved in host-pathogen interactions and disease progression, some of which representing potential therapeutic targets.

We are grateful for the reviewer’s encouraging remarks.

The way the case numbers are presented is confusing. The introduction mentions a total number of 21519 critical cases. However, Suppl. Table 4, which lists the cohorts included in the “critically ill” meta-analysis, includes 24216 patients. I understand the need for brevity in this format, still exclusion criteria and potential overlap between analyses should be clarified.

We have corrected this error and carefully proof-read the manuscript and supplementary file to check for similar mistakes.

Most study participants were genotyped and not sequenced: only 7491 of the 24216 cases are part of a WGS dataset (column 2 of Suppl. Table 4). Those were all analyzed in the second paper of the GenOMICC study (Kousathanas et al.). This means that the new analysis will only be able to identify new association with relatively common genetic variants, present on - or imputable from - genotyping chips, and not with rare and ultra-rare variants, which are more likely to have functional consequence and big effect sizes. This limitation should be clearly explained.

We have added the following sentence to the discussion to make this limitation clear:

“Our analysis is limited to common variants that are detectable on genotyping arrays and imputation panels.”

Page 4, “These variants are, by definition, molecular changes that alter clinical outcomes in Covid-19”. This is definitely not true. The 49 reported lead variants show a statistical association with severe COVID-19 status but haven’t been demonstrated to be causal.

Our phrasing here was clumsy and implied that we think the variants themselves are causal. We thank the reviewer for drawing our attention to it. We have rephrased this as follows:

“Although most lead variants are not directly causal, they highlight molecular changes that alter clinical outcomes in Covid-19, and hence may have direct therapeutic relevance.”

A couple of “particularly interesting” results from the transcriptomic analysis are presented, but it is unclear how they were chosen. If I understand the rationale to select potential drug targets, I don’t see any specific reason to insist on genes implicated in viral entry rather than on any other gene subset. Also, please explain why, for the JAK1 signal, “the direction of effect is not clear from genetic data”.

We have expanded our description of the evidence supporting each drug that was under active consideration by the independent UK Covid-19 Therapeutics Advisory Panel (UK-CTAP) in a new supplementary

table (Supplementary Table 8). We have added the following sentence in the Discussion to explain our interest in mechanisms of viral entry:

“Our results also demonstrate the capacity of host genetics to reveal core mechanisms of disease.”

We have rewritten the description of JAK1 association to improve clarity.

A comparison of the results obtained here and by the COVID-19 Host Genetics Initiative (Nature, Aug 2022) would be very helpful – but might be beyond the scope of this short report.

We have completed this, together with a comparison with previous reports of new genome-wide significant findings. We have added a new table (Supplementary Table 1) showing which findings are significant in each report. We have reproduced this table above in Table 1 for convenience.

Several references are wrongly formatted or incomplete

We have corrected this.

Referee #2 (Remarks to the Author):

Baillie and team provide a global update to their original work evaluating the host genetics of critical COVID-19. They identify 49 genome wide significant associations and use gene expression and protein expression to further understand potential drug targets. This continues to be important work that focuses on those at the mores severe end of the spectrum, hospitalization with covid-19. However, there are several comments noted that need expanded explanation so that the proper conclusions can be drawn from this work.

- 1) the use of admixed populations (like Brazilians) is a welcome addition, but all of the PCA plots are of everyone together. What we need to see is the case-control match per site. The subtlety of local genetic variation that can affect results can be lost when you put multiple ancestral populations together. For a specific site where the Brazilians or other non-white populations are contributing we need to see the allele frequencies in the different populations. But, we should be seeing those for everyone regardless as there is not homogeneity in European alleles.

We agree with the reviewer that we should have included this in our first submission. We have added PCA plots showing case-control matching for each cohort (Supplementary Figure 47, Supplementary Figure 48, Supplementary Figure 49 and Supplementary Figure 50) and reproduce these here for convenience. For the published cohorts details of PCA calculation and plots can be found in Table 2. We also added citations to the supplementary materials file cohorts.xlsx.

Table 2: References for the cohorts included in the study.

Study	Reference
GenOMICC WGS	2
GenOMICC Brazil	3
HGI v6	4
SCOURGE	5
23andme	6

- 2) The authors reference heterogeneity. But, it's not clear what that means. Does that mean heterogeneity of the effect size? If so, how is that considered across different allele frequencies? Or is this classic heterogeneity test used in meta-analysis to determine differences in the direction of the beta?

To calculate heterogeneity we used a Cochran's Q-test, which tests for differences in effect size across cohorts. . A small heterogeneity p-value in a hit means that effect sizes in the significant snp are not similar across the meta-analysed cohorts. To clarify this point, we change the wording in the meta-analysis methods: "*P-values for heterogeneity in effect sizes between studies were calculated using a Cochran's Q-test implemented in METAL.*"

Figure 1: PCA plot for cases and controls in SCOURGE

Figure 2: PCA plot for cases and controls in GenOMICC Saudi Arabia

- 3) Table 1. This lists all the reported variants and the citation indicates when the genetic variants was first identified. But, this table should provide more. Are these 49 novel genetic findings for this paper (also...supplementary title says 41)? How many are new to this paper and GenOMICCnew? And of those that are not novel, how many does GenOMICC2 replicate/validate? For example, is EFNA4 only seen in GenOMICC but not in GenOMICnew? Some way to differentiate would be helpful so we can better understand what is new, what is replicated and what is validated.

We agree that this suggestion (similar to a comment also from Reviewer 1) will improve the presentation of our results. We have included a new table (Supplementary Table 1, see Table 1 above in this document) in which we show which findings are replicated in each report of a large-scale GWAS of this phenotype. We have also corrected the title of the Supplementary Information, and clarified in the table legends that GenOMICC^{new} refers to the new findings in the current manuscript.

- 4) It's important not only that we are identifying new variants/genetic regions but that we are fine mapping those that have been identified through the use of global ancestrally different populations. However, that is not addressed and seems like a missed opportunity to identify causal alleles or putative causal alleles for these genes.

We agree that this is important and have discussed the challenges in this analysis in our response to the editor above. We completed an additional analysis to fine map each region, with full results presented in Supplementary Figure 38.

- 5) The Miami plot is difficult to follow (both in main paper and supplementary) given the density of the plot. For example, it's clear that on chromosome 5 IRF1 is seen in both, but on chromosome 9, ABO is not. This likely is due to differences in the population (for ABO). But then when we look at chromosome 17 which variant are we looking at and which gene? There are multiple noted here but it's not clear which are seen in GenOMICC only. What does bolding mean?

We have changed the legend to include the following sentence to clarify the bold highlighting: "*Genome-wide significant associations that have not been previously reported are indicated in bold.*"

For detailed browsing of our results, we have now provided expanded data tables and fine mapping plots in Supplementary Figure 38. We will also release an interactive data browser for this data release shortly at genomicc.org/data, as with our previous work.

- 6) no sample sizes are indicated in the table 1 or figure 1 or text. This makes interpretation really hard.

We are very constrained by space in this table but have added sample sizes to the supplementary tables to make a full interpretation possible.

7) in methods. variants were filtered out if not present in the 3 largest studies. This can be problematic and needs to be explained. Was this solely done for power and thus everything in this expanded global study is focused only on European alleles? I raise this because then it should be explicit in the text, and then explained that all the additional genetic evaluations were done with only European ancestry as described (meta, conditional etc). It seems disappointing that the populations were expanded but the approach is still euro centric.

In a meta-analysis with cohorts with such different numbers of individuals, variants in high LD can have very different p-values just because the number of individuals analysed is very different (e.g. a significant variant present in all cohorts can be in high LD with a non-significant variant only present in a small cohort of ~1000 individuals). This can have an important effect in post-GWAS analyses like TWAS or fine-mapping. To minimise this effect, for post-GWAS analysis we only used summary statistics where a large number of individuals had been analysed, although summary statistics were calculated for all variants. Even though the variants chosen come from cohorts of mainly European ancestry (GenOMICC EUR, SCOURGE and HGI), all cohorts have been used in the meta-analysis of these variants, and therefore results include effects from all ancestries. Manhattan plots for the ~23M variants present in any of the studies do not show any extra peak in variants not present in the largest cohorts. Summary statistics for all variants will be made available after publication at genomicc.org/data.

To clarify this point, we changed the methods meta-analysis section:

“Since each SNP of the meta-analysis can be present in different cohorts, there may be large differences of p-values in SNPs with high level of LD, which may have an effect on downstream analyses. For this reason, variants not present in one of the three biggest studies: GenOMICC European ancestry, HGIv6, or SCOURGE were filtered out from post-GWAS analyses.”

8) Methods: was a fixed effect or random effect meta analysis performed given the variation in cohorts?

A fixed effect meta-analysis was performed. As mentioned above, heterogeneity between cohorts is non-significant for most variants (only significant in LZTFL1, presumably due to differences in phenotype definition between cohorts and CHCCR1 in the HLA region). We have clarified this point in the methods as follows:

“All meta-analyses across studies were performed using a fixed effect inverse-variance weighting method and control for population stratification...”

In addition, we have now performed a random effect meta-analysis for the lead snps to enable comparison with the fixed effect, finding reassuringly concordant effect sizes (Table 3), even for those few variants with greater heterogeneity and discordant P-values.

Table 3: Comparison of fixed and random effect model for lead snps.

chr:pos(b38)	rsid	Gene	BETA.fixed	SE.fixed	P.fixed	BETA.random	SE.random	P.random	HetP
1:9067157	rs2478868	SLC2A5	-0.081	0.013	1.5e-10	-0.075	0.018	2.5e-05	0.018
1:64948270	rs12046291	JAK1	0.094	0.014	5.1e-11	0.094	0.014	4.8e-11	0.68
1:77501822	rs71658797	AK5	0.12	0.021	2.9e-09	0.1	0.033	0.0016	0.25
1:155066988	rs114301457	EFNA4	0.87	0.14	1.5e-09	0.87	0.14	1.5e-09	1
1:155175305	rs7528026	TRIM46	0.29	0.036	1.5e-15	0.29	0.045	8e-11	0.46
1:155197995	rs41264915	THBS3	0.2	0.019	7.6e-24	0.2	0.02	7.6e-24	0.73
1:155278322	rs11264349	HCN3	-0.059	0.015	7.3e-05	-0.059	0.015	7.5e-05	0.36
2:60480453	rs1123573	BCL11A	0.11	0.014	1e-14	0.11	0.019	3e-08	0.24
3:45796521	rs2271616	SLC6A20	0.17	0.02	1.1e-16	0.16	0.046	0.00071	0.00013
3:45818159	rs17713054	LZTFL1	0.71	0.021	7.7e-254	0.68	0.089	3.5e-14	8e-15

chr:pos(b38)	rsid	Gene	BETA.fixed	SE.fixed	P.fixed	BETA.random	SE.random	P.random	HetP
3:45873093	rs35482426	LZTFL1	-0.63	0.031	6.1e-91	-0.63	0.031	9.4e-91	0.2
3:101790631	rs11706494	NXPE3	0.075	0.013	9.4e-09	0.075	0.013	9.4e-09	0.58
3:146522652	rs343314	PLSCR1	0.14	0.026	4.6e-08	0.14	0.048	0.0045	0.12
4:25446871	rs7664615	ANAPC4	0.099	0.017	1.5e-08	0.1	0.022	3.6e-06	0.28
4:105673359	rs72670002	ARHGEF38	0.14	0.025	4.4e-08	0.099	0.053	0.061	0.037
4:167824478	rs1073165	.	0.076	0.013	1.1e-09	0.076	0.012	1.1e-09	0.78
5:132422622	rs2269821	IRF1- AS1	0.11	0.018	3e-10	0.11	0.033	0.00073	0.15
6:31153455	rs111837807	CCHCR1	-0.22	0.021	8.6e-26	-0.24	0.053	7.9e-06	2.3e-15
6:31571991	rs2071590	LTA	0.08	0.013	3.1e-10	0.08	0.013	2.8e-10	0.34
6:32702687	rs2858305	HLA- DQA1	-0.076	0.013	2.1e-09	-0.075	0.013	3.2e-08	0.02
6:41522644	rs41435745	FOXP4	0.34	0.037	1.5e-20	0.34	0.05	3e-11	0.45
7:75623396	rs1179620	HIP1	-0.082	0.014	2.3e-09	-0.082	0.014	2.3e-09	0.39
7:100032719	rs2897075	ZKSCAN1	0.081	0.013	8.9e-11	0.081	0.013	9.3e-11	0.9
8:60532539	rs13276831	RAB2A	0.069	0.012	1.7e-08	0.069	0.013	2.6e-07	0.64
9:21206606	rs28368148	IFNA10	-0.53	0.09	5.3e-09	-0.52	0.1	1.6e-07	0.31
9:33425186	rs60840586	AQP3	0.098	0.017	9.7e-09	0.12	0.042	0.0051	0.035
9:133271182	rs879055593	ABO	0.12	0.015	1e-16	0.12	0.015	9.6e-17	0.96
10:79946568	rs721917	SFTPD	-0.076	0.013	7.6e-09	-0.078	0.025	0.0018	0.07
11:1219991	rs35705950	MUC5B	-0.15	0.02	3.8e-14	-0.15	0.02	3.7e-14	0.82
11:34482745	rs61882275	ELF5	-0.12	0.013	1e-22	-0.12	0.016	1.1e-14	0.54
12:112919637	rs2660	OAS1	0.1	0.013	2.8e-15	0.1	0.013	2.9e-15	0.45
12:132481571	rs11614702	FBRSL1	0.1	0.012	2.1e-16	0.1	0.012	2e-16	0.74
13:112881427	rs12585036	ATP11A	0.14	0.015	9.6e-22	0.13	0.025	3.3e-07	0.18
16:89196249	rs117169628	SLC22A31	0.15	0.018	2.6e-16	0.15	0.021	1.2e-12	0.42
17:40003082	rs12941811	PSMD3	-0.075	0.012	1.2e-09	-0.076	0.015	3.5e-07	0.42
17:46085231	rs8080583	KANSL1	-0.12	0.015	1.8e-16	-0.12	0.015	1.6e-16	0.39
17:49863303	rs77534576	TAC4	0.29	0.036	8.7e-16	0.29	0.038	5.2e-14	0.3
19:4717660	rs12610495	DPP9	-0.23	0.015	9.1e-51	-0.21	0.044	1.8e-06	0.0019
19:10352442	rs34536443	TYK2	0.4	0.036	2.2e-28	0.4	0.036	2.1e-28	0.97
19:10414696	rs142770866	PDE4A	0.22	0.023	9.4e-21	0.22	0.023	9.4e-21	0.65
19:48702915	rs516246	FUT2	-0.1	0.013	1.4e-15	-0.091	0.02	7.6e-06	0.4
19:50374423	rs35463555	NR1H2	0.097	0.013	1.9e-13	0.097	0.013	1.8e-13	0.44
20:6489447	rs2326788	CASC20	-0.076	0.013	1.4e-08	-0.076	0.013	1.4e-08	0.99
21:33229937	rs188401375	IFNAR2	-0.3	0.064	3.1e-06	-0.3	0.063	3e-06	0.4
21:33237639	rs9636867	IFNAR2	-0.19	0.013	5.2e-48	-0.19	0.028	4.8e-12	0.03
21:33287378	rs8178521	IL10RB	0.12	0.016	6.1e-15	0.12	0.027	3.9e-06	0.31
21:33980963	rs76608815	ATP5PO	0.17	0.02	7.4e-17	0.17	0.023	3.8e-13	0.19
21:41479527	rs915823	TMPRSS2	0.093	0.016	2.1e-09	0.093	0.016	2.1e-09	0.95
23:15523993	rs35697037	ACE2	0.042	0.0073	6.8e-09	0.042	0.0073	6.7e-09	0.38

9) methods: how was pca done for Brazilians and South Asians? as noted above, we need to see this. Also, an inadvertent line is left in that needs to be removed from this section (clearly intended for coauthors and not reviewers).

We thank the reviewer for drawing our attention to this error, which we have corrected.

The Brazil analysis has now been published and we refer to this paper in the resubmitted manuscript.³ We added the PCA calculation in methods.

“Principal components for cases and controls were calculated with PLINK 2.0. Due to the level of admixture and complex genetic structure present in the Brazilian population, three different ancestry subgroups were defined (European, African, Native American) using k-means clustering.”

For GenOMICC Saudi Arabia, we calculated the PCA with the following methodology:

“To infer ancestry, a first step was done projecting the genotype data into 1000Genomes principal components calculated with GCTA 1.9. Most individuals had admix ancestry according to PCA projection (Supplementary Figure 48). Outliers belonging to East Asian, South Asian or African ancestries were removed from the analysis because there were not enough numbers to perform an ancestry-specific GWAS.

Figure 3: Manhattan plot with all variants included in all cohorts for critical illness meta-analysis

Figure 4: Manhattan plot with all variants included in all cohorts for hospitalised meta-analysis

Then we performed a principal component analysis of the remaining individuals using GCTA 1.9. Individuals with a deviation more than 2sd from the mean in the 10 first principal components were considered outliers and removed from the dataset.”

10) a Flow chart of the methods of the study would be valuable, as well as an expanded table that shows where the samples came from, sample sizes by case/control status, country of origin, genotyping platform, sex (with numbers by case/control not overall, so sex stratified is interpretable), and if they were a part of GenOMICC or GenOMICCnew.

We expanded the tables for cohorts in a supplementary material file cohort.xlsx including genotyping platform, numbers by sex in GWAS included in the sex-specific analysis, country of origin and an indication of which cohorts were part of the previous analysis². We also added a flowcharts of the methodology of the paper in Supplementary Figure 4, reproduced here for convenience.

Figure 5: Flowchart summarising the pipeline of the meta-analysis and post-GWAS analysis of the paper

11) Finally, and most importantly. With the passage of time, treatments and vaccines have been made available and this complicates the overall evaluation of host genetics. Please indicate if the cases or controls have been vaccinated or treated. There is also no summary on comorbidities in these controls or cases. In the original GenOMICC paper there is at least a brief summary and at the least that should be done here. This matters because:

We have summarised such information as is available and included it in the supplementary material. However we note that information on vaccination status and comorbidities status is largely lacking for many cohorts like HGI or 23&me . This includes GenOMICC where detailed information, through linkage to electronic health records, is only available in a subset of individuals who were also recruited to the ISARIC study. Furthermore where information on, for example, comorbidity is available it differs in level of detail between cohorts. Given these limitations we have summarised the following information:

- Comorbidities for the subset of GenOMICC UK cases who were also recruited to the ISARIC study, GenomiccBrazil, and SCOURGE. (see Supplementary Table 3], [Supplementary Table 4, and Supplementary Table 5)
- Distribution of treatment with Dexamethasone over time in cases in the subset of GenOMICC UK cases who were also recruited to the ISARIC study. (see Supplementary Figure 9a)
- Distribution of vaccination status information over time in cases in the subset of GenOMICC UK cases who were also recruited to the ISARIC study. (see Supplementary Figure 9b)

We have reproduced Supplementary Figure 9b here in Figure 6 for convenience.

Figure 6: Steroid treatment and vaccination status for a subset of GenOMICC UK cases who were also recruited to the ISARIC4C study.

More generally a problem when interrogating changes of effects with the passage of time is that these could arise from many confounded processes, including changes in treatment, changing viral strains, changes in vaccination uptake in the population, and a host of other factors which changed over the course of the pandemic. To get a general measure of whether there may be potential changes in host genetic effects for the lead variants identified here, we computed trajectories of case allele frequencies for the variants

identified in the critical illness meta analysis for individual month of recruitment. This analysis was restricted to a subset of genotyped GenOMICC UK cases for whom recruitment date information was available. We do not find any clear patterns for changes in allele frequency (see Supplementary Figure 41). However, we also note that the sample size for this analysis is relatively small making it difficult to identify anything but extreme effects.

Overall our ability to stratify accurately based on clinical variables is limited by the depth of clinical data at our disposal. This concern has two implications. Firstly, we will miss some potentially important signals. This is always the case with large-scale genetic research and we will continue to work to deepen our findings. The second possibility would be more concerning - if there is a risk that the findings we report here are spurious, because of any combination of the potential confounders listed by the reviewer. If this were the case, we would expect that some of the signals we report would be markedly different in populations with differing access to vaccines, treatments and other interventions. We would see this as widespread heterogeneity between studies, which we do not observe.

- a) comorbidities in the controls suggest that the populations are better matched. A better study would be to propensity match the cases to the controls not just considering ancestry, age and sex but also comorbidities.

We note that we have little control over control selection in a majority of studies contributing to the meta analysis. Where we have control over control selection, i.e., within the GenOMICC, information on co-morbidities in cases is restricted to a subset of individuals. However we do propensity match cases and controls on BMI. We have previously performed extensive sensitivity analyses by matching by BMI, and did not find it to have a material effect on GWAS results.^{2,7}

- b) vaccines change the immunity profile of the patients and offer a layer of protection. reporting on if the cases were vaccinated would be the first step to understand if hospitalization was post vaccine immunity. And then of course which vaccines were present per country.

With regards to the effect of vaccination status in particular, the vast majority of GenOMICC cases were recruited in the UK before vaccinations became available (see the new Supplementary Figure 9). Similarly only a minority of GenOMICC Brazil hospitalised cases were vaccinated (3.3%). Most cases in the various cohorts contributing to HGI results were also recruited before widespread vaccine uptake. So while we agree that potential changes in host genetic effects pre- and post-widespread vaccination are of interest we do not think that we have sufficient sample sizes for the latter regime for meaningful inference. However, we have summarised the information on vaccination status of cases and provided information about changes in allele frequency for lead variants over time (Supplementary Figure 41).

- c) treatments. If a similarly matched (by age and sex) patient is provided Paxlovid or monoclonal antibodies that keeps them from needing to be hospitalized, that has less to do with genetics than the medicine. And this is why ancestry and access and country become issues again because much of the treatment and vaccine differed by country and not everyone had access equity. Much of this could be overlooked in the early days, but it's very hard to overlook it now and to attribute all of these findings to genetics when they could be spurious or confounded associations.

We completely agree and have applied for funding to do detailed matching with clinical data in collaboration with Health Data Research UK. We anticipate that it will take a year to complete this work, which we hope will reveal effect modifiers for genetic associations that may well have therapeutic implications.

References

1. Kanai, M. *et al.* Meta-analysis fine-mapping is often miscalibrated at single-variant resolution. *Cell Genomics* **0**, (2022).
2. Kousathanas, A. *et al.* Whole-genome sequencing reveals host factors underlying critical COVID-19. *Nature* **607**, 97–103 (2022).
3. Pereira, A. C. *et al.* Genetic risk factors and COVID-19 severity in Brazil: results from BRACOVID study. *Human Molecular Genetics* **31**, 3021–3031 (2022).
4. Pathak, G. A. *et al.* A first update on mapping the human genetic architecture of COVID-19. *Nature* **608**, E1–E10 (2022).
5. Cruz, R. *et al.* Novel genes and sex differences in COVID-19 severity. *Human Molecular Genetics* **31**, 3789–3806 (2022).
6. Shelton, J. F. *et al.* Trans-ancestry analysis reveals genetic and nongenetic associations with COVID-19 susceptibility and severity. *Nature Genetics* **53**, 801–808 (2021).
7. Pairo-Castineira, E. *et al.* Genetic mechanisms of critical illness in COVID-19. *Nature* **591**, 92–98 (2021).

Reviewer Reports on the First Revision:

Referees' comments:

Referee #1 (Remarks to the Author):

I'd like to thank the authors for their carefully crafted response to my comments and those of the second reviewer. The corrections and additional analyses have made the paper clearer and more thorough. I don't have any additional concern.

Referee #2 (Remarks to the Author):

The authors have done an excellent job of responding to the original comments.

1) The addition of the new table guides the paper, and the extra supplementary figures/tables fill in the blanks that were missing before. However, it's possible the tables are mislabeled? The fine mapping refers to Supplementary Figure 19 in multiple places--but those are MR figures. Supplementary Figure 27 is the fine mapping using SuSie but the legends should include what the dark blue represents (LD or the credible set at a certain threshold). It's also hard to determine what the fine mapping means (did the additional non European studies aid in narrowing down the interval)?

The goal of the fine mapping was to truly refine these regions since you have multi-ethnic data for this extreme outcome. But, it's not clear from these plots if that was possible (noting response to Editors). I think a few lines to the main text of the paper will be helpful, even if it's to say this is challenging to do--because you are the only ones who can do this. Ideally, your work will guide functional work and the more detail you provide the more successful those studies will be.

2) I appreciate the response regarding treatment and vaccine, and fully recognize the difficulty in identifying all sources of selection bias and confounding. However, treatment is on the causal pathway not necessarily as a confounder (if you get treated you won't get as sick) and that affects your outcome and the inferences we would make on putative causal alleles. Given that, you need to acknowledge this in the main manuscript as a weakness that could alter your results in either direction (not just mask true associations). If you do that along with the additional details on limited vaccine uptake in these cohorts etc you will address the complexity of this work. Genetics of this COVID outcome is not simple and it's not simply if you have a variant you will have a serious outcome. Conveying the multiple factors involved, and yet the identification of these variants across study populations is important and acknowledges that the epidemiology of transmission, disease, treatment is complex.

Author Rebuttals to First Revision:

Referees' comments:

We'd like to give our sincere thanks to both reviewers for their thoughtful comments on our paper, which have significantly improved the manuscript.

Referee #1 (Remarks to the Author):

I'd like to thank the authors for their carefully crafted response to my comments and those of the second reviewer. The corrections and additional analyses have made the paper clearer and more thorough. I don't have any additional concern.

Referee #2 (Remarks to the Author):

The authors have done an excellent job of responding to the original comments.

- 1) The addition of the new table guides the paper, and the extra supplementary figures/tables fill in the blanks that were missing before. However, it's possible the tables are mislabeled? The fine mapping refers to Supplementary Figure 19 in multiple places—but those are MR figures. Supplementary Figure 27 is the fine mapping using SuSie but the legends should include what the dark blue represents (LD or the credible set at a certain threshold). It's also hard to determine what the fine mapping means (did the additional non European studies aid in narrowing down the interval)?

Yes, we mislabelled these figures. Many thanks for picking this up. We have corrected this in the revised manuscript.

The goal of the fine mapping was to truly refine these regions since you have multi-ethnic data for this extreme outcome. But, it's not clear from these plots if that was possible (noting response to Editors). I think a few lines to the main text of the paper will be helpful, even if it's to say this is challenging to do—because you are the only ones who can do this.

We have added a paragraph to the fine-mapping results section, explaining the limitations of fine-mapping in this context. We have reproduced it here for convenience:

We have performed fine-mapping to compute credible sets of variants for the identified loci. In a small number of regions, the finemapping algorithm fails to converge, with quality controls pointing at the lack of a suitable linkage disequilibrium panel as a potential cause. Finemapping using meta analysis summary statistics from heterogenous cohorts, as is the case here, is difficult and remains an open problem with existing finemapping methods recently shown to perform poorly in this context.¹ While we note these limitations, we provide the obtained credible sets to aid any further functional work.

Ideally, your work will guide functional work and the more detail you provide the more successful those studies will be.

We appreciate this observation and in response we have added gene browser views to all of the fine mapping figures in the Supplementary Information. We have reproduced one example in Response to Review Figure 1 here.

(a)

(b)

Response to Review Figure 1: Fine mapping results obtained using SuSiE for regions surrounding lead variants. Lead variants are indicated from conditional analysis are indicated by diamonds. Variants contained in 95% credible sets are colored by set membership. Protein coding genes are shown in the lower panels.

2) I appreciate the response regarding treatment and vaccine, and fully recognize the difficulty in identifying all sources of selection bias and confounding. However, treatment is on the causal pathway not necessarily as a confounder (if you get treated you wont get as sick) and that affects your outcome and the inferences we would make on putative causal alleles. Given that, you need to acknowledge this in the main manuscript as a weakness that could alter your results in either direction (not just mask true associations). If you do that along with the additional details on limited vaccine uptake in these cohorts etc you will address the complexity of this work. Genetics of this COVID outcome is not simple and it's not simply if you have a variant you will have a serious outcome. Conveying the multiple factors involved, and yet the identification of these variants across study populations is important and acknowledges that the epidemiology of transmission, disease, treatment is complex.

This is an important caveat and we agree that in theory, our design may assign genetic associations that reflect failure of response to vaccine or drug treatment, to primary susceptibility to Covid-19. The same is true for other environmental factors such as exposure to different pathogens earlier in life. We have redrafted the text as suggested to make this clear to readers, and to explain our reasons for concluding that it is unlikely that vaccine- or therapy-related mechanisms underlie the results we report here.

References

1. Kanai, M. *et al.* Meta-analysis fine-mapping is often miscalibrated at single-variant resolution. *Cell Genomics* **0**, (2022).